



# The three-dimensional life cycle of potential vorticity cutoffs: A global ERA-interim climatology (1979-2017)

Raphael Portmann[1], Michael Sprenger[1], and Heini Wernli[1]

[1]ETH Zurich, Institute for Atmospheric and Climate Science, Zurich, Switzerland

**Correspondence:** Raphael Portmann (raphael.portmann@env.ethz.ch)

**Abstract.** The aim of this study is to explore the nature of potential vorticity (PV) cutoff life cycles. While climatological frequencies of such upper-level cyclonic vortices are well known, their life cycle and in particular their three-dimensional evolution is poorly understood. To address this gap, a method is introduced that allows tracking PV cutoffs as three-dimensional objects. As it is based on isentropic air parcel trajectories, the detailed evolution of the cutoffs on isentropic surfaces, including their associated cross-tropopause mass fluxes, can be analyzed. The novel method is applied to the ERA-interim dataset for the years 1979-2017 and the first global climatology of PV cutoffs is presented that is independent of the selection of a vertical level and identifies and tracks PV cutoffs as three-dimensonal features. More than 40'000 PV cutoff life cycles are identified and analyzed in the almost 40-year data set. Known frequency maxima of PV cutoffs are confirmed and, additionally, bands in subtropical areas in the summer hemispheres and a circumpolar band around Antarctica are identified. A detailed investigation of PV cutoff life cycles in different genesis regions reveals that PV cutoff genesis occurs as a result of distinct Rossby wave breaking scenarios. In addition, there is a remarkable regional variability of PV cutoff mobility, lifetimes and vertical evolution. This regional variability of PV cutoff behaviour can to some extent be explained by differences in cross-tropopause mass fluxes and the varying frequencies of different lysis scenarios on isentropic surfaces, i.e. diabatic decay and reabsorption to the stratospheric reservoir. It is found that, on a global average, reabsorption occurs about as frequently as diabatic decay, but on higher isentropic levels. Further, the temporal link between PV cutoffs and associated surface cyclones is investigated. Novel insights are that (i) the frequency and characteristics of this link strongly depend on the region, and (ii) PV cutoffs are frequently involved in surface cyclogenesis a few days after their formation. PV cutoffs forming from similar Rossby wave breaking scenarios in different regions also show remarkable similarities in other characteristics of their life cycle. Based on that, a classification of PV cutoff life cycles into three types is proposed: Type I forms from anticyclonic Rossby wave breaking equatorward of the jet stream, Type II is the result of anticyclonic Rossby wave breaking followed by cyclonic Rossby wave breaking between the polar and the subtropical jets, and Type III forms from cyclonic wave breaking in the storm track regions. While diabatic decay is particularly frequent for Types I and II, reabsorption dominates for the Type III life cycle.

## 1 Introduction

Meso- to synoptic-scale intrusions of anomalously cold air masses with a closed cyclonic circulation in the mid and upper troposphere frequently occur in all extratropical regions. In the subtropics and mid-latitudes, these upper-level closed cyclones





often form when air from the poleward side of the jet stream is transported far equatorward, forming an elongated cold-air tongue. Subsequently, the tongue breaks up into one or more upper-level cyclonic vortices separated from the main polar reservoir (Appenzeller and Davies, 1992), which are usually located equatorward of the jet stream and isolated from the main westerly flow. This process is known as Rossby wave breaking (RWB, Berggren et al., 1949) and the resulting upper-level

cyclonic systems, which are often termed cutoff lows (COLs), have been first characterized comprehensively by Palmén and Newton (1969). Classically, they are identified as closed geopotential height contours in the mid or upper troposphere (e.g. Bell and Bosart, 1989; Nieto et al., 2005; Munoz et al., 2020). COLs are associated with an anomalously low tropopause, i.e. with stratospheric air in regions that are climatologically tropospheric. Stratospheric air masses exhibit high values of potential vortictiy (PV), typically exceeding 2 PVU [1 PVU = $10^{-6}$m$^{-2}$s$^{-1}$K kg$^{-1}$], whereas tropospheric air masses typically have PV

values below 2 PVU. Therefore, PV is a useful quantity to identify COLs and to describe their behavior (e.g., Hoskins et al., 1985; Browning, 1993; Appenzeller et al., 1996; Wernli and Sprenger, 2007). In the PV framework, COLs are usually identified as closed regions with PV values larger than 2 PVU on an isentropic surface. They are isolated from the main stratospheric high-PV reservoir, and termed stratospheric PV cutoffs [for brevity hereafter PV cutoffs]. PV cutoffs are inherently the same phenomenon as COLs, as illustrated in a case study by Bell and Bosart (1993). But COLs have classically been regarded as

upper-level closed cyclones equatorward of the jet stream, following the picture of Palmén and Newton (1969), whereas the concept of PV cutoffs extends towards the pole, as long as a stratospheric reservoir can be meaningfully defined and separated from the PV cutoff on the considered isentropic level.

Many studies show the high relevance of PV cutoffs for surface weather in specific regions, in particular the formation of (intense) surface cyclones and heavy precipitation events. For example, Porcu et al. (2007) found that more than a third of the

Meditarranean COLs are associated with a surface cyclone. In case studies, PV cutoffs have been reported to be dynamical key elements for the intensification of a subtropical cyclone in the South Atlantic (Mosso Dutra et al., 2017) and the genesis of strong Mediterranean cyclones (Fita et al., 2006). COLs accompany suptropical cyclones in most of the cases in the southwestern South Atlantic (Gozzo et al., 2014) and the eastern North Atlantic (González-Alemán et al., 2015), where they frequently influence the occurrence of tropical transition (Bentley et al., 2017). The work of Mallet et al. (2013) indicated that

PV cutoffs can also lead to the genesis of polar lows. Furthermore, they significantly contribute to (extreme) precipitation in the Mediterranean region (Porcu et al., 2007; Toreti et al., 2016), the Alps (Awan and Formayer, 2017), the Great Plains and western United States (Abatzoglou, 2016; Barbero et al., 2019), Northeast China (Hu et al., 2010), South Africa (Favre et al., 2013), southeastern Australia (Chubb et al., 2011), and Iraq (Al-Nassar et al., 2020). More specifically, they can play a key role in triggering heavy convective storms by favoring the release of conditional instability via destabilization and dynamical forc-

ing (e.g. Romero et al., 2000; Mohr et al., 2020). They can also act as "moisture collectors" (Piaget et al., 2015) if they remain quasi-stationary and repeatedly advect warm and moist air towards a region where it is forced to ascend, i.e., a baroclinic zone or high orography (e.g. Meier and Knippertz, 2009; Grams et al., 2014; Piaget et al., 2015; Raveh-Rubin and Wernli, 2015).

The life cycle of PV cutoffs is strongly governed by diabatic processes. The latent heating associated with cloud formation in the vicinity of PV cutoffs, likely together with turbulent mixing, can modify their evolution and eventually lead to their rapid

diabatic decay, resulting in irreversible mixing of stratospheric air into the troposphere (e.g., Shapiro, 1978; Price and Vaughan,



1993; Wirth, 1995; Gouget et al., 2000; Yates et al., 2013), so-called stratosphere-to-troposphere transport (STT). Portmann et al. (2018) showed that PV cutoffs can also diabatically grow and intensify, likely due to radiative cooling at cloud tops or humidity gradients at the tropopause, potentially leading to troposphere-to-stratosphere transport (TST). Previous studies (e.g.,Sprenger et al., 2007) have shown that PV cutoffs are often associated with both STT and TST, with STT being 2-3 times larger. The modification of PV cutoffs by diabatic processes can strongly depend on the considered isentropic level. This indicates that PV cutoffs are potentially complex three-dimensional features, that can intensify on a higher isentropic level and at the same time decay on a lower isentropic level (Portmann et al., 2018). In addition, as already stated by Hoskins et al. (1985), instead of diabatically decaying, a PV cutoff "*(...) could of course be removed by simply being advected back along isentropic surfaces into the polar stratospheric reservoir*" (a process we call "reabsorption"), but that "*synoptic experience suggests that the chances of this happening in less than a week are small*". Later studies did not pick up this topic and therefore a quantitative estimation of the relative frequencies of diabatic decay and reabsorption is still missing.

The frequencies, geographical distribution, seasonality, and tracks of the "classical" COLs equatorward of the jet stream are well known in both hemispheres. Hotspots in the Northern Hemisphere are the eastern North Pacific, eastern North Atlantic and the Mediterranean, and northern China-Siberia (Bell and Bosart, 1989; Kentarchos and Davies, 1998; Nieto et al., 2005). In the Southern Hemisphere, they tend to occur around the main land masses, i.e. South America, South Africa, and Australia / New Zealand (Fuenzalida et al., 2005; Reboita et al., 2010; Pinheiro et al., 2017). Most COLs have lifetimes of 2-3 days and generally travel eastward. Some studies suggested that they are very mobile and travel hundreds of kilometers (Kentarchos and Davies, 1998; Nieto et al., 2005; Reboita et al., 2010) whereas others point out their quasi-stationarity (Bell and Bosart, 1989; Pinheiro et al., 2017). The first climatology of PV cutoffs was presented by Wernli and Sprenger (2007). Nieto et al. (2008) showed that the geographical distribution of COLs at 200 hPa and PV cutoffs generally agree well, if PV cutoffs are identified at the appropriate isentropic level, depending on the season and region. However, Reboita et al. (2010) showed that also climatologies of COLs strongly depend on the considered pressure level.

While many studies focused on COLs located equatorward of the jet stream, others showed that there exist mid- and upper-level closed cyclones poleward of the jet stream, e.g. over the Hudson Bay, south of Greenland, and the northern Pacific (Bell and Bosart, 1989; Parker et al., 1989; Kentarchos and Davies, 1998; Wernli and Sprenger, 2007; Munoz et al., 2020). Munoz et al. (2020) provided the first climatology of COLs that covers both hemispheres (albeit restricted to the mid-latitudes) and captured the classical COLs at lower latitudes but also COLs at higher latitudes by considering two pressure levels (200 hPa and 500 hPa). Focusing on similar systems in polar regions, Kew et al. (2010) investigated positive PV anomalies in the lowermost stratosphere and Hakim and Canavan (2005) local minima of tropopause-level potential temperature, which are sometimes termed tropopause polar vortices (TPV, Cavallo and Hakim, 2010).

Their relevance for surface cyclones, precipitation, and STT in many regions on the globe explains the high research interest in PV cutoffs in the last three decades. However, despite the large number of climatological studies, a global quantitative analysis of their importance for these aspects is lacking. Also, a climatological perspective on their three-dimensional life cycles and their modification by diabatic processes, including the frequencies of decay and reabsorption, is missing. A reason for this is that current climatologies are restricted by and strongly depend on the selection of a vertical level. Also, they are not global





and many focus on features equatorward of the jet stream only. This study aims to compile a climatology of PV cutoffs that fills these knowledge gaps and provides a basis for a comprehensive global analysis of PV cutoff life cycles, their diabatic modification, and their surface impacts. While COLs have been tracked previously, the climatology presented in this study is the first that explicitly tracks PV cutoffs. For the tracking, a novel method is introduced that is based on air parcel trajectories

and allows quantifying cross-tropopause mass fluxes.

The identification and tracking of PV cutoffs is introduced in detail in Sect. 2. In Sect. 3, a global climatological overview of PV cutoff frequencies, their genesis and lysis is presented. This is followed by a comprehensive analysis of various novel aspects of their life cycle in Sect. 4. Section 5 discusses links to the literature, summarizes the main conclusions and provides an outlook for further research topics that could be addressed with this climatological dataset of PV cutoffs.

## 105   2   Data and Methods

### 2.1   Data

All analyses in this study are based on the ERA-interim dataset (Dee et al., 2011) for the years 1979-2017. Data are available every 6 h on 60 vertical levels and have been interpolated from T255 spectral resolution to a regular grid with a horizontal resolution of 1°. PV is computed from the primary ERA-interim variables. Horizontal winds and PV are interpolated onto a

stack of isentropic levels from 290-350 K with a 5 K interval. For the same time period, masks of stratospheric and tropospheric PV streamers based on the method of Wernli and Sprenger (2007), and of cyclones identified and tracked according to Wernli and Schwierz (2006) are retrieved from the dataset described by Sprenger et al. (2017).

### 2.2   PV cutoff identification

The PV perspective is adopted in this study because it offers several advantages to identify and track COLs compared to other

approaches. First, the identification of PV cutoffs as closed 2 PVU contours on isentropic surfaces is unambiguous, conceptually simple, in principle does not require additional criteria or variables [as for example the methods by Nieto et al. (2005) and Pinheiro et al. (2017)], does not depend on the hemisphere considered [as the approach by Munoz et al. (2020)], and therefore strongly reduces the methodological sensitivity [as exists for COL identification, see Pinheiro et al. (2019)]. Second, the invertibilty principle of PV allows for a very intuitive interpretation of the effect of PV cutoffs on the surrounding atmo-

sphere (Hoskins et al., 1985). And third, PV is conserved on isentropic surfaces, which means that (i) the movement of PV cutoffs is quasi-adiabatic rendering a tracking on isentropic surfaces comparably straightforward, and (ii) deviations from the adiabatic advection of the PV cutoff are indicators of diabatic processes. However, as for any other previously used approach, the restriction of the identification to single levels would neglect the fact that PV cutoffs are often highly three-dimensional features. Therefore, in this study, PV cutoffs are identified and tracked as three-dimensional objects within a stack of isentropic

levels from 290-350 K in 5 K intervals, extending the approach by Portmann et al. (2018). This method allows us to investigate PV cutoffs in the subtropics, where they are most frequent on isentropic levels around 330-350 K (Wernli and Sprenger, 2007),





the mid-latitudes (310-330 K), and polar regions (290-310 K). Note that with this choice of levels the identification of features close to the pole is limited, as there some features occur on even lower isentropic levels, in particular in winter. The tropopause in TPVs over the Canadian Arctic, for example, reaches isentropic levels below 280 K (Cavallo and Hakim, 2010).

Our identification of PV cutoffs starts on single isentropic levels, essentially using the algorithm by Wernli and Sprenger (2007) with the modification that, here, the stratospheric reservoir does not necessarily have to encompass the pole, but is defined as the largest area bounded by a closed 2 PVU contour on each isentrope. This becomes particularly relevant at lower isentropic levels and towards the pole, because there the size of the stratospheric reservoir decreases and often does not encompass the pole. The method of Wernli and Sprenger (2007) has the major drawback that it identifies also features with PV larger than

2 PVU produced by diabatic processes (e.g. within extratropical cyclones) or frictional forces near high topography, i.e. features that are not of stratospheric origin. Therefore, the labeling algorithm of Skerlak et al. (2015) is used here, which assigns to each grid point a label that classifies it as stratospheric if it is three-dimensionally connected to the stratospheric reservoir and if it has a specific humidity of less than $0.1 \, \mathrm{g \, kg^{-1}}$. This label can be used do separate true PV cutoffs from diabatically produced PV features. Then, the PV cutoffs on the different isentropic levels are clustered if they overlap with each other

and hence form a three-dimensional PV cutoff. Finally, PV cutoffs larger than $5 \cdot 10^6 \, \mathrm{km^2}$ (about half the area of the US) at a certain isentropic level are removed from that level to avoid the identification of very large PV cutoffs, which often occur on higher isentropic levels. The resulting three-dimensional PV cutoffs are in the following referred to as PV cutoff objects.

## 2.3 Lagrangian PV cutoff tracking

The tracking takes advantage of the material conservation of PV, i.e. the quasi-adiabatic movement of PV cutoffs. There are some similarities to the tracking developed by Kew et al. (2010) to track PV anomalies in the lowermost stratosphere, which is based on advection of the PV anomalies by the isentropic wind. But the tracking presented in this study uses isentropic air parcel trajectories started from each grid point within the PV cutoff and calculated foreward for 6 hours. The final positions of these short trajectories can be regarded as "adiabatic forecast" of the PV cutoff six hours later, and it serves to accurately

track the cutoff in time. In addition, the deviation of the observed evolution from this adiabatic forecast can be used to quantify cross-tropopause transport. This method to quantify cross-tropopause mass fluxes is conceptually similar to the approach by Gray (2006). As another advantage compared to previous methods of COL tracking, the trajectory-based approach also works in regions with strong advection, for example near the jet stream, when consecutive features do not overlap spatially. The tracking connects PV cutoff objects to non-branching tracks, i.e. without merging and splitting, and consists first of tracking

on isentropic surfaces, and second, a connection of isentropic tracks to 3D tracks. These two steps are illustrated in Figs. 1 and 2, respectively, and are now discussed in detail.

### 2.3.1 Tracking on isentropic surfaces

A 3D PV cutoff object consists of one or more 2D PV cutoffs, one on each isentropic level. The 2D PV cutoffs at a given time $t_0$ are referred to as *parents* (grey features in Fig. 1). Isentropic tracks are constructed forward in time, by allowing only





one successor per track (subsequently referred to as *child*, green features labeled accordingly in Fig. 1 a-d), i.e. in the case of a splitting of a PV cutoff, the smaller part is ignored. This substantially eases the analysis of the tracks. From each *parent*, 6-hourly isentropic forward trajectories are started from each grid point (black crosses in Fig. 1), using the Lagrangian analysis tool Lagranto (Wernli and Davies, 1997; Sprenger and Wernli, 2015). A 2D PV cutoff at time $t_0+6\,\mathrm{h}$ is a potential *child* if it inherits at least one air parcel from the considered *parent*. In the following, a variety of situations is considered that may occur during this step.

In the most simple situation, the movement of the 2D PV cutoff during this time interval is perfectly adiabatic (Fig. 1a) and all trajectories arrive within a 2D PV cutoff (the *child*) at the same level at time $t_0+6\,\mathrm{h}$ (blue crosses in Fig. 1a). In a situation with diabatic activity (Fig. 1b), the adiabatic forecast of the PV cutoff (blue and red crosses in Fig. 1b) may deviate from reality at time $t_0+6\,\mathrm{h}$ (blue and orange crosses / the green feature in Fig. 1b). In this case, some trajectories end up outside of the *child* (red crosses in Fig. 1b), showing that the 2D PV cutoff shrinks due to STT. Also, the 2D PV cutoff may grow due to TST (orange cross in Fig. 1b). Note that, as a limitation of this step, orange grid points could also represent merging of stratospheric PV that detaches from the stratospheric reservoir with the PV cutoff and red grid points could be merging of PV cutoff air with the stratospheric reservoir. We assume that these events are relatively rare such that the resulting overestimation of STT and TST is small. If, additionally to STT and TST, splitting occurs (Fig. 1c), the 2D PV cutoff is selected as *child* that inherits most air parcels from the *parent* (lower green feature in Fig. 1c). The trajectories arriving within the other 2D PV cutoff(s) at time $t_0+6\,\mathrm{h}$ (blue-gray crosses and upper green feature in Fig. 1c) are counted as shrinking due to splitting. It may also occur that two (or more) *parents* merge to one *child* (Fig. 1d). In this case, the *child* is attributed to the *parent* (*parent 1* in Fig. 1d) from which it inherits more air parcels (number of blue crosses vs. number of gray crosses). The trajectories that the *child* inherits from the other parent(s) (*parent 2* in Fig. 1d) are counted as growth due to merging and the track ends for the other parent(s).

If the track of a 2D PV cutoff does not end via merging to another 2D PV cutoff, it does so either via complete diabatic decay (Fig. 1e), or (complete or partial) reabsorption to the stratospheric reservoir (Fig. 1f). If all trajectories from the *parent* arrive in a region with PV<2 PVU (red crosses in Fig. 1e), complete diabatic decay (involving STT) occurs. Air parcels arriving in a region with PV>2 PVU but not within a PV cutoff are counted as reabsorption (blue crosses in Fig. 1f). In the case of merging, air parcels from the parent(s) for which the track ends (*parent 2* in Fig. 1d) and that end up in the child are also considered as reabsorption.

Once all parents at time $t_0$ have been considered, the same is repeated for the subsequent time interval. In this way, tracks are continued for all 2D PV cutoffs identified as *child* in the previous time step, and new tracks are initialized for all other 2D PV cutoffs.

### 2.3.2 Construction of 3D tracks

After isentropic tracks are constructed, PV cutoff objects are concatenated to tracks representing the three-dimensional evolution. The following steps are required:





(i) *Identify overlapping tracks*: An isentropic track is selected and all isentropic tracks on all isentropes are identified that at least once overlap with the selected track (i.e. contain the same 3D PV cutoff object). This search is repeated for all overlapping tracks until no further tracks are found (see Fig. 2).

(ii) *Create non-branching 3D track from overlapping tracks*: The overlapping tracks are used to connect PV cutoff objects to a non-branching track according to the following rules: (a) The first time step a track of the overlapping tracks exists at any isentropic level marks the start of the 3D track. If there is more than one PV cutoff object at this time step that belongs to the overlapping tracks (which occurs if two tracks merge at a later time step), the dominant PV cutoff object is identified as the one with more isentropic levels or, if the number of levels is equal, the larger cutoff is selected [see

red and blue markers in the box (a) in Fig. 2]; (b) Then, for the next time step, only PV cutoff objects are considered as successors if they are connected via an isentropic track with the previous PV cutoff object (see red and blue markers in the box (b) in Fig. 2). (c) If rule (b) has been applied and still more than one PV cutoff object is part of the overlapping tracks at a later time step (which occurs if isentropic tracks are continued differently on different isentropic levels) the same criteria (depth and size) are applied as in rule (a) to select the dominant cutoff [see red and blue markers in the box

(c) in Fig. 2]. Eventually, tracks are only retained if they have a minimum lifetime of 24 h.

To focus on PV cutoffs that become dynamically relevant, i.e. potentially have a substantial effect on the static stability and wind field, they are required to extend over at least three isentropic levels at least once during their lifetime (i.e. reach an isentropic depth of at least 15 K). With this filtering, about 30% of the previously identified tracks are retained in both hemispheres. As a result, a total number of 46'353 PV cutoff tracks provide the basis for this study, which is the largest dataset of PV

cutoffs/COLs analyzed so far. On average, this amounts to about 100 PV cutoffs per month. To visualize tracks and locate genesis and lysis events the PV cutoff centre is computed as the average of the coordinates of all grid points within the PV cutoff object weighted by their PV value. The area of the PV cutoff is determined as the area of the projection of the 3D PV cutoff onto a 2D plane, i.e. it includes all grid points that are part of the PV cutoff on at least one isentropic level. Examples demonstrating the application of the tracking to individual cases are shown in the supplementary material S3.

### 2.3.3 Limitations

The tracking method has two important limitations which are briefly mentioned here. First, only non-branching tracks are allowed and the decision criteria used may not represent the most reasonable track continuation in some cases. However merging and splitting are comparably rare. Therefore, this limitation does not question the usefulness of the method and the quality of the results presented here. At the poles, where mass fluxes associated to merging and splitting are larger and

the definition of PV cutoffs becomes less obvious, this limitation is strongest. Second, the tracking procedure requires the computation of a large number of trajectories and is therefore computationally expensive. This limits its application to datasets much larger than ERA-interim or to large operational ensemble forecasts for quasi real-time purposes.





## 2.4  Computation of mass fluxes

Diabatic processes can change PV of air parcels and thereby lead to a mass transport across the tropopause. In addition, quasi-
adiabatic merging with another PV cutoff or injection of mass from the stratospheric reservoir can increase, and splitting can
decrease, the mass of a PV cutoff. With the Lagrangian tracking introduced in section 2.3, these mass changes can be quantified.
On a given isentropic level, the 6-hourly change of mass of the PV cutoff due to TST, STT, merging, and splitting equals the
number of grid points, i.e., trajectories, experiencing this process multiplied by the mass represented by the trajectories. This
basic approach to quantify cross-tropopause transport using air parcel trajectories has been introduced by Wernli and Bourqui
(2002) and applied in various studies thereafter (e.g. Sprenger et al., 2003; Bourqui, 2006; Skerlak et al., 2014). The mass of
a trajectory initiated at grid point $i$ can be calculated from the pressure thickness of the isentropic layer $\theta \pm 2.5\,\mathrm{K}$ at time $t$
according to

$$M_\mathrm{i}(\theta, t) = \frac{1}{g} \cdot A_\mathrm{i} \cdot [p_\mathrm{i}(\theta - 2.5\,\mathrm{K}, t) - p_\mathrm{i}(\theta + 2.5\,\mathrm{K}, t)], \tag{1}$$

where $A_\mathrm{i}$ denotes the area represented by grid point $i$, $p_\mathrm{i}(\theta, t)$ the time- and level-dependent pressure at grid point $i$, and $g$
the gravitational acceleration. To facilitate the computation, the mass averaged over all $N$ grid points within a PV cutoff at
isentropic level $\theta$ and time $t$ is used as representative mass for these grid points at the level $\theta$. It is computed as

$$M_\theta(t) = \frac{1}{N} \sum_{i=1}^{N} M_\mathrm{i}(\theta, t). \tag{2}$$

The mass flux due to a certain process (e.g. STT) can then be calculated for each PV cutoff at each time step of its life cycle as
total mass transport over all levels divided by the cutoff area $A(t)$

$$m_{STT}(t) = \frac{\sum_\theta n_\theta^{STT}(t) M_\theta(t)}{A(t)}, \tag{3}$$

where $n_\theta^{STT}(t)$ is the number of trajectories experiencing STT on the isentropic level $\theta$ at time $t$. The division by the area yields
a measure that is independent of the size of the PV cutoff and is therefore better comparable among different PV cutoffs and to
other studies. There are two major differences to the approach by Wernli and Bourqui (2002) and subsequent trajectory-based
quantifications of STT and TST. First, the minimum time required for an air parcel to stay at one side of the tropopause (so-
called residence time) is 6 h while in other studies mostly residence times of one or several days have been used. As shown by
Skerlak et al. (2014), the cross-tropopause mass flux decreases with increasing residence time $\tau$ according to a power law ($\tau^\kappa$,
$\kappa \approx -0.5$). Therefore, the values in this study tend to be higher than in other studies (approximately 3 times higher compared
to a 48 h residence time). Second, in order to enable a tracking of cutoffs on isentropes, this approach is based on isentropic
trajectories instead of three-dimensional kinematic trajectories used in other studies.






## 3   Climatology of PV cutoff frequencies, genesis, and lysis

This section provides a global overview of PV cutoff occurrence and favoured genesis and lysis regions. To this aim Fig. 3 shows maps of seasonal mean PV cutoff frequencies for boreal winter (DJF), spring (MAM), summer (JJA), and autumn (SON). These frequencies indicate the percentage of time steps at which a 3D PV cutoff is located at a particular grid point, in-

dependent on the number of levels covered by the 3D PV cutoff. PV cutoff frequencies are substantially higher in the Northern Hemisphere (annual mean hemispheric average of 4.3%) than in the Southern Hemisphere (2.1%) in all seasons except DJF, when frequencies are similar in both hemispheres. Figure 3 shows a seasonal cycle with hemispheric averages about two times larger in the summer season than in the winter season in both hemispheres. The number of tracks (not shown) exhibits a much weaker seasonal cycle with about 14% (27%) more cutoff tracks in JJA (SON) than in DJF in the Northern Hemisphere, and

53% more cutoff tracks in DJF than in JJA in the Southern Hemisphere, indicating that cutoffs in summer are longer-lived.

In DJF (Fig. 3a), the large northern hemispheric maximum over Southern Europe and the Mediterranean with frequencies up to 11% is conspicuous. Other frequency maxima in the Northern Hemisphere occur over Newfoundland, the southwestern US, and near the Sea of Japan. In the Southern Hemisphere, PV cutoff frequencies have a clear maximum in a large tilted band from the central subtropical South Pacific towards the southeast reaching southern South America. A similar band is also present

over the South Atlantic, with two maxima east of Brazil and west of South Africa. High frequencies occur also in the southern Indian Ocean, over New Zealand, and in a circumpolar band at 60°S around Antarctica.

In MAM (Fig. 3b). PV cutoff frequencies over Europe decrease and shift northward while they strongly increase over the North Pacific. Additional frequency maxima appear over the Canadian Arctic and central Russia. The appearance of these polar frequency maxima may also be due to generally lower altitudes of the isentropic levels in MAM compared to DJF. The

290 K lower bound used in this study may enable cutoff detection in MAM closer to the pole. The frequencies in the Southern Hemisphere decrease in most regions and the circumpolar band shifts equatorward and becomes less symmetric. The maxima east of Brazil and in the subtropical South Pacific disappear and a new maximum appears over southeastern Australia reaching far into the South Pacific.

In JJA (Fig. 3c), frequencies over the Canadian Arctic and the North Pacific further increase. Over the US and over central

and southern Europe the frequencies strongly decrease (with the exception of the Iberian Peninsula), while they increase over northern Europe and particularly strongly south of Iceland. In the subtropical North Pacific and subtropical North Atlantic large northeastward sloping bands of high frequencies appear, similar to their southern hemispheric counterparts in DJF. The Southern Hemisphere is now dominated by a zonal band between about 30°N and 60°N with local maxima over southern Australia and New Zealand, the central South Pacific, southern South America, and South Africa.

Frequencies in SON (Fig. 3d) are similar to MAM. Increasing frequencies compared to JJA are discernible over the US, and central and southern Europe. The maxima over the Arctic, in the North Pacific and the two bands in the subtropical North Pacific and North Atlantic are still visible but frequencies substantially decrease. In the Southern Hemisphere, the circumpolar band shifts poleward and becomes more symmetric again and the tilted bands over the South Pacific and South Atlantic start to reappear.





To better understand these frequency patterns, it is insightful to look at the patterns of genesis and lysis frequencies. Note again that lysis can be due to reabsorption or diabatic decay. Figures 4 and 5 show seasonal maps of the number of genesis and lysis events per season that occur within a 500 km distance of each grid point. Genesis frequencies are mostly but not always largest where also PV cutoff frequencies are largest, indicating variability in cutoff mobility, size, and lifetime in different regions. For example, the high frequency over the Mediterranean in DJF (Fig. 3a) does not seem to be linked exclusively to

frequent genesis over the Mediterranean (Fig. 4a) but also frequent movement of PV cutoffs forming over central and northern Europe into the Mediterranean, where lysis is very frequent (Fig. 5a). In the Southern Hemisphere in DJF the well-known (see section 1) genesis maxima close to the west coast of South America and west of South Africa are visible (Fig. 4a). The lysis maxima in these regions are shifted eastward over land suggesting effects of orography, land-atmosphere interaction, and continental convection on PV cutoff lysis. Generally, it seems that in summer lysis is enhanced over land (e.g. in JJA over

the Iberian Peninsula and the Rocky Mountains/Pacific Coast Ranges in Fig. 5c), whereas in winter lysis maxima often occur over sea surfaces, in particular where enhanced sea surface temperatures are expected (e.g. in the Mediterranean and in regions of all western boundary currents, especially over the western North Atlantic and east of Australlia; see Fig. 5a,c). A further conspicious lysis maximum is discernible over the central US / southern Rocky Mountains in all seasons except JJA. This also indicates effects of orography but may also be related to the supply of moist and warm air form the Gulf of Mexico.

A further interesting aspect is revealed when comparing the lifetime and distance between genesis and lysis in the two hemispheres. Table 1 shows that in both hemispheres, more than 50% of all PV cutoff tracks persist for less than three days and 12-18% for seven days or longer, with a tendency of southern hemispheric PV cutoffs to last shorter. However, comparing the distances between genesis and lysis (Table 2) shows that PV cutoffs tend to travel further in the Southern Hemisphere than in the Northern Hemisphere: 62% travel more than 3000 km and only 9% less than 1000 km in the Southern Hemisphere, versus

48% and 15% in the Northern Hemisphere, respectively. This might be related to the presence of a strong circumpolar jet stream that allows rapid advection of PV cutoffs once they have formed.

## 4 The nature of PV cutoff life cycles

The tracking procedure introduced in Sect. 2.3 allows for a comprehensive analysis of PV cutoff life cycles. In this section, subsets of PV cutoff tracks are investigated, selected according to their genesis region. In DJF and JJA, six genesis regions are

selected each such that a broad spectrum of regions is covered (see blue boxes in Fig. 4a,c, for more details see Table S1 in the supplementary material). In the following, various aspects of the life cycles of PV cutoffs with genesis in these twelve regions are investigated. We start with a discussion of the synoptic configurations of PV cutoff genesis. Subsequently, PV cutoff tracks and their vertical evolution and lifetimes are discussed. The vertical evolution is then linked to the occurrence of diabatic decay and reabsorption, and the evolution of stratosphere-troposphere exchange along the life cycles. To put these aspects into the

context of the climatological analysis presented in Sect. 3, their geographical distribution is also discussed. As a first step towards understanding surface impacts of PV cutoffs, the last part of this section investigates the link to surface cyclones, in particular focusing on the chronology of the life cycles of PV cutoffs and surface cyclones.



## 4.1 Synoptic configuration of PV cutoff genesis

It is well known that the genesis of PV cutoffs is often the result of anticyclonic RWB (e.g., Thorncroft et al., 1993; Ndarana
and Waugh, 2010), which occurs on the anticyclonic shear side, i.e., equatorward, of the jet stream and is characterized by an
equatorward extending and backward tilted PV streamer (Martius et al., 2007) that then breaks up into one or more PV cutoffs
(Appenzeller et al., 1996). However, already Thorncroft et al. (1993) pointed out that also cyclonic RWB may result in cutoff
formation. Cyclonic RWB is known to predominantly occur on the cyclonic shear side, i.e., poleward, of the jet stream and is
characterized by the cyclonic wrap up of a PV streamer (Martius et al., 2007) around the surface cyclone, which is positioned
poleward of the PV streamer and classically results in an occluded surface cyclone (Thorncroft et al., 1993; Wernli et al., 1998).
Given the high genesis frequencies in the storm track regions (Fig. 4), PV cutoffs there are expected to form due to cyclonic
RWB.

To check whether the genesis of PV cutoffs in the selected regions follows these archetypes, Fig. 6 shows composites of 250 hPa
geopotential height anomalies and wind speed, mean sea level pressure, and frequencies of surface cyclones and PV streamers
for all genesis events in the six PV cutoff genesis regions selected in DJF. In all regions, geopotential height anomalies show
a wave-like pattern with a negative anomaly within and a positive anomaly upstream and/or poleward of the region of PV
cutoff genesis. Also, all regions show high frequencies of stratospheric PV streamers (yellow contours) in the vicinity of the
PV cutoff formation. This illustrates again the three-dimensional aspect of PV cutoff formation: A PV cutoff initially forms on
lower isentropic levels while there is still a PV streamer on higher isentropic levels that may break up at later times. Beside
these consistencies, there are substantial regional differences. PV cutoffs with genesis over the eastern South Pacific and South
Africa show clear signatures of anticyclonic wave breaking with PV cutoff genesis on the anticyclonic shear side of the jet
stream and at the eastern flank of a subtropical surface anticyclone and downstream and equatorward of a surface cyclone in
the storm track region (Fig. 6d,e). In the western Mediterranean, PV cutoff genesis occurs on the anticyclonic shear side of the
polar jet over the North Atlantic and on the cyclonic shear side of the subtropical jet over northern Africa (Fig. 6c). The tilt of
the region with high PV streamer frequencies and enhanced surface cyclone frequencies between Greenland and Iceland are
also indicative of anticyclonic RWB over Europe. Together with the enhanced cyclone frequencies in the genesis region this
indicates that first, wave breaking occurs anticyclonically but as the equatorward extending PV streamer is influenced by the
cyclonic shear of the subtropical jet, it can break cyclonically to form a PV cutoff located between the polar and the subtropical
jet streams. A similar but less clear scenario occurs also over California (Fig. 6a). The synoptic pattern in the storm track
regions (western North Atlantic and southern Indian Ocean close to Antarctica) strongly resembles cyclonic wave breaking
resulting in a PV cutoff poleward of the jet stream (Fig. 6b,f). Enhanced cyclone frequencies just below and slightly poleward
of the stratospheric PV streamer, on the cyclonic shear side of the jet stream are clear indicators of this. These results and
interpretations are consistent with the maxima of anticyclonic (cyclonic) wave breaking over the southwestern US and central
Europe (the North Atlantic) found by Martius et al. (2007).

Figure 7 shows similar results for JJA, although with substantially weaker geopotential height anomalies and lower wind speeds
in the Northern Hemisphere. PV cutoffs over the central and eastern subtropical North Atlantic form from anticyclonic wave





breaking equatorward of the jet stream (Fig. 7b,c) and PV cutoffs in the Hudson Bay show signatures of cyclonic wave break-
ing (Fig. 7a). Over Australia and the Baltic Sea, the synoptic patterns are similar to the one in the Mediterranean, with first
anticyclonic, and subsequent cyclonic wave breaking (Fig. 7d,f). However, the combined scenario over Australia, in contrast
to the Mediterranean, does not show many surface cyclones in the genesis area. Over the Sea of Okhotsk the scenario is less
clear (Fig. 7e).

It has to be kept in mind that behind the composite structures there can be a high variability in the wave breaking between
individual cases. In particular, the PV streamer may not be tilted but rather meridionally aligned, or that a PV streamer is not
present at all, as it seems to occur for many cases over the Hudson Bay. Nevertheless, the results of this section provide clear
evidence for distinct wave breaking scenarios leading to PV cutoffs in different regions.

## 4.2 PV cutoff tracks

Next, an overview of the tracks is given for all selected genesis regions. Figure 8 shows the tracks and lysis points for the
six regions selected in DJF. For all regions, PV cutoffs tend to travel eastward. Most PV cutoffs forming over California
move across the Rocky Mountains, where some tracks already end (Fig. 8a), consistent with the lysis maximum there in Fig.
5a. However, many continue to move northeastward over the southern and eastern US, and some even travel far into the North
Atlantic. PV cutoffs forming over the western North Atlantic mostly remain over the North Atlantic, some travel into the Arctic
Ocean, and a few over Europe and into the Mediterranean (Fig. 8b). PV cutoffs with genesis over the western Mediterranean
first travel southeastward across the Mediterranean and then tend to move on a more eastward or northeastward path over
eastern Europe and the Middle East (Fig. 8c). Very few end their life cycle in the genesis box. The tracks with genesis over
the eastern South Pacific and over South Africa are strikingly similar (Fig. 8d,e): they lead southeastward across the southern
tips of South America (South Africa) where many tracks end over land and some reach further into the South Atlantic (South
Pacific). PV cutoffs forming close to Antarctica travel far eastward and a few almost fully around Antarctica (Fig. 8f).

In JJA, PV cutoffs also mostly travel eastward in all selected regions except the ones forming over the central subtropical North
Atlantic (Fig. 9b). There, they tend to move along a northeastward tilted band in either direction or remain stationary (many
end their life cycle in the genesis box). Together with PV cutoffs forming over the eastern subtropical North Atlantic (Fig. 9c),
they contribute to the lysis maximum over the Iberian Peninsula (Fig. 5c). PV cutoffs with genesis over the Hudson Bay (Fig.
9a) and the Sea of Okhotsk (Fig. 9e), i.e. at the beginning of the North Atlantic and North Pacific storm tracks, move eastward
and contribute to the high lysis frequencies in the centre and at the end of the storm tracks (see Fig. 5c). PV cutoffs forming
over the Baltic Sea are relatively stationary and mostly end their life cycle in northeastern Europe or western Russia (Fig. 9d).
Finally, cutoffs with genesis over Australia are very mobile with many of them moving far into the South Pacific (Fig. 9f).

## 4.3 Vertical evolution and half lifes

Further differences and similarities of the life cycles of PV cutoffs forming in the selected regions become apparent when
looking at their vertical evolution. Figure 10 shows, for each of the six genesis regions selected in DJF, the frequency of PV



cutoffs on all considered isentropic levels during their life cycle. First, it becomes obvious that the isentrope with the highest frequency and the vertical range of isentropes with PV cutoffs vary substantially between regions. For example, PV cutoffs over California can be found at levels from 290 K up to 345 K (Fig. 10a), whereas over the Mediterranean they are restricted to levels below 330 K (Fig. 10c), even if the level with the highest frequency is similar (310-315 K in both regions). Also PV cutoffs close to Antarctica occur preferably between 295-315 K (Fig. 10f), whereas PV cutoffs forming over the eastern South

Pacific and South Africa occur from 310 K up to 350 K (Figs. 10d,e). Figure 10 also shows that the frequencies are highest about one day after genesis and for some regions the frequency maximum rises to higher levels during the days after genesis (e.g. California, western North Atlantic, the Mediterranean, see Fig. 10a-c). A main reason for these two properties is that PV cutoffs form on a lower isentropic level first and as the break-up of the PV streamer continues, additional vertical levels become part of the PV cutoff until it reaches its full vertical extent. Interestingly, the timescale of roughly one day until reaching

the maximum frequency is consistent across all regions, indicating that it can serve as a global estimate of the average time required for the complete break-up of a PV streamer into PV cutoffs. After one day, the cutoff frequency gradually decreases and the number of active cutoff tracks follows relatively closely an exponential decay. This indicates that the probability of a PV cutoff track to end is relatively constant, i.e. the "death" of PV cutoffs can be regarded as an exponential decay process. We use an exponential curve fitted to the number of active tracks after genesis to estimate the half life $\tau_{1/2}$ of the PV cutoffs

once they have overcome the 24 h minimum duration required in this study. This value ranges between 38 h (South Africa) and 64 h (Mediterranean). Hence, the median lifetime is between 52 and 88 h and the expected (i.e. mean) lifetime $\overline{T}$ [computed as $\overline{T} = 24\,h + \tau_{1/2} \cdot \ln(2)^{-1}$] between 79 and 116 h. This reveals that the lifetime of PV cutoffs is highly variable between regions. A similar picture appears for JJA (Fig. 11), where maximum frequencies are also found roughly one day after genesis in all regions and they decay approximately exponentially afterwards. Interestingly, the isentrope of maximum frequency gradually

decreases during the life cycle of the PV cutoffs over the central subtropical North Atlantic (Fig. 11b), indicating a downward growth, while over Australia it rises rapidly by more than 5 K within the first two days (Fig. 11f), indicating rapid decay at lower levels. Striking is also the very large half life (114 h, $\overline{T} = 188$ h) of the PV cutoffs forming over the central subtropical North Atlantic (Fig. 11b), which is more than two times longer than for the neighboring region over the eastern subtropical North Atlantic (Fig. 11c). PV cutoffs forming over the Sea of Okhotsk and the Hudson Bay are also relatively long lived with

half lifes of 90 h and 69 h, respectively (Fig. 11a,e), while the ones over the Baltic Sea and Australia have shorter half lifes of 58 h and 51 h.

## 4.4 Diabatic decay and reabsorption

The vertical evolution of the 3D PV cutoffs discussed in the previous section is determined by the appearance and disappearance of 2D PV cutoffs on isentropic levels. A PV cutoff appears on an isentropic level either during the break-up process or as

it grows downward, involving TST. While these processes are not directly quantified in this study, the processes leading to disappearance, i.e. diabatic decay and reabsorption, are. Whenever a PV cutoff disappears on an isentropic surface, it is quantified what fraction of the PV cutoff experiences diabatic decay (i.e. undergoes STT) and what fraction is reabsorbed by the stratospheric reservoir (see section 2.3 and Fig. 1e,f). The diabatic decay fraction can vary between 0% (pure reabsorption)





and 100% (complete diabatic decay). Intermediate values indicate simultaneous partial reabsorption and partial decay. Note
that this analysis identifies events on isentropic surfaces, which means that, even if a 2D PV cutoff disappears, the 3D PV cutoff
may still persist afterwards. Therefore, reabsorption and decay do not necessarily occur only during PV cutoff lysis. Here, we
first investigate the global statistics of decay and reabsorption, then use the quantification to provide further insight into the
vertical characteristics of the climatological life cycles, and finally provide a global overview of the geographical distribution
of these two possible scenarios.

Figure 12a shows the decay and reabsorption statistics for five different categories of the decay fraction for all 2D PV cutoffs
identified globally. It shows that almost pure reabsorption (decay fraction < 25%) is equally frequent as complete diabatic
decay (decay fraction = 100%), while intermediate scenarios are relatively rare. When considering all events with a decay
fraction less than 50% as reabsorption and all other events as diabatic decay, reabsorbtion accounts for almost half (47%)
of the disappearances of 2D PV cutoffs. During lysis of 3D PV cutoffs, reabsorption is with a share of 54% even a little
more frequent than diabatic decay. This result disagrees with the expectation by Hoskins et al. (1985) that diabatic decay
dominates. Figure 12b,c provides the following explanation for this result: Reabsorption predominantely occurs for large PV
cutoffs with comparatively high PV values. For an individual 3D PV cutoff this is usually the case on higher isentropic levels,
where it is closer to the stratospheric reservoir. Hence, at higher levels, chances are high that the PV cutoff is reabsorbed by
the stratospheric reservoir. It may also occur that the reabsorption is transient, i.e. a PV cutoff is reabsorbed and later again
detached from the stratospheric reservoir several times during its life cycle. This behavior has been shown by Portmann et al.
(2018) for two case studies over Europe. Complete diabatic decay, on the other hand, occurs typically at lower levels where the
PV cutoff is smaller and has lower PV values.

This aspect is now discussed further for the selected genesis regions. Again, all disappearance events of 2D PV cutoffs with
a decay fraction less than 50% are considered as reabsorption and all other events as diabatic decay. The vertical distribution
and temporal evolution of reabsorption and diabatic decay events for the genesis regions in DJF is shown in Fig. 13. Indeed, in
all regions the diabatic decay maxima occur at lower levels than the reabsorption maxima. Hence, the reduction of PV cutoff
frequencies with lifetime (see Fig. 10) is, at lower levels, mainly due to diabatic decay, and at higher levels due to reabsorption.
Beside these common aspects, the temporal evolution and relative frequencies of diabatic decay and reabsorption substantially
differ between regions. For example, over California, the Mediterranean and the western North Atlantic (Fig. 13a,b,c), PV
cutoffs experience frequently diabatic decay already immediately after genesis, while over the eastern South Pacific and South
Africa (Fig.13d,e), this is the case only about one day later. PV cutoffs with genesis over Antarctica experience only rarely
diabatic decay, which explains the rather adiabatic evolution without vertical displacement of the maximum frequency. Lysis
of the 3D PV cutoff occurs as diabatic decay with a frequency of around 60% for PV cutoffs forming over the eastern South
Pacific and South Africa, and a bit less frequently for PV cutoffs with genesis over California (55%) and the Mediterranean
(47%; indicated by the numbers on the top right in each panel). For PV cutoffs forming in the storm track regions (western
North Atlantic, Antarctica), this value is substantially lower, i.e. a majority of the PV cutoffs there end their life cycle with
reabsorption.

In JJA (Fig. 14) the general pattern with decay at lower levels and reabsorption at higher levels is similar to DJF. Unexpected





are the frequent decay events for PV cutoffs over the central subtropical North Atlantic (Fig. 14b), as at the same time the
frequencies at these levels increase over time. This indicates that the many decay events must be overcompensated by the
downward extension (i.e. many appearances at lower levels), such that on average these PV cutoffs grow downward. In all
other regions, the maxima of diabatic decay (reabsorption) are consistent with a reduction of PV cutoff frequencies at lower
(higher) levels. Particularly frequent decay is observed for PV cutoffs over Australia (Fig. 14f). In most regions, diabatic
decay is less frequent during lysis than reabsorption, consistent with the global average. The highest ratios of diabatic decay
events during lysis (between 40 and 50%) are found for the central and eastern subtropical North Atlantic, the Baltic Sea, and
Australia. The lowest ratios are found for the Hudson Bay and the Sea of Okhotsk, i.e. in these regions lysis mainly occurs
as reabsorption. In fact, in both regions and especially over the Sea of Okhotsk, reabsorption dominates during the whole life
cycle.

Returning again to all PV cutoffs in DJF and JJA, the geographical distributions of reabsorption and diabatic decay events of
2D PV cutoffs are visualized (Fig. 15). The maps show the seasonal mean frequency of reabsorption (Fig. 15a,c) or decay
(Fig. 15b,d) occurrence within a 500 km distance of a particular grid point in DJF and JJA. The geographical patterns of both
categories resemble the PV cutoff frequencies presented in Fig. 3. This reveals that they can both occur during all phases of
the life cycle. However, frequencies are particularly high where also lysis frequencies are high (Fig. 5a,c). Lysis maxima at
higher (lower) latitudes are preferentially near reabsorption (decay) maxima, showing that the prevailing lysis mechanism,
similar to the genesis mechanism, is dependent on the latitude, i.e. region. During DJF, in particular the lysis maxima over
the central/southern US, the US east coast, the Mediterranean, southern South America, South Africa, and south of Australia
are dominated by diabatic decay and the ones over Alaska, south of Greenland, northeastern Russia, and the southern Ocean
around Antarctica by reabsorption. In JJA, diabatic decay dominates lysis maxima over the western US, the Iberian Peninsula,
South Africa and south of Australia. Reabsorption more strongly contributes to lysis maxima over the northern North Pacific,
the Hudson Bay, south of Iceland, and again the southern Ocean around Antarctica. Some conspicuous decay and reabsorption
maxima occur that cannot directly be related to lysis maxima. They are mainly located in the tilted subtropical bands in summer
(consistent with Fig. 14b). This is likely related to the longevity and quasi-stationarity in combination with the diabatically
active lower levels of PV cutoffs in this region. These PV cutoffs may experience frequent transient reabsorption events and it
may also happen frequently that a PV cutoff decays and reappears again at a lower isentropic level (i.e transient decay events,
see also supplementary material S3, Figure S1).

## 4.5 Stratosphere-troposphere exchange

Diabatic modification of PV cutoffs (i.e. TST and STT) does not only occur when they appear or disappear on an isentropic
surface but potentially during the whole life cycle. In this section, we first investigate how STT and TST mass fluxes evolve
during the life cycles of the PV cutoffs in the selected genesis regions and, second, discuss the geographical distribution of
net cross-tropopause mass fluxes. The mass fluxes due to merging and splitting are comparably small and are therefore not
considered further. As PV cutoff lifetimes are very variable, a more meaningful comparison of STT and TST of individual
tracks is achieved by normalizing the tracks between the fix points genesis and lysis (for further details see supplementary





material S1). STT and TST along normalized life cycles are shown in Figs. 16 and 17. For all regions in DJF (Fig. 16), TST is highest during genesis and gradually decreases towards the end of the life cycle. For PV cutoffs with genesis in California and

the Mediterranean, STT is generally larger than TST and shows a u-shaped evolution, with high STT values around genesis and lysis (Fig. 16a,c). This results in particularly large negative net mass fluxes (i.e. the difference between TST and STT) towards the end of the life cycles, indicating that the PV cutoff shrinks due to diabatic processes (subsequently referred to as 'diabatic erosion' ). Increasingly large negative net mass fluxes during the life cycle also occur for cutoffs over the eastern South Pacific and South Africa (Fig. 16d,e). There, TST gradually decreases and STT increases (eastern South Pacific) or remains fairly

constant (South Africa), rather than showing a u-shaped evolution. As a result, net mass fluxes are small at the beginning of the life cycle. A strikingly different behavior occurs for PV cutoffs in the storm track regions (western North Atlantic and Antarctica, Fig. 16b,f), where both, STT and TST, decrease and net fluxes remain fairly constant during the life cycle, with STT slightly dominating over TST over the western North Atlantic and vice versa over Antarctica. The dominance of TST over STT over Antarctica is striking because it indicates that PV cutoffs there tend to grow rather than decay. The evolutions of STT

and TST in good agreement with the numbers of reabsorption and decay events during the normalized life cycle (numbers at the bottom of each panel in Fig. 16). When STT is much larger than TST, diabatic decay is more frequent than reabsorption, for example towards the end of the life cycle of PV cutoffs over the eastern South Pacific (Fig. 16d). Vice versa, when TST is larger than STT, reabsorption is more frequent than decay, as for the whole life cycle of PV cutoffs over Antarctica. In all regions, the highest frequencies of both reabsorption and decay occur during the last part of the life cycle, consistent with the

geographical agreement of maxima of 2D PV cutoff decay/reabsorption events and lysis maxima of 3D PV cutoffs (Figs. 5a,c and 15).

In JJA (Fig. 17), similar STT and TST evolutions can be observed. PV cutoffs with genesis over Australia and the Baltic Sea show a u-shaped STT evolution with STT dominating over TST (Fig. 17d,f), similar to California and the Mediterranean in DJF. A u-shaped evolution of STT also occurs over the Sea of Okhotsk, but net mass fluxes remain comparably small and

in the middle of the life cycle TST even tends to dominate over STT (Fig. 17e). Net mass fluxes also remain fairly low and STT and TST fairly constant for PV cutoffs forming over the Hudson Bay (Fig. 17a). Sea of Okhotsk and Hudson Bay show an evolution most similar to Antarctica and western North Atlantic in in DJF. An interesting evolution occurs for PV cutoffs with genesis in the central and eastern subtropical North Atlantic (Fig. 17b,c): During the first about three quarters of the life cycle, TST dominates over STT. Afterwards, STT starts to increase and finally becomes larger than TST. This shows that PV

cutoffs in these regions first grow diabatically and only late in their life cycle diabatic erosion dominates. Again, reabsorption tends to be more frequent than decay for regions where STT is roughly equal or smaller than TST, for example for the Hudson Bay and the Sea of Okhotsk (Fig. 17a,e). Diabatic decay dominates for regions where STT is larger than TST, for example during the first four fifths of the life cycle for PV cutoffs over Australia (Fig. 17f). Interestingly, during the last fifth of the life cycle for this region, reabsorption dominates even if STT is much larger than TST. This indicates that, most likely, these PV

cutoffs approach the stratospheric reservoir, rendering reabsorption more likely to occur at higher isentropic levels, while they experience substantial STT and diabatic decay at lower levels. The case study in the supplementary material S3 (Figure S2) confirms this behavior. A similar evolution occurs over the Baltic Sea and over the Mediterranean in DJF (Fig. 16c).





Considering all PV cutoffs during DJF and JJA, the geographical distribution of average net cross-tropopause mass fluxes (TST minus STT) during the presence of a PV cutoff is visualized for both seasons in Fig. 18. Distinct regions with enhanced
negative cross-isentropic mass fluxes appear. In DJF (Fig. 18a) these are the western parts of the North Atlantic and the subtropical North Pacific, the lee of the southern Rocky Mountains, the Middle East, southern South America and western South Atlantic, South Africa and south-western Indian Ocean, and southern Australia and the western South Pacific. In JJA (Fig. 18b) large negative mass fluxes occur mainly in the vicinity of high orography in the Northern Hemisphere (the northern Rocky Mountains, the Alps, the Caucasus, Central Asia) and around the western and subtropical parts of the ocean basins in
the Southern Hemisphere, with particular hot spots over coastal southern Brasil, western Australia, and the subtropical South Pacific. Clear regional patterns also appear for enhanced positive net mass fluxes. They mainly occur over the subtropical ocean basins in the summer season and in polar regions. This striking change of the sign of the average net mass fluxes related to PV cutoffs between regions also agrees well with the life cycle perspective for the selected genesis regions discussed above. There are preferred regions where PV cutoffs are predominantely eroded diabatically and where they remain adiabatic or even grow
diabatically. In conclusion, the probability of a PV cutoff to experience diabatic erosion is dependent more on where it is than how old it is. The geographical regions with strongly negative net mass fluxes qualitatively agree well with certain diabatic decay and lysis maxima.

## 4.6  Link to surface cyclones

In this final results section, a step is made towards understanding the surface impacts of PV cutoffs during their life cycle.
While it is well known that PV cutoffs can be associated to surface cyclones in some regions, the linkage between PV cutoff life cycles and cyclone life cycles and how it varies between regions has not been investigated systematically. Therefore, the main question addressed in this section is: When during their life cycle are PV cutoffs linked to surface cyclones and how old are these cyclones?

An insightful way to address this question is to count all PV cutoffs that are at lifetime $t_{cutoff}$ linked to a surface cyclone
at lifetime $t_{cyclone}$ and show these counts in a $t_{cutoff}$ vs. $t_{cyclone}$ diagram. A link is identified where the centre of a surface cyclone is within a distance of 600 km around the PV cutoff centre. Note that both, PV cutoffs and surface cyclones, are required to have a minimum lifetime of one day in this study. Figure 19 shows such a diagram for the genesis regions in DJF. For example, the green/blue boxes in the leftmost column in Fig. 19b show that, at their genesis, PV cutoffs over the western North Atlantic are preferentially linked to surface cyclones that are about 1-3 days old. The diagonal displacement of these
green/blue boxes for higher PV cutoff lifetimes indicate that the PV cutoffs stay linked to the surface cyclones for a few days, i.e. they remain linked and propagate and become older together. The transition to beige/grey colors along this diagonal shows that the number of these initial links decreases with time. A link disappears either because the PV cutoff decays, the surface cyclones decays, or the surface cyclone and the PV cutoff move away from each other. The green/blue boxes in the lowermost row in Fig. 19b indicate that many PV cutoffs with genesis in the western North Atlantic are linked to surface cyclogenesis after
a cutoff lifetime of 0.5-2 days. These links remain for a few days and then gradually disappear. These results show that most PV cutoffs with genesis over the western North Atlantic tend to be either linked to an 'old' surface cyclone at their genesis or





be involved in cyclogenesis 1-2 days later. After about 2 days, almost 160 out of 206 PV cutoffs are linked to a surface cyclone (red curve) with the age of the surface cyclone ranging from 0 to more than 4 days. In total 183 PV cutoffs in this region are linked to a surface cyclone at least once during their life cycle (number at top right in Fig. 19b).

Figure 19 shows that each region has its characteristic pattern. Half of the PV cutoffs forming over California (Fig. 19a) are linked to a surface cyclone resulting in two peaks, one after about 1 day and one after 3 days: The first peak is the result of cyclogenesis ±1 day around PV cutoff genesis and the second one involves cyclogenesis after 2-3 days lifetime. About 70% of PV cutoffs with genesis over the Mediterranean (Fig. 19c) are linked to surface cyclones, often with cyclogenesis around 1 day prior to PV cutoff genesis. Some PV cutoffs are also involved in cyclogenesis after 2-3 days lifetime. Only roughly a third of

the PV cutoffs forming over the eastern South Pacific are linked to surface cyclones during their life cycle (Fig. 19d). The link preferentially occurs during the first 2 days of the PV cutoff life cycle and at or shortly after cyclogenesis. Over South Africa, almost 60% of the PV cutoffs are linked to surface cyclones. Mostly, the link first occurs at or shortly after cyclogenesis and after about 1 day cutoff lifetime. A rapid drop in the number of linkages occurs after about 2 days cutoff lifetime, which may be partially related to enhanced PV cutoff lysis rates around that time (as shown by the rapid drop of the green curve in Fig.

10e). PV cutoffs over Antarctica show a similar pattern as the ones over the western North Atlantic, although with a less clear second surface cyclone peak. Most often, PV cutoffs in this region are already initially linked to surface cyclones that formed roughly 1-2 days prior to PV cutoff genesis.

Also in JJA, the different regions show distinct patterns (Fig. 20). Many PV cutoff genesis events over Australia occur almost simultaneously with surface cyclogenesis (Fig. 20f). This pattern is reminiscent of PV cutoffs over California and partially the

Mediterranean, and South Africa in DJF. PV cutoffs over the central (eastern) subtropical North Atlantic (Fig. 20b,c) are less frequently linked to surface cyclones. In the eastern subtropical North Atlantic, this linkage consists mainly of cyclogenesis after 1-3 days life time, reminiscent of South Africa in DJF. Over the Hudson Bay (Fig. 20a), PV cutoffs follow a similar pattern as over the western North Atlantic in DJF with many PV cutoffs that are early in their life cycle linked to surface cyclones with cyclogenesis a few days prior to PV cutoff genesis. Also in this region, PV cutoffs can be additionally involved in surface

cyclogenesis after 1-3 days of their lifetime. Over the Baltic Sea and the Sea of Okhotsk, the patterns are less clear and more or less any kind of linkage to surface cyclones can occur.

## 5 Discussion and Conclusions

In this study a novel approach to identify and track PV cutoffs as three-dimensional objects was introduced and applied to ERA-interim reanalyses for the years 1979-2017. This climatology of PV cutoffs is the first that is global and independent of the

selection of a vertical level. As the tracking is based on isentropic trajectories, it further allows quantifying different processes during the PV cutoff life cycle (stratosphere-troposphere exchange, diabatic decay, reabsorption). The resulting climatological dataset was used to study the life cycle of PV cutoffs in detail. In the following the main findings are summarized and compared to results from previous studies, starting with climatological frequencies and followed by the various investigated aspects of the life cycle. Based on these discussions, three archetypes of PV cutoff life cycles are proposed. Finally, a brief outlook for further





research is given. Note again that, in this study, short lived (lifetime below 24 h) and shallow (isentropic thickness below 15 K during the whole life cycle) have not been considered.

## 5.1    Climatological frequencies and seasonal cycle

Many of the presented frequency, genesis and lysis maxima of PV cutoffs are consistent with previous climatological studies.
In the Southern Hemisphere, many frequency and genesis maxima agree well with e.g. Fuenzalida et al. (2005), Reboita et al. (2010), and Pinheiro et al. (2017). In the Northern Hemisphere, they compare favorably with Nieto et al. (2005) mainly in subtropical latitudes, with e.g. Bell and Bosart (1989) in higher latitudes, and with Munoz et al. (2020) in most regions if COLs at 500 hPa and 200 hPa are considered. However, in this study all of them are identified based on a consistent methodology and independent of the selection of a vertical level. Interestingly, the high PV cutoff frequencies reaching from California
north-eastward across the US into the North Atlantic, in combination with the preferred movement of the California cutoffs along that band, agree very well with the frequencies and the movement of upper tropospheric storm track features identified by Hoskins and Hodges (2002). A similar agreement occurs also for the enhanced frequencies over the North Pacific in DJF. Also, some of the identified maxima have not or only little been described in the literature before. In particular, the circumpolar band around Antarctica and the far equatorward reaching band in the South Pacific in DJF have not been documented as regions of
COL occurrence yet. The circumpolar band around Antarctica and its seasonal change in symmetry agrees very well with the climatology of upper tropospheric storm track features discussed in Hoskins and Hodges (2005). The maximum east of Brazil was mentioned only by Reboita et al. (2010). Our global analysis reveals also that the summer hemispheres have two preferred latitudes of PV cutoff occurrence. One in the subtropics mostly in east- and poleward tilted bands over the Pacific and Atlantic, where frequent anticyclonic RWB is known to occur (e.g. Bowley et al., 2019) and another one at higher latitudes, in particular
in the storm track regions.

In fact, some PV cutoff frequency maxima are in striking agreement with high cyclone frequencies. For example, the frequency maxima over the Canadian Arctic and south of Iceland in JJA agree particularly well with the surface cyclone maxima found by Wernli and Schwierz (2006). Frequency maxima over the southern Indian Ocean close to Antarctica in DJF also correspond well with surface cyclone maxima identified by Jones and Simmonds (1993). Furthermore, a careful comparison to the clima-
tology of jet streams (Koch et al., 2006) reveals that high PV cutoff frequencies occur often slightly poleward of jet frequency maxima (western North Atlantic, Japan, around Antarctica in DJF) and sometimes poleward of the subtropical jet maxima and equatorward of the polar jet maxima (Mediterranean and central Europe in DJF, Australia in JJA). But PV cutoffs are also frequent in regions where strong jet streams are absent, in particular over the subtropical ocean basins in summer.

A particularly interesting geographical pattern was also found for PV cutoff lysis frequencies. In summer, lysis frequencies are
enhanced over the eastern parts of the land masses, while in winter they are largest over the western parts of the land masses and the eastern parts of the ocean basins. This is also in agreement with the geographical distribution of enhanced negative cross-tropopause mass fluxes. COL lysis rates as identified by Pinheiro et al. (2017) show a similar qualitative seasonal pattern. This indicates that in summer, PV cutoffs decay mainly due to orographic effects (e.g. friction, orographic convection)





and continental convection, and in winter enhanced diabatic activity related to warm ocean surfaces is more important. The

modelling study of Garreaud and Fuenzalida (2007) supports this hypothesis: They showed that continental convection related

to orography was crucial for the decay of a COL over the Andes in March 2005.

Several previous studies found a clear seasonal cycle with about four times more COLs forming in summer than in winter for

both hemispheres but also that it depends strongly on the considered pressure level (Nieto et al., 2005; Reboita et al., 2010;

Munoz et al., 2020). The seasonal cycle in this study is much weaker, in agreement with the findings of Wernli and Sprenger

(2007), if they took into account all isentropic levels from 305-370 K. Hence, its seems that the strong seasonal cycle found in

previous studies is mainly related to the seasonal cycle of the vertical (pressure or isentropic) level on which they occur, rather

than a seasonal cycle in the frequency of PV cutoff / COL formation [for illustration of this aspect, see Wernli and Sprenger

(2007), their Figure 8b].

## 5.2   PV cutoff life cycles

Subsets of PV cutoff tracks were selected based on twelve different genesis regions and their life cycle was studied in detail.

It was found that a characteristic difference between PV cutoffs in different genesis regions is the genesis mechanism: PV

cutoff genesis occurs either as anticyclonic RWB equatorward of the jet stream, anticyclonic RWB followed by cyclonic RWB

between the polar and the subtropical jets, or cyclonic RWB poleward of the jet stream. The life cycle analysis provided novel

insight into the mobility, life times, and vertical extent of PV cutoffs. First, PV cutoffs can be very mobile in many regions.

Hence, the picture that PV cutoffs are mainly quasi-stationary systems is misleading. In fact, many PV cutoffs travel several

thousand kilometers. This result stands in contrast to studies pointing out the quasi-stationarity of COLs (e.g. Bell and Bosart,

1989; Pinheiro et al., 2017) or even assume it to justify assumptions for the tracking procedure (e.g. Munoz et al., 2020). Then,

median life times can be highly variable between regions and range from two days over South Africa to six days over the

central subtropical North Atlantic. The frequent occurrence of PV cutoffs with long life times (larger than three days) is in

agreement with Pinheiro et al. (2017), who found mean life times of 6-8 days in large parts of the Southern Hemisphere, but

is in contrast to other studies, that found that a vast majority of COLs have life times shorter than three days (e.g. Kentarchos

and Davies, 1998; Nieto et al., 2005; Reboita et al., 2010). These differences in mobility and life time most likely arise from

the different identification and tracking methods. The 3D approach used in this study is expected to result in longer lifetimes

and larger travel distances than approaches based on single levels.

We highlighted the relevance of the vertical dimension when investigating PV cutoffs (see Figs. 10 and 11). PV cutoffs can

extend over many isentropic levels and in different regions show different climatological vertical 'footprints' during their life

cycle. While some tend to rise to higher levels, others tend to sink to lower levels. Remarkably consistent across regions is

the average duration of about one day until the climatological maximum vertical extent is reached, i.e. until the full three-

dimensional break-up process is completed after PV cutoff genesis.

These vertical footprints of PV cutoffs are modulated by the appearence and disappearance of PV cutoffs on single isentropic

levels. Disappearence on an isentropic level occurs via diabatic decay or reabsorption. For the first time, it was quantified how



frequently PV cutoffs decay diabatically or are reabsorbed by the stratospheric reservoir when they disappear at an isentropic level. Results showed that reabsorption is as frequent as decay and occurs on higher isentropic levels for 2D PV cutoffs with

higher PV and a larger area. Diabatic decay is the main reason why in many regions PV cutoffs tend to disappear with time at lower levels. Interestingly, the relative frequency of diabatic decay and reabsorption varies greatly between regions, resulting in distinct vertical evolutions and different life times. For example, during the life cycle of PV cutoffs with genesis over the eastern South Pacific diabatic decay is 60% more frequent than reabsorption, while over Antarctica reabsorption is almost twice as frequent as diabatic decay. Decay and reabsorption both occur preferentially towards the end of a (3D) PV cutoff life cycle.

The geographical distribution showed that lysis maxima in higher (lower) latitudes are generally dominated by reabsorption (decay). Decay and reabsorption are also particularly frequent in the subtropical bands with high PV cutoff frequencies in the summer (especially over the North Atlantic and North Pacific) where lysis is not particularly frequent. This is most likely the result of frequent transient decay/reabsorption events associated to the long-lived and quasi-stationary PV cutoffs in this region.

While diabatic decay is already an indication of high STT activity, in this study, STT and TST were specifically quantified during the whole life cycle of the PV cutoffs. It was found that STT dominates over TST for PV cutoffs in the global average and for many investigated genesis regions, which is well in agreement with previous studies (e.g Sprenger et al., 2007; Wu and Lü, 2016). Despite the various differences in the methodologies, the magnitude of STT and TST mass fluxes found in this study is comparable to the magnitudes found by other studies investigating similar systems, when the different residence times are

taken into account for the Lagrangian-based diagnostics. For example, Bourqui (2006) found values of up to $10000\,\mathrm{kg\,km^{-2}s^{-1}}$ for STT and up to $3000\,\mathrm{kg\,km^{-2}s^{-1}}$ for TST within a PV cutoff over the Mediterranean using a residence time of $12\,\mathrm{h}$. For intense North Atlantic cyclones in DJF, Reutter et al. (2015) found average STT (TST) values between $500\text{-}1000\,\mathrm{kg\,km^{-2}s^{-1}}$ ($300\text{-}500\,\mathrm{kg\,km^{-2}s^{-1}}$) for a $48\,\mathrm{h}$ residence time. Using a Eulerian diagnostic, Lamarque and Hess (1994) found an average STT flux of about $2100\,\mathrm{kg\,km^{-2}s^{-1}}$ over four days associated to a tropopause fold in a PV cutoff over the Southern US. As a

novel finding, TST balances or even dominates over STT for PV cutoffs over the subtropical ocean basins in summer and over polar regions, meaning that PV cutoffs in these regions tend to grow rather than decay by diabatic processes. PV cutoffs in regions where TST dominates over or at least balances STT are more likely to end their life cycle with reabsorption than with diabatic decay. In most regions, TST tends to decrease with lifetime and the net fluxes are strongly determined by the evolution of STT. STT is often particularly large towards the end of the life cycle and in some regions it is also large initially and follows

a u-shaped evolution.

Finally, as a step towards using the PV cutoff climatology to study their surface impacts, the chronology of the linkage between PV cutoffs and surface cyclones was investigated. A first conclusion was that the frequency with which a PV cutoff is linked to a surface cyclone during its life cycle strongly depends on the region. While in the storm track regions, most PV cutoffs are linked to surface cyclones, PV cutoffs in subtropical regions are so less frequently. By analyzing the frequencies of the possible

combinations of PV cutoff life times and life times of the linked surface cyclones, it was found that most regions have different combinations that preferentially occur. PV cutoffs can be linked to surface cyclones that formed up to a few days before PV cutoff genesis but also, as an important novel finding, they are frequently involved in surface cyclogenesis up to several days





after they have formed. This highlights that, while PV cutoffs have mostly been considered as weather systems at the end of a baroclinic life cycle, they can also be initiators of new baroclinic life cycles.

### 5.3 Three archetypes of PV cutoff life cycles

Some of the properties of PV cutoff life cycles discussed in the previous section show remarkable similarities among different regions. In the first place, this is true for the three identified genesis mechanisms, which are directly linked to the position of PV cutoff genesis relative to the jet stream(s). As identified in this study, PV cutoffs with similar genesis regions, i.e. mechanisms, also exhibit similarities regarding other dynamical aspects of their life cycle. This suggests that different types of PV cutoff life cycles exist. While such classifications have been proposed and modified for extratropical cyclones since Petterssen and Smebye (1971), a classification of the life cycles of upper-level closed cyclones based on dynamical or physical properties is almost inexistent. A first step into this direction was already made by Palmén and Newton (1969), who separated COLs into four types according to the geometry of the associated upper-level flow field [their Fig. 10.4]. A separation of closed cyclones at 500 hPa according to their relative position to the jet stream has already been performed by Bell and Bosart (1989), but mainly because of practical reasons rather than dynamical or physical differences. Price and Vaughan (1992) separated COLs into three types, depending if they were located within equatorward extensions of the subtropical jet (subtropical type COL), the polar jet (polar type COL), well poleward of the polar jet (polar vortex type COL). Thereafter, the development of COL classifications has ceased. However, we consider such a classification essential to further improve the understanding of PV cutoff / COL life cycles and their regional variability. Here, a classification into three archetypal PV cutoff life cycles is proposed that is, on the one hand, to some extent in line with the previous classifications using criteria like the jet-relative position (Bell and Bosart, 1989; Price and Vaughan, 1992) or the shape of the wave breaking (Palmén and Newton, 1969), and, on the other hand, includes additionally various dynamical and physical aspects of their life cycles. The main characteristics of these types are described in Table 3 and are in the following briefly summarized.

- Type I (*anticyclonic*) PV cutoffs (central and eastern subtropical North Atlantic, eastern South Pacific, South Africa) form equatorward of the jet stream by anticyclonic RWB most frequently over subtropical ocean basins in summer and are not very frequently associated to surface cyclones. These PV cutoffs in the subtropical ocean basins are characterized by positive net cross-tropopause mass fluxes, resulting in a downward growth of the PV cutoffs as long as they remain over this area. If they form in the eastern part of the ocean basins, they can easily get under the influence of eastward motion, resulting in their rapid decay as they are advected over land. During such rather short life cycles, they are sometimes involved in surface cyclogenesis. If they form in the central and western parts, they may remain stationary or even move westward and can have very long life times. The final phase of Type I PV cutoffs often involves enhanced STT, and diabatic decay is generally more frequent than reabsorption, in particular for PV cutoffs over South Africa and the eastern South Pacific.

- Type II (*between-jets*) PV cutoffs (California, Mediterranean, Baltic Sea, Australia) form between the polar and subtropical jet streams by anticyclonic RWB equatorward of the polar jet followed by a cyclonic break-up poleward of the





subtropical jet. They are often associated to surface cyclones with genesis shortly before or at PV cutoff genesis. Already initially STT dominates over TST and frequent diabatic decay occurs at lower levels. STT tends to follow a u-shaped evolution with the highest values at the beginning and the end of the life cycle. Diabatic decay is frequent during all phases of the life cycle, increasing towards the end and in some cases at the beginning. While diabatic decay is more
frequent than reabsorption during most of the life cycle, reabsorbtion and decay are roughly equally important or reabsorption even dominates in the final phase of Type II PV cutoffs. This is likely the case because, as they experience diabatic erosion, they approach the stratospheric reservoir and frequently a remnant of the PV cutoff is reabsorbed before it can decay completely.

– Type III (*cyclonic*) PV cutoffs (western North Atlantic, Antarctica, Hudson Bay) form poleward of the jet stream by
cyclonic RWB in the storm track regions. At their genesis, they are typically associated to surface cyclones that formed 1-3 days earlier, but can also be involved in cyclogenesis after a few days life time. STT and TST are roughly equal during the whole life cycle, while one of the two tends to be slightly larger, depending on the region (see western North Atlantic vs. Antarctica). These differences may be associated to the generally moister and warmer atmospheric conditions over the western North Atlantic than over the Southern Ocean. As a result of the quasi-balance between STT and TST, diabatic
decay is infrequent during the whole life cycle and their three-dimensional evolution is "quasi-adiabatic", i.e. not much vertical displacement occurs. Again PV cutoffs over the western North Atlantic are an exception where diabatic decay is frequent. Type III PV cutoffs end their life cycle preferentially by reabsorption.

These types are of course rough classifications and, in reality, individual PV cutoffs may undergo a mixture thereof. PV cutoffs over the Sea of Okhotsk could not be attributed clearly to one life cycle type but rater feature a mixture of Types II and III.
Despite these limitations, the proposed types may serve to better compare and contrast the highly varying evolutions of PV cutoffs in different regions of the globe. The supplementary material S3 (Figures S1-S3) illustrates example cases for each of the three types. It remains a challenge for future research to identify the underlying dynamical and thermodynamical mechanisms responsible for the major differences between the three life cycle types.

## 750  5.4  Outlook

The PV cutoff climatology presented in this study opens various opportunities for further research on this relevant flow feature. An interesting open question directly emerging from this study is for example how many of the identified surface cyclogenesis events linked to a PV cutoff can be reasonably explained by the dynamical forcing of the PV cutoff. An aspect that we will address in a forthcoming study is the role of PV cutoffs for (extreme) precipitation and surface cyclones/cyclogenesis on a
global scale. A related topic may be to quantify trends in PV cutoff frequencies and the link to precipitation trends [as for example investigated for Australia by Lavender and Abbs (2013)]. This would potentially contribute to the discussion about the role of dynamic and thermodynamic drivers of precipitation changes (e.g. Pfahl et al., 2017). Also the role of long-lived subtropical PV cutoffs (e.g. over the central subtropical North Atlantic) for the export of tropical moisture into the mid-latitudes





and subsequent heavy precipitation events [as shown by Piaget et al. (2015) for a case study] could be investigated. A potential
link to the ocean circulation could be studied by focusing on the frequent diabatic decay events and strongly negative net
cross-tropopause mass fluxes in the regions of the western boundary currents. Finally, it is an open question if and how the
diabatic modification and therefore the life cycle of PV cutoffs changes under global warming.

*Data availability.* All data is available from the authors upon request.

*Author contributions.* RP prepared all analyses and the manuscript. HW provided scientific advice throughout the whole project, helped
setting up the tracking algorithm, and provided valuable suggestions for improving the manuscript. MS provided technical support and
guidance throughout the whole project and provided valuable inputs that helped improving the manuscript.

*Competing interests.* The authors declare that they have no conflict of interest.

*Acknowledgements.* RP acknowledges funding from the ETH research grant ETH-07 16-2.



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





**Figure 1.** Schematic visualisation of the tracking methodology and the associated quantification of STT, TST, diabatic decay, and reabsorption on an isentropic surface in longitude-latitude space for situations with (a) perfect adiabatic advection of the PV cutoff, (b) adiabatic advection and the presence of diabatic processes (c) the same as (b) but with splitting, (d) the same as (b) but with merging, (e) complete diabatic decay, and (f) reabsorption with little diabatic decay. See text for a detailed discussion.



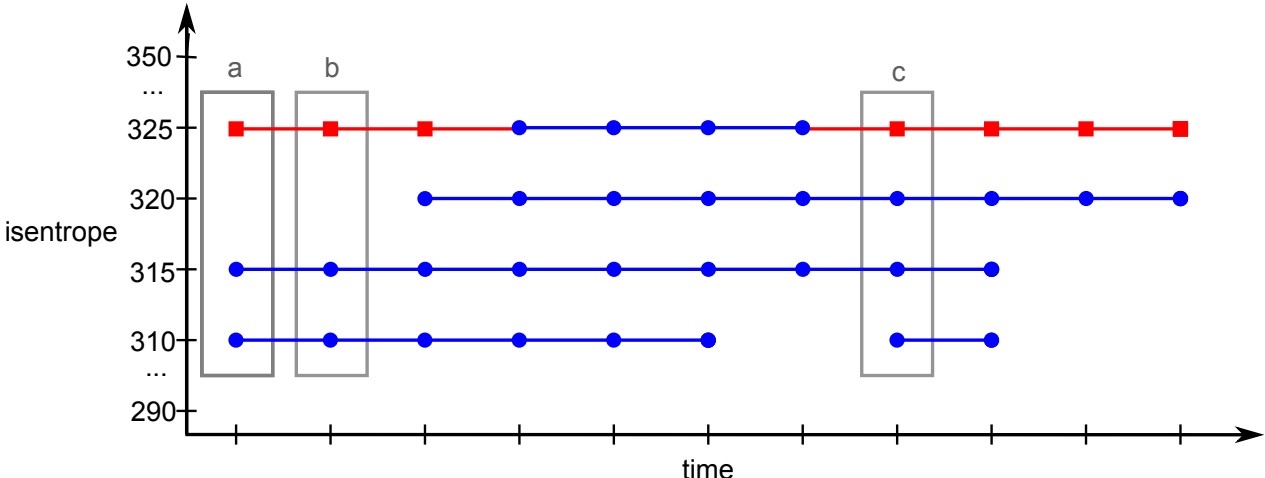

**Figure 2.** Schematic visualization of the construction of a 3D PV cutoff track from all overlapping isentropic 2D PV cutoff tracks. Isentropic 2D tracks making up the final 3D track are marked with blue lines and dots, and the tracks that are removed with red lines and squares. See text for a detailed discussion.

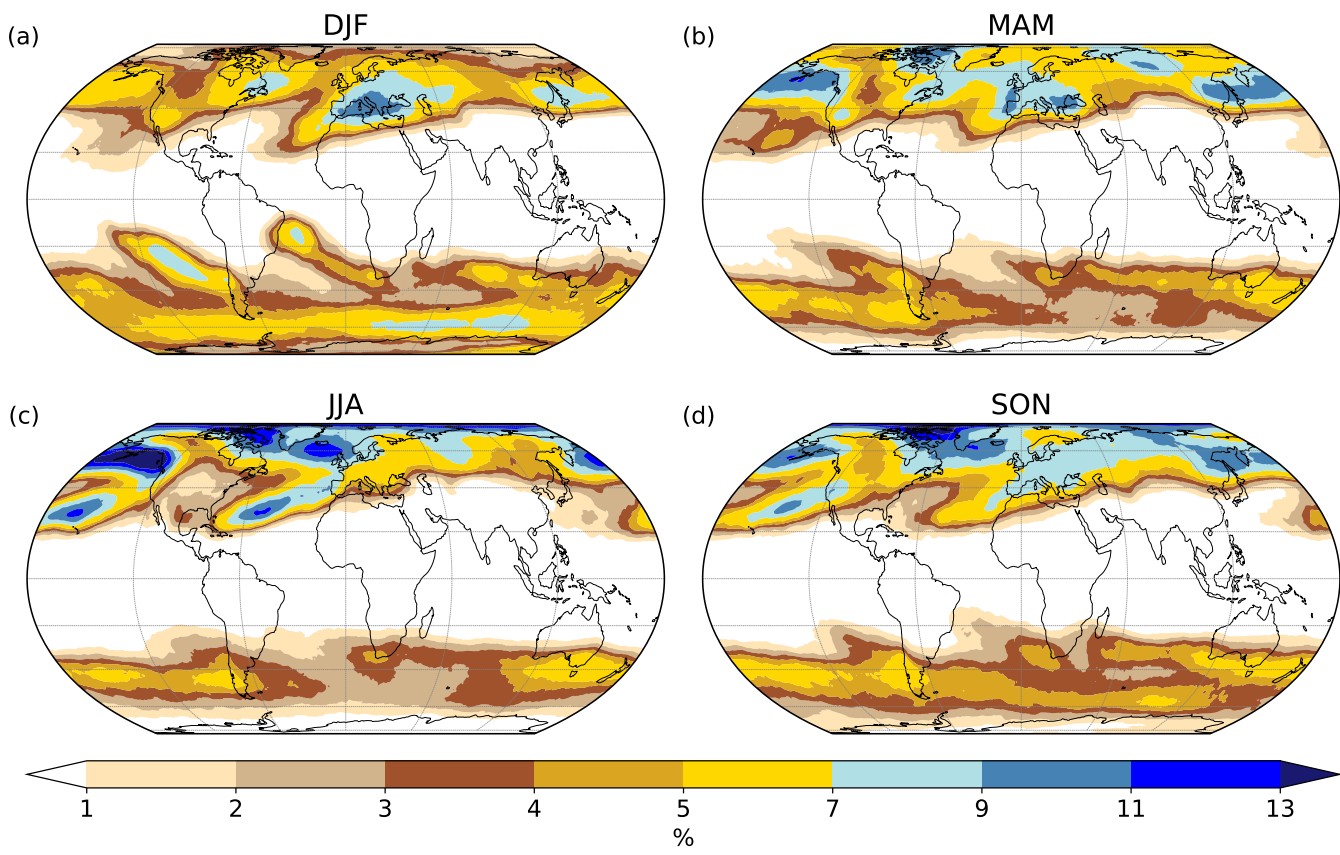

**Figure 3.** Seasonally averaged PV cutoff frequencies (shading, in %) for (a) DJF, (b) MAM, (c) JJA and (d) SON during the period 1979-2017.

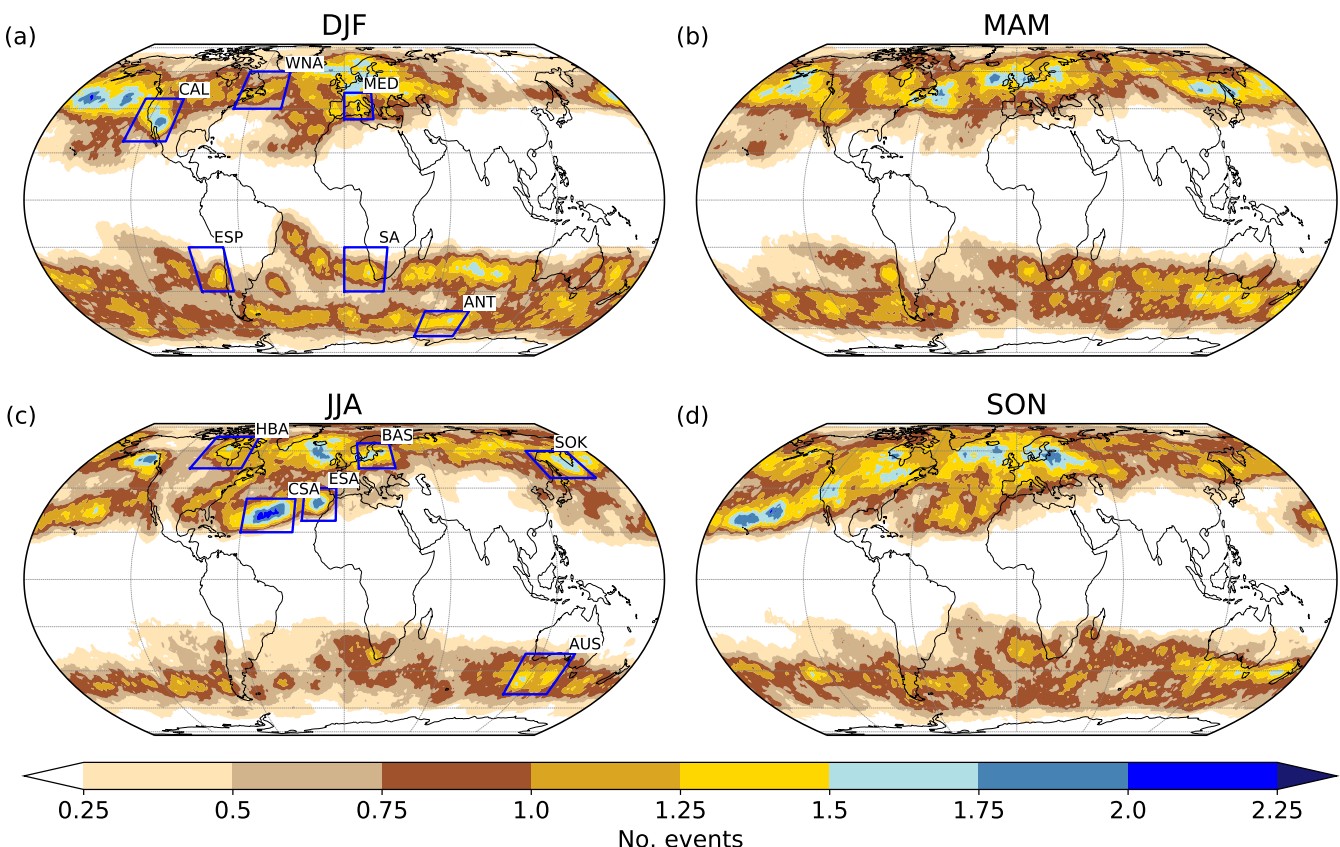

**Figure 4.** Seasonal average number of PV cutoff genesis events within a distance of 500 km in (a) DJF, (b) MAM, (c) JJA, and (d) SON during the period 1979-2017. The location of a genesis event is determined by the centre of the first PV cutoff of a track.

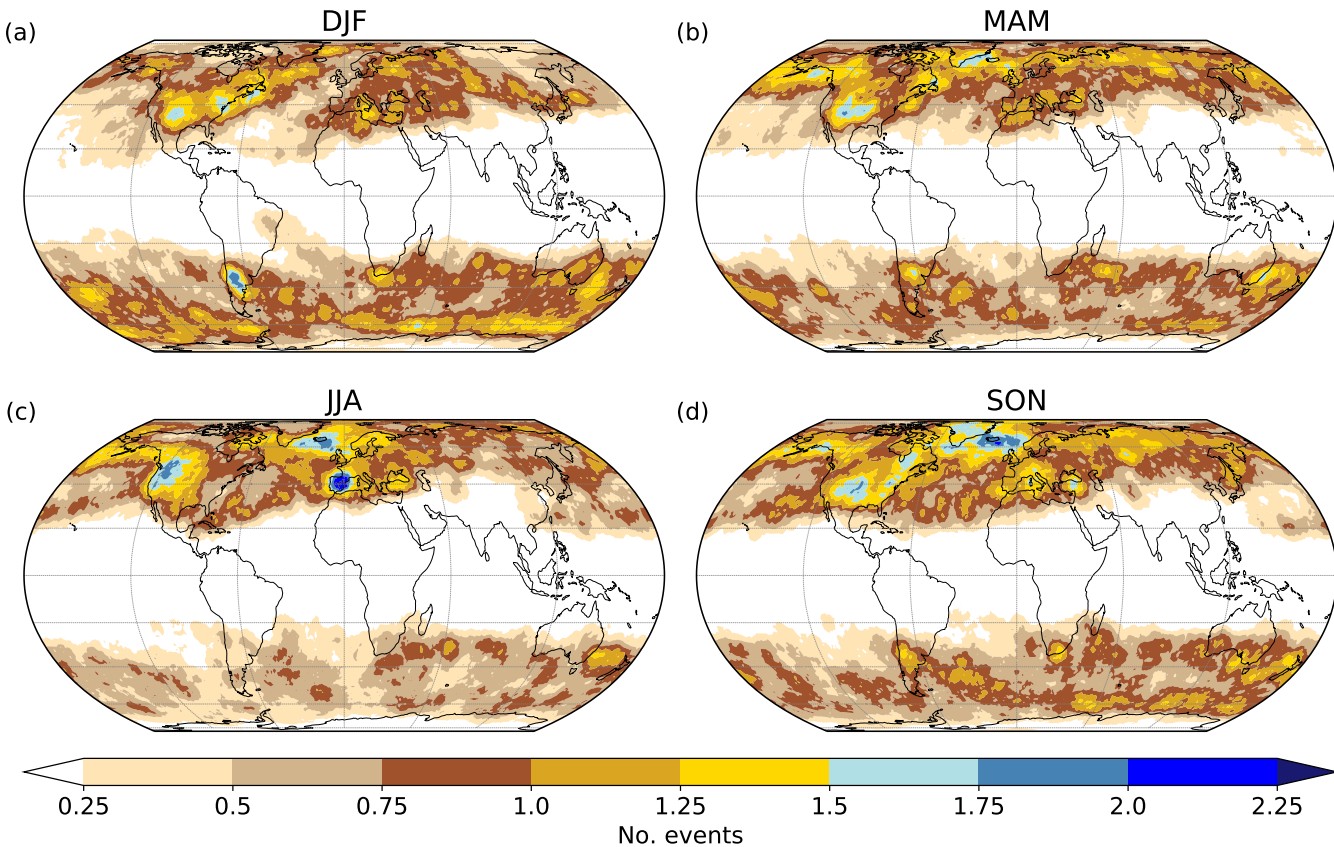

**Figure 5.** Seasonal average number of PV cutoff lysis events within a distance of 500 km in (a) DJF, (b) MAM, (c) JJA, and (d) SON during the period 1979-2017. The location of a lysis event is determined by the centre of the last PV cutoff of a track.





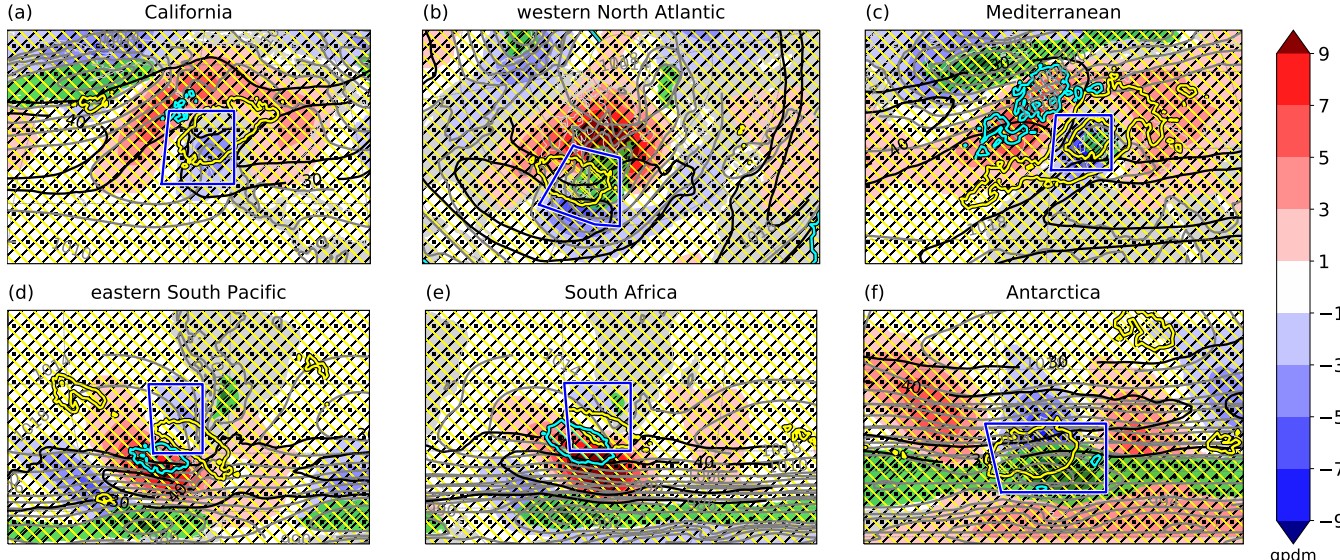

**Figure 6.** Synoptic composites during PV cutoff genesis in DJF in the regions: (a) California, (b) western North Atlantic, (c) Mediterranean, (d) eastern South Pacific, (e) South Africa, and (f) Antarctica. Shown are sea level pressure (grey contours, every 4 hPa), wind speed at 250 hPa (black contours, 30 and 40 m s$^{-1}$), anomaly of geopotential height at 250 hPa (shading, in gpdm), and frequencies of stratospheric PV streamers (yellow, 50 and 70%), tropospheric PV streamers (cyan, 50 and 70%), cyclones (green shading, 30 and 50%), dashed (dotted) if more than 5% higher (lower) than climatology.





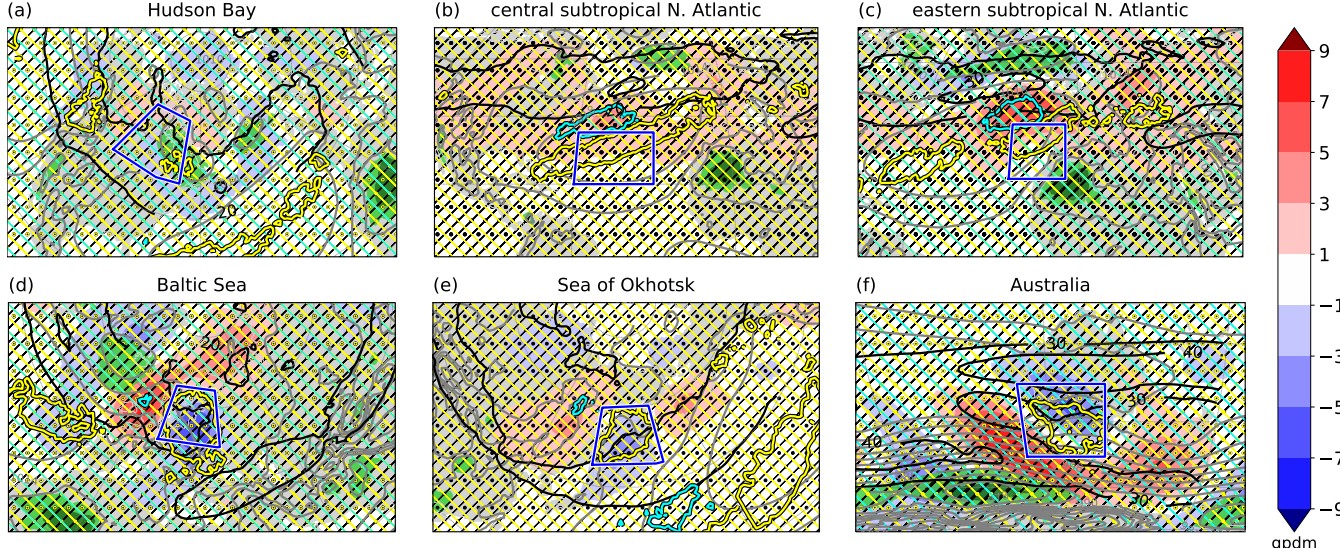

**Figure 7.** Same as Fig. 6 but for PV cutoff genesis in JJA in the regions (a) Hudson Bay, (b) central subtropical North Atlantic, (c) eastern subtropical North Atlantic, (d) Baltic Sea, (e) Sea of Okhotsk, and (f) Australia. For all regions in the Northern Hemisphere, an additional contour of wind speed at 250 hPa ($20\,\mathrm{m\,s^{-1}}$, black contour) is shown.





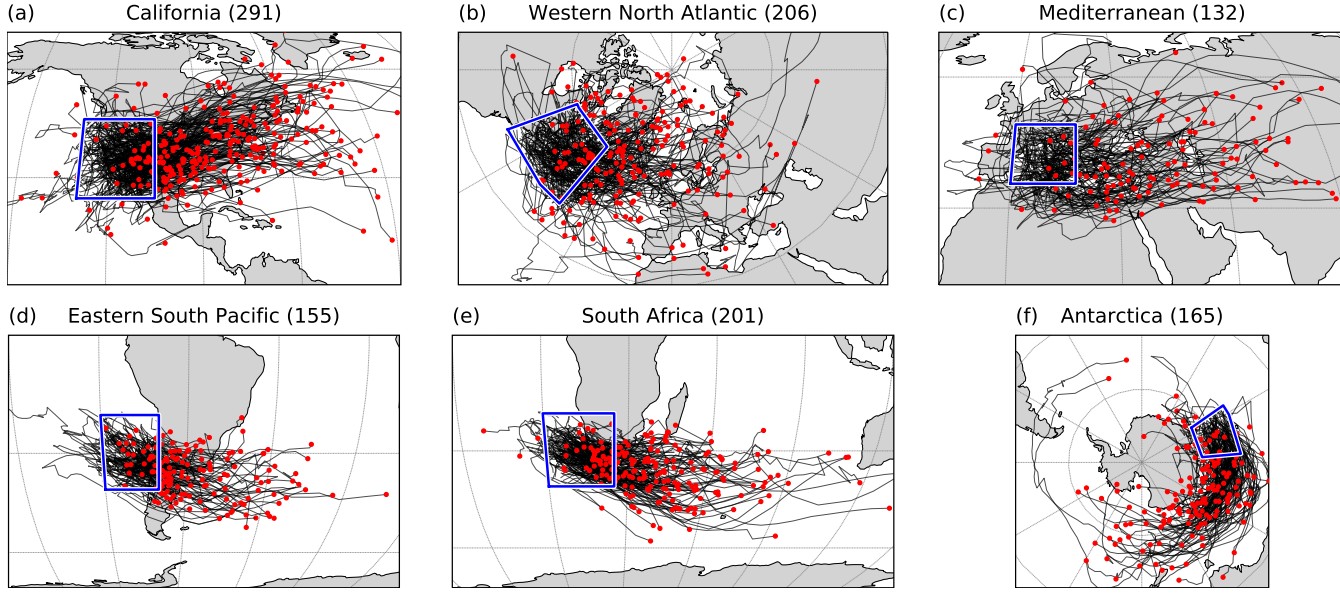

**Figure 8.** Tracks (black lines) and lysis points (red dots) of PV cutoffs with genesis over the six selected regions (blue boxes) in DJF: (a) California, (b) western North Atlantic, (c) Mediterranean, (d) eastern South Pacific, (e) South Africa, and (f) Antarctica.





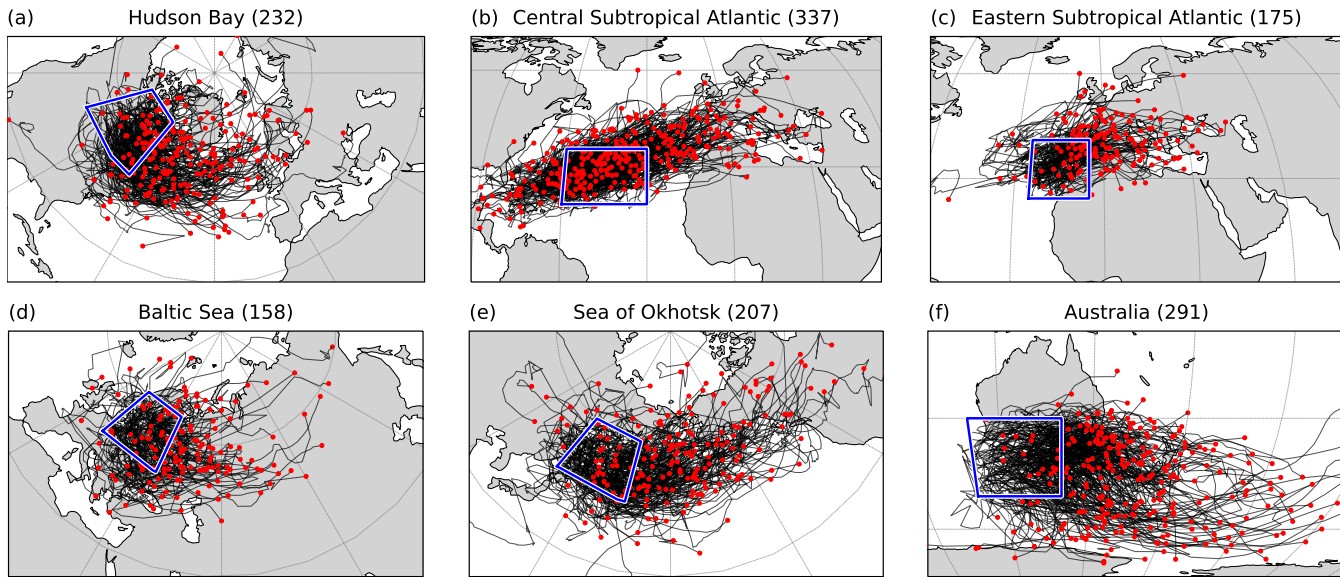

**Figure 9.** Same as Fig. 8 but for JJA and (a) Hudson Bay, (b) central subtropical North Atlantic, (c) eastern subtropical North Atlantic, (d) Baltic Sea, (e) Sea of Okhotsk, and (f) Australia.



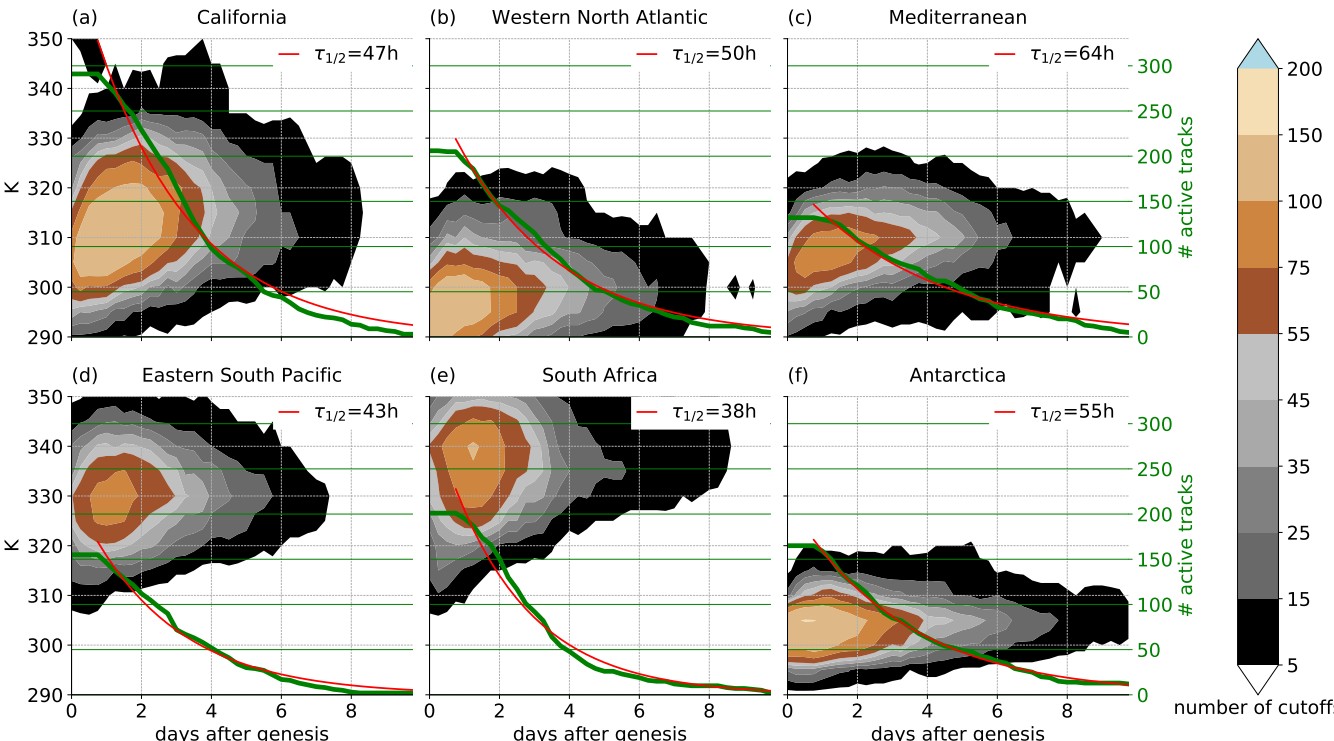

**Figure 10.** Number of PV cutoffs present on all isentropic levels (shading) and total number of active cutoff tracks (green curve) and an exponential function fitted to the number of active tracks after a lifetime of one day (red curve) as a function of lifetime for PV cutoffs with genesis over the six selected regions in DJF: (a) California, (b) western North Atlantic, (c) Mediterranean, (d) eastern South Pacific, (e) South Africa, and (f) Antarctica. Values in the upper right corners correspond to PV cutoff half lifes

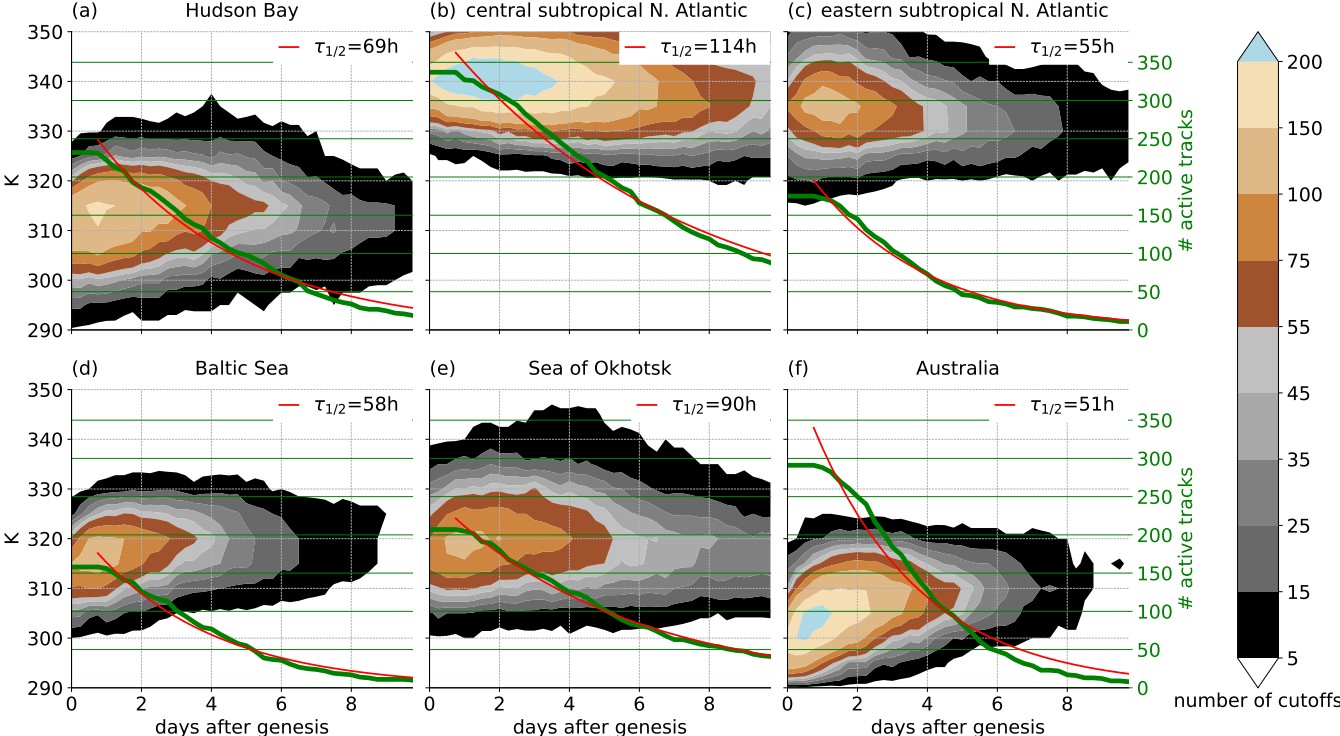

**Figure 11.** Same as Fig 10 but for JJA and (a) Hudson Bay, (b) central subtropical North Atlantic, (c) eastern subtropical North Atlantic, (d) Baltic Sea, (e) Sea of Okhotsk, and (f) Australia.





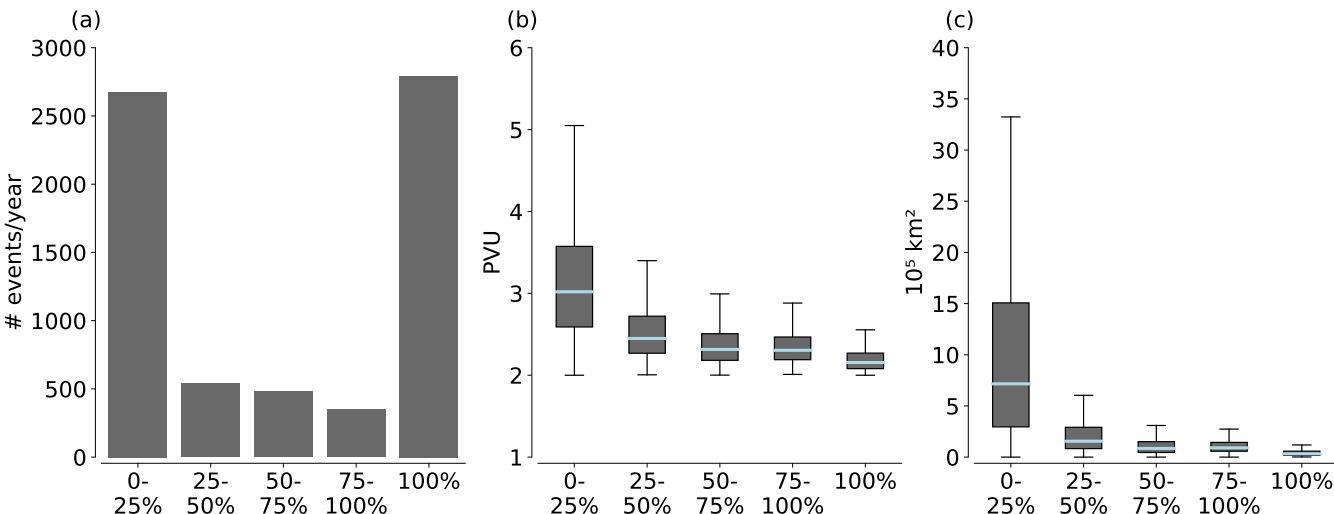

**Figure 12.** (a) Histogram of the percentage of air parcels in a PV cutoff experiencing decay whenever the PV cutoffs disappears on an individual isentrope (average annual number of events, 0% corresponds to pure reabsorption, 100% to complete diabatic decay), and (b,c) standard box plots (outliers not shown) of (b) mean PV and (c) area of the PV cutoff at the isentropic level from which it disappears for the histogram classes in (a).



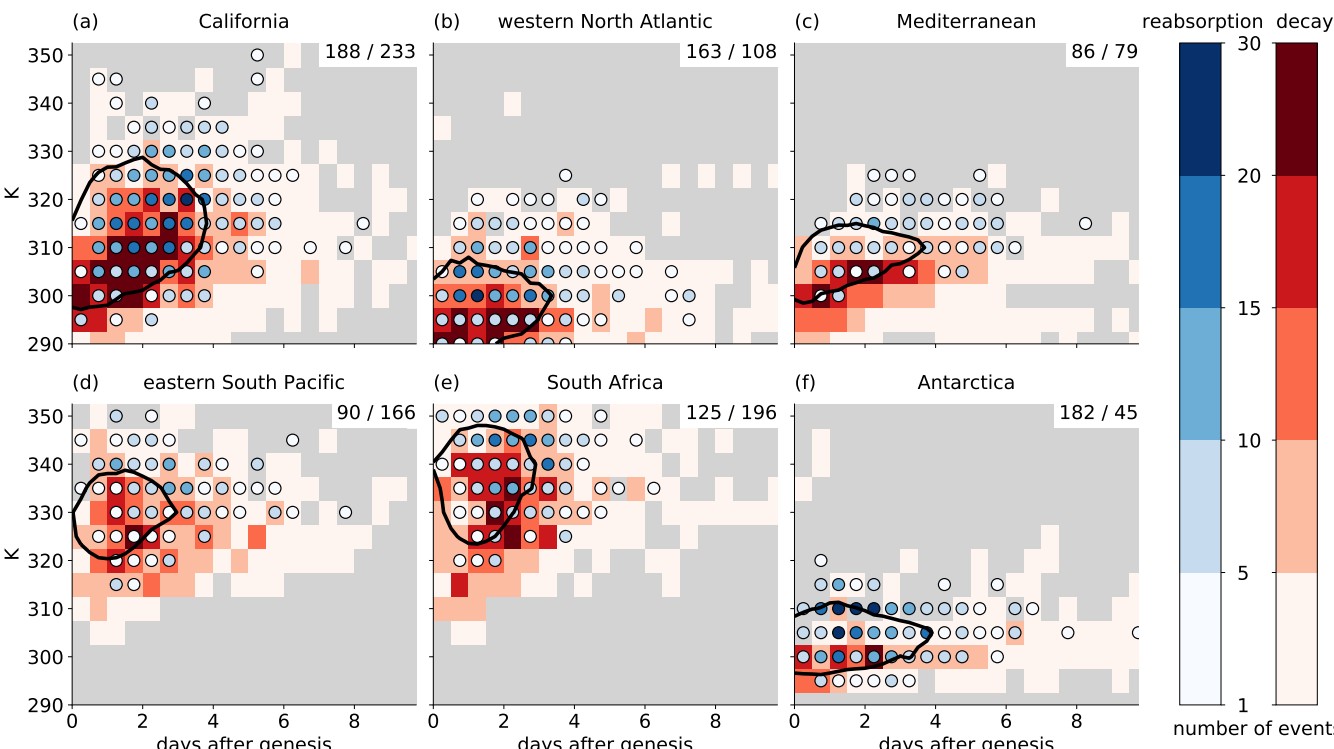

**Figure 13.** Number of reabsorption events (blue shaded circles) and diabatic decay events (red shaded rectangles) as a function of PV cutoff lifetime (binned into 12 hourly intervals) and isentropic level. The climatological vertical evolution of the PV cutoffs is indicated by the black contour (contour for 55 PV cutoffs as shown in Fig. 10). Shown are the six genesis regions in DJF: (a) California, (b) western North Atlantic, (c) Mediterranean, (d) eastern South Pacific, (e) South Africa, and (f) Antarctica.



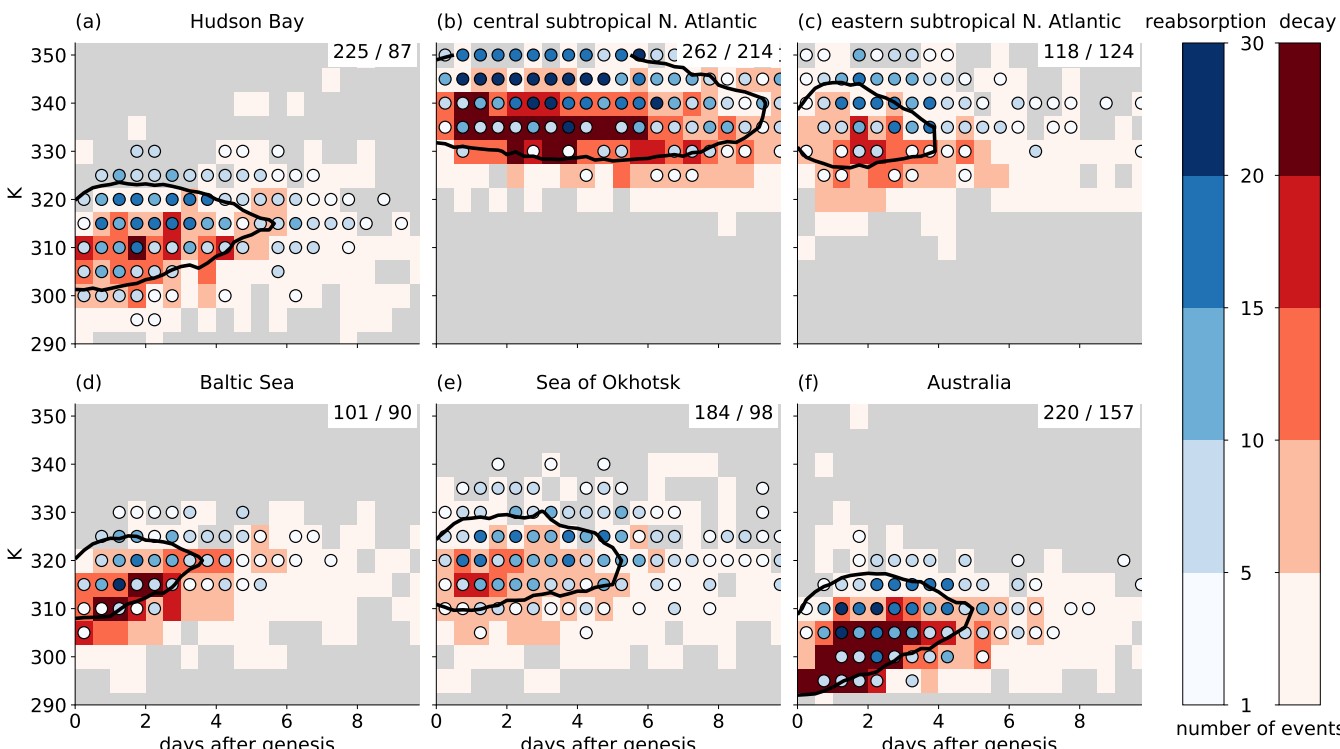

**Figure 14.** Same as Fig 13 but for JJA and (a) Hudson Bay, (b) central subtropical North Atlantic, (c) eastern subtropical North Atlantic, (d) Baltic Sea, (e) Sea of Okhotsk, and (f) Australia.



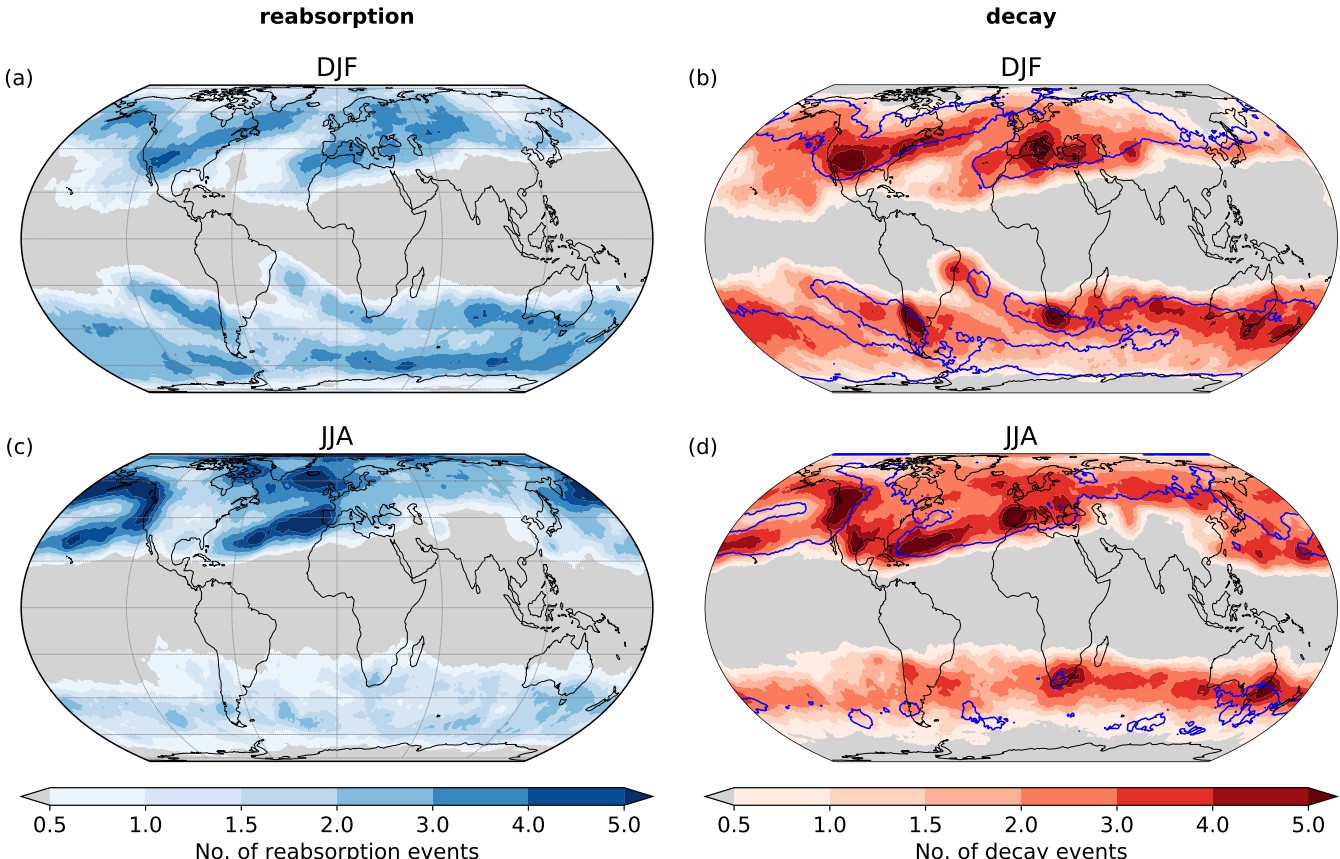

**Figure 15.** Average number of (a,c) reabsorption (less than 50% decay, blue shading) and (b,d) diabatic decay (more than 50% decay, red shading) events per season within a distance of 500 km for (a,b) DJF and (c,d) JJA. As reference, the contour for two reabsorption events per season (blue contour) is shown in panels (b,d).

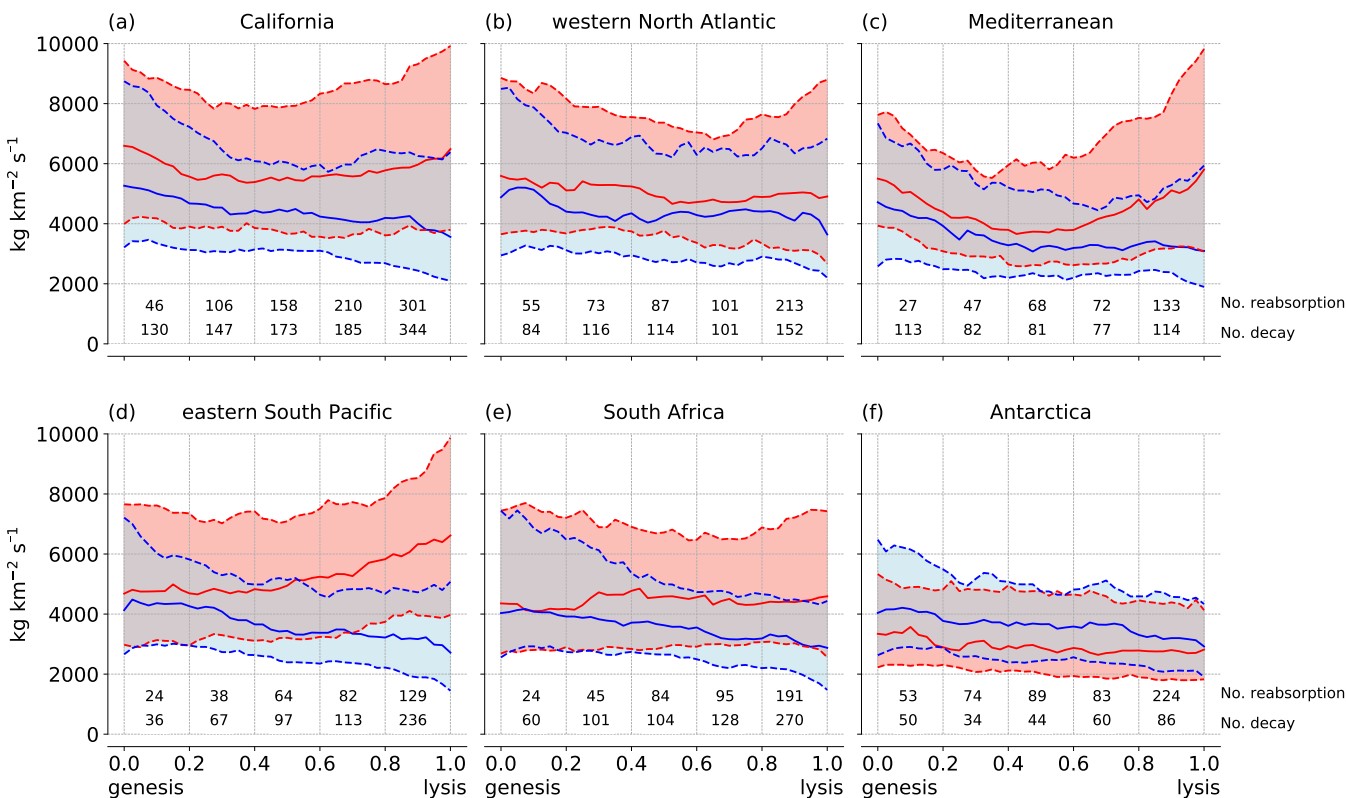

**Figure 16.** Median (solid lines) and lower and upper quartiles (dashed lines) of STT (red) and TST (blue) associated with PV cutoffs during the normalized life cycle (0: genesis, 1: lysis). The numbers in the normalized-time bins give the number of reabsorption and decay events, respectively. Shown are results for PV cutoffs in DJF from the genesis regions: (a) California, (b) western North Atlantic, (c) Mediterranean, (d) eastern South Pacific, (e) South Africa, and (f) Antarctica.





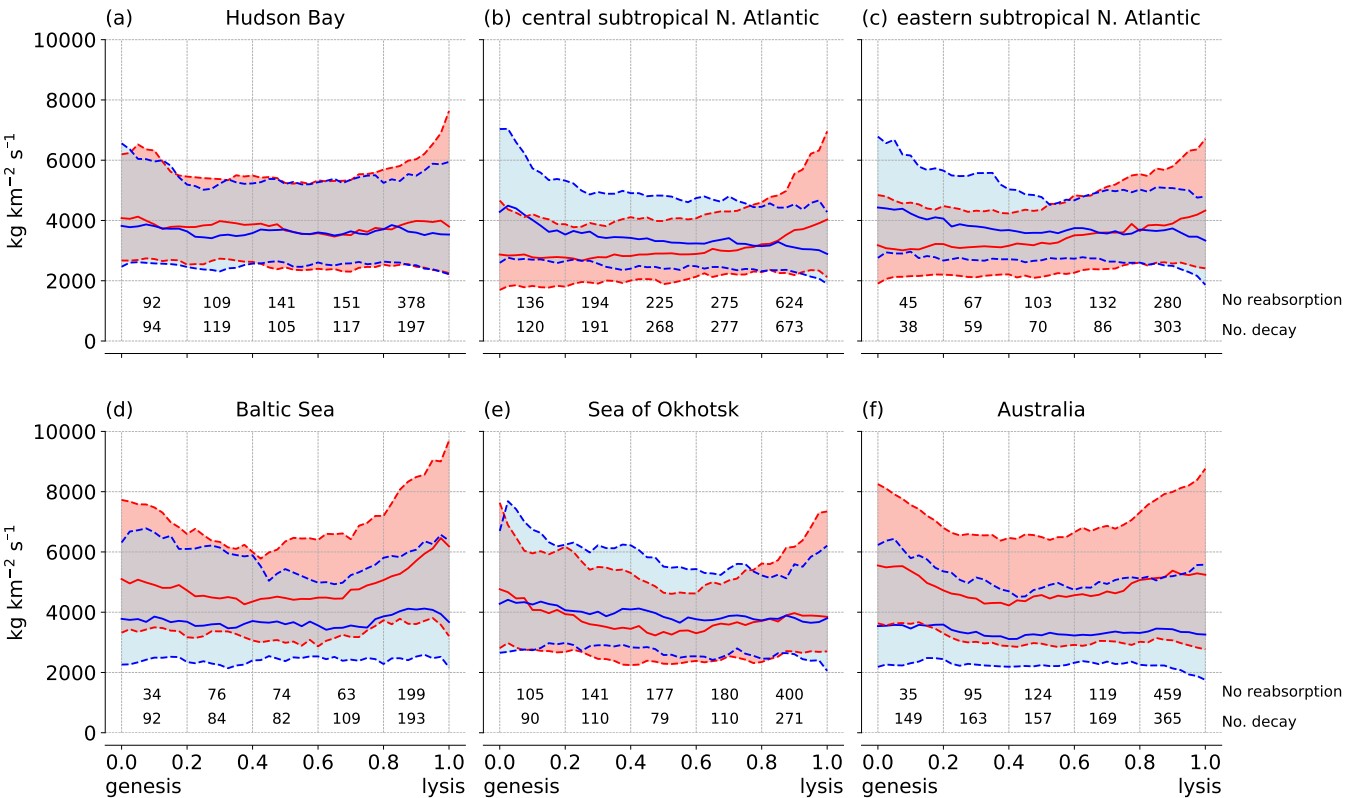

**Figure 17.** Same as Fig 16 but for JJA and (a) Hudson Bay, (b) central subtropical North Atlantic, (c) eastern subtropical North Atlantic, (d) Baltic Sea, (e) Sea of Okhotsk, and (f) Australia.

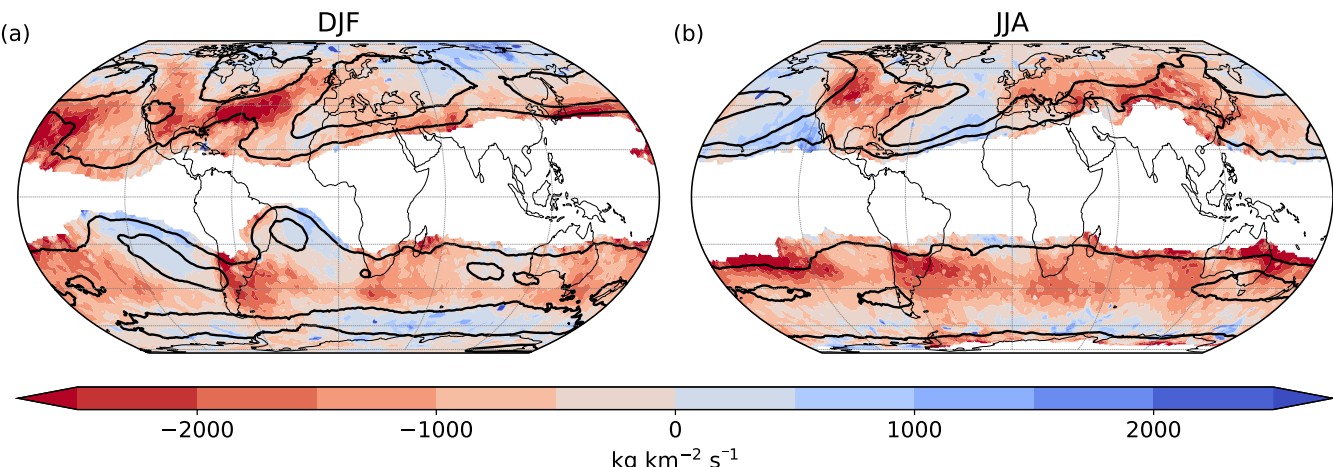

**Figure 18.** Average net cross-isentropic mass flux associated with PV cutoffs (shading) and PV cutoff frequency (black contours, 1 and 5%) during (a) DJF and (b) JJA 1979-2017. Grid points with a PV cutoff frequency below 0.25% (approximately one per season) are not shown.



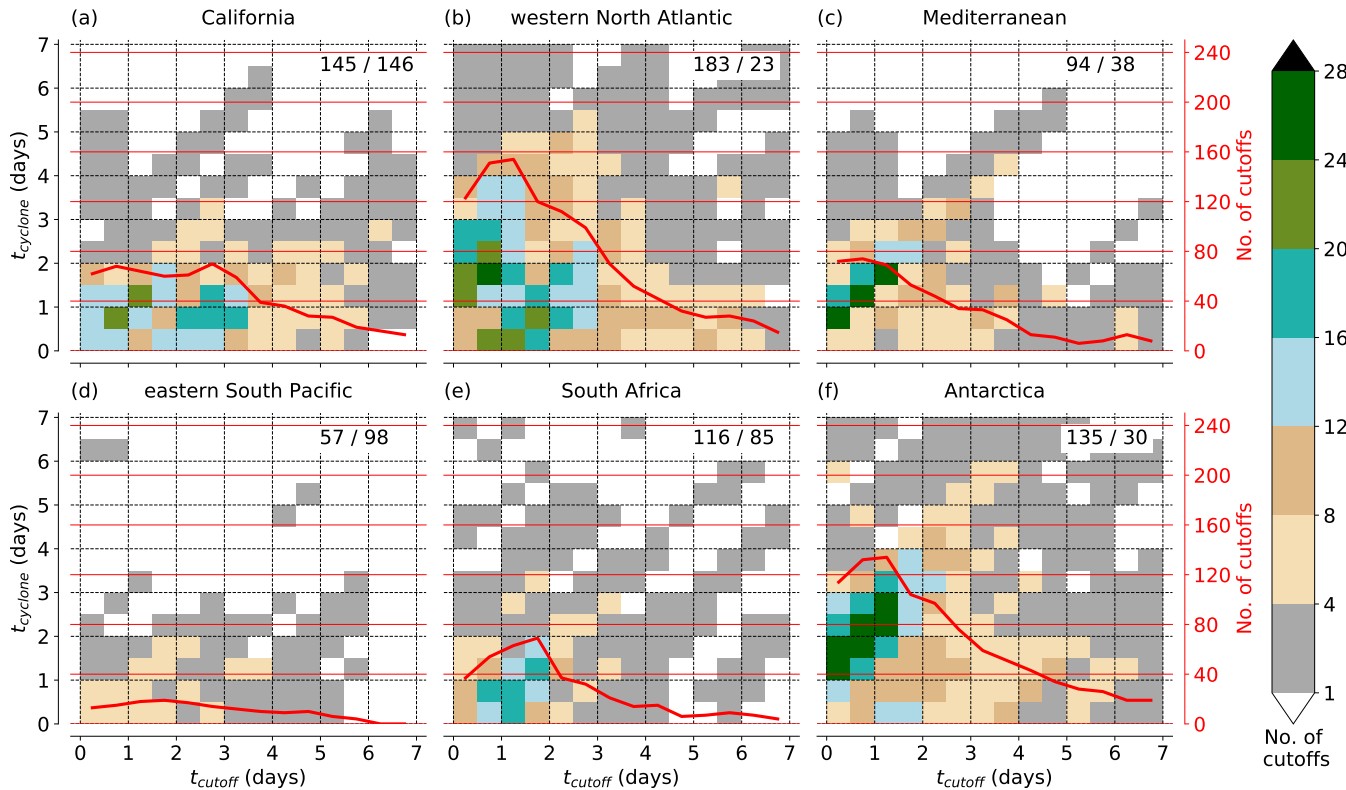

**Figure 19.** Total number of PV cutoffs linked to a surface cyclone (shading) as a function of cutoff lifetimes ($t_{cutoff}$) on the horizontal axis and surface cyclone lifetimes ($t_{cyclone}$) on the vertical axis (shading), binned into 12-hourly time intervals. The red contour shows the total number of cutoffs linked to surface cyclones as a function of cutoff life time. PV cutoffs positioned in the lowermost row of the diagram are linked to a surface cyclone within 12 h after surface cyclogenesis and PV cutoffs positioned in the leftmost column of the diagram are linked to a surface cyclone within 12 h after PV cutoff genesis. Total numbers of PV cutoffs with / without at least one link to a surface cyclone during their life cycle are indicated at the top right in each panel. Shown are results for PV cutoffs with genesis in DJF in the regions: (a) California, (b) western North Atlantic, (c) Mediterranean, (d) eastern South Pacific, (e) South Africa, and (f) Antarctica.



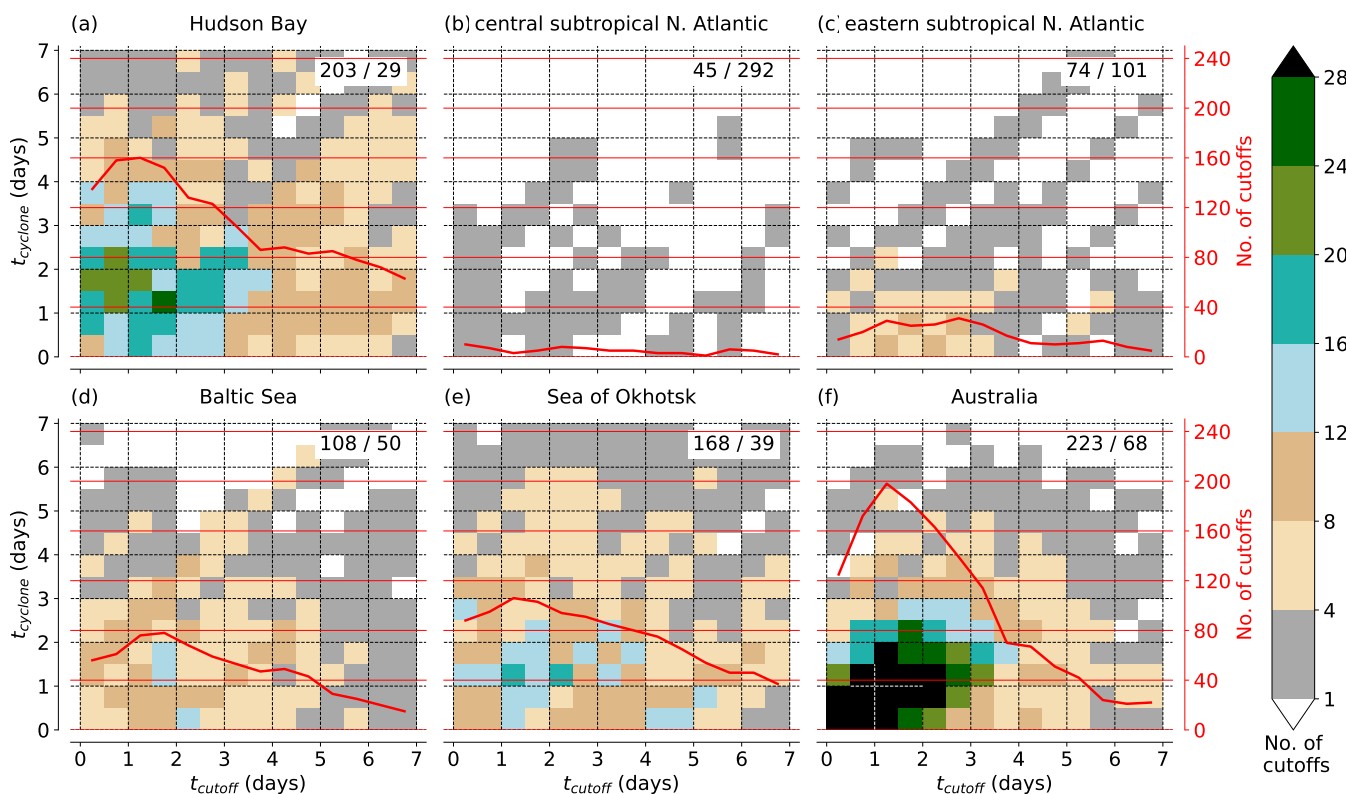

**Figure 20.** Same as Fig 19 but for PV cutoffs in JJA with genesis in the regions: (a) Hudson Bay, (b) central subtropical North Atlantic, (c) eastern subtropical North Atlantic, (d) Baltic Sea, (e) Sea of Okhotsk, and (f) Australia.



**Table 1.** Percentage of all PV cutoff tracks from 1979-2017 within four lifetime categories for the Northern Hemisphere (NH, 24933 tracks in total) and Southern Hemisphere (SH, 21420 tracks in total)

|     | <2 days | 2-3 days | 3-7 days | ≥ 7 days |
| --- | --- | --- | --- | --- |
| NH  | 22% | 33% | 27 % | 18% |
| SH  | 21% | 38% | 28% | 12% |





**Table 2.** Same as Table 1 but for four categories of the distance between genesis and lysis.

|  | <1000 km | 1000-3000 km | 3000-5000 km | ≥ 5000 km |
|---|---|---|---|---|
| NH | 15% | 37% | 21% | 27% |
| SH | 9% | 29% | 24% | 38% |



**Table 3.** Characteristics of the three proposed types of PV cutoff life cycles.

|  | Type I (*anticyclonic*) | Type II (*between-jets*) | Type III (*cyclonic*) |
|---|---|---|---|
| genesis regions | eastern South Pacific, South Africa, central and eastern subtrop. North Atlantic. | Mediterranean, California, Australia, Baltic Sea | western North Atlantic, Antarctica, Hudson Bay |
| genesis mechanism | anticyclonic RWB, particularly in subtropics | anticyclonic RWB followed by cyclonic RWB | cyclonic RWB in storm track regions |
| season | particularly summer | year around, particularly winter | year around |
| jet-relative position | equatorward | between polar and subtropical jet | poleward |
| propagation | east- or westward along tilted band or stationary | eastward | eastward |
| vertical evolution | *initially*: upward and downward growth *later*: decay at lower levels or continuous downward growth possible | decay at lower levels | not very strong vertical displacement (exception: western North Atlantic) |
| STE | *intially*: STT equal or smaller than TST *finally*: STT dominates | STT larger than TST, STT particularly large at the beginning and the end | STT and TST roughly equal; regional differences which one dominates |
| reabsorption/decay | *initially* decay not frequent, *later*: decay frequent, sometimes even more frequent downward growth, lysis occurs preferentially as decay | frequent diabatic decay throughout; reabsorption more frequent towards the end, lysis occurs slightly more frequently as reabsorption than as diabatic decay | diabatic decay not frequent (exception western North Atlantic), lysis occurs preferentially as reabsorption |
| link to surface cyclones | *frequency*: rare to frequent; often involves cyclogenesis after PV cutoff genesis | *frequency*: frequent to very frequent; often involves cyclogenesis ± 1 day around PV cutoff genesis | *frequency*: very frequent; often involves cyclones with cyclogenesis 1-3 days before PV cutoff genesis |