# Peer review of "The three-dimensional life cycles of potential vorticity cutoffs: A global ERA-interim climatology (1979-2018)"

_Weather and Climate Dynamics, 2020_

## Referee Comment (RC1) · Anonymous Referee #1 · 23 Aug 2020

General comment

The authors present a comprehensive global climatology of potential vorticity of cut-offs that considers various crucial aspects of these structures, including their global geographical frequency distribution, where they are generated and dissipate, their life cycles, tracks, decay and absorption processes, vertical structure, the stratospheric tropospheric exchange associated with them and their links to surface processes. The methodology employed is sound and the fact that the limitations of the tracking algorithm have been outlined is also good. This is a critical contribution to the current body of work in cut-off low climatologies because, not only does it broaden the scope of

published COL climatologies, it has the potential to contribute to regional predictability studies at the medium range forecasting time scale and shorter. I think that the paper is very well written. I therefore recommend publication subject to the following issues having been adequately addressed.

Specific comments

The comments in the section of this are considered major in the sense that they might require additional calculations:

1. Line 125: The feature shown in Figure 3 off the coast of Brazil, is in my opinion, a very important phenomenon for that part of the SH but might be under estimated on the 350 K isentropic surface and previous experience suggests that it is most clearly seen on the 360 K isentropic surface. If one plots the climatology 360 K PV = -2PVU contour for DJF for instance, a clear PV overturning region (or at the very least a region of very strong undulations) is seen in that part of the SH, thus showing the dominance of that structure. I therefore suggest that, at least for the SH, the analysis be extended to include the 360 K isentropic surface.

2. Section 3 is good but could benefit from integrating the effect of the zonal flow. So I suggest that the authors consider superimposing the zonal flow isotachs in Figures 3 to 5 and consider seasonal changes in the flow and the impact of these on the seasonal frequencies discussed in this section. I think that we would see dominance of these frequencies in areas of diffluent and confluent flow. It would also provide an opportunity to view the frequencies relative to the jet core. These can be inferred by the experienced reader, but to render this discussion accessible to those readers who are not familiar with RWB dynamics (as noted in the manuscript, the cut-offs are residues of RWB events), this could improve the readability and impact of the study.

3. The discussion in Section 4.1 is very interesting and has interesting and important implications for the various regions considered there in. The region off the coast of Brazil might benefit from such considerations (Figure 18 illustrates the importance of

this region and as reviewer I feel it requires as much attention as the other regions that have been considered in the study). Many studies have shown that it is characterised by cyclogenesis. Please consider an additional regional analysis for that region.

4. Could the authors consider composite evolution of at least some of the regions? For instance, with no bias from myself as a reviewer but based on some experience of knowledge of the region thus my use of it to illustrate my point. There exists an interesting phenomenon that occurs in the South Atlantic Ocean/South African domain referred to as the ridging of the South Atlantic Ocean Anticyclone (SAOA) that has been shown to be influenced by lower stratospheric processes. It is most clearly seen in the MSLP pressure fields and is defined as an extension of the SAOA eastward and this ridging process has important implications for the region. Also in South America there is a strong presence of Rossby wave trains that propagate into that continent from the west of Drake Passage, aspects of which could be characterised by this analysis (the 250 hPa anomalies indicate this in Figure 6). I am certain that there are other interesting dynamical processes in the other regions that are unique to those regions that could be alluded to in this discussion, if the authors so please. All in all, I think that evolving composite could reveal some of these regional dynamical issues.

Minor comments

1. In Fig 6 please use hatched instead of dashed. Also are the composite means statistically significant in that figure?

2. Line 372 please replace South Pacific with Indian Ocean.

---

## Referee Comment (RC2) · Anonymous Referee #2 · 2 Sep 2020

General comments

The authors present a global climatology of cutoffs for the ERA-interim period. The introduction nicely motivates cutoffs as highly relevant atmospheric features. The study contains several novel and valuable aspects: I commend the authors on taking a potential vorticity and Lagrangian (trajectory) perspective to identify and track cutoffs. This approach enables to authors to provide a global perspective because the different levels on that cutoffs "live" in different latitudes is automatically accounted for. Furthermore, the 3D aspect of the cutoff evolution is of particular interest. A further potentially intriguing result, the classification of cutoff life cycles, however, remains mere speculation.

My assessment of the manuscript in its current version is much more critical than that of the Anonymous Referee #1 (available online at the time of this writing). To me, the authors miss most of the potential that a feature-based climatology offers to link the evolution of cutoffs to other dynamical features of interest. My main conceptual concern here is that the authors stratify the data with respect to small geographical regions and then present "composites" centered on these regions, instead of centering composites on the identified feature (here: the cutoffs), which is an often-used and more powerful approach to feature-based analysis. The authors' geographical approach severely affects the quality of this study. Consequence of this approach are that i) only a small percentage of their data contributes to this analysis, casting doubt on the robustness of the results; ii) the relation to other dynamical features is diagnosed in a rather indirect sense and the differences between regions are often not clear enough to provide sound evidence for the authors' results. Too much of the presentation is thus suggestive, indicative, speculative; iii) pages of presentation are spend on discussing differences between, e.g., cutoffs over the Sea of Okhotsk as compared to cutoffs over the Baltic Sea or the Hudson Bay, making for a tedious, cumbersome read. For a manuscript submitted to a journal with "Dynamics" in its name, I would have hoped that this study provides a much more direct assessment of dynamical characteristics and relationships. I have further concerns with respect to a conspicuous lack of testing the sensitivity of results to (inevitable) free parameters and statistical significance. In addition, I question the accuracy of the authors' computation of nonconservative tendencies as a residuum from diagnosed adiabatic evolution.

I elaborate on these criticisms below in the section "Critical specific comments", where I also suggest means to alleviate my concerns. Some of this elaboration duplicates comments given above. In addition, I have several "Major specific comments" for the authors' consideration. A few "Minor specific comments" are provided to parts of the manuscript that I believe do not need to undergo major revisions, but I refrain from pro-

viding comprehensive minor and technical comments on the manuscript in its current form.

Comprehensively addressing my concerns will require substantial further analysis and revisions. I am optimistic, however, that this work will eventually be publishable. It will then make a very valuable contribution to a weather-system perspective on climate dynamics. I leave it to the editor's or the authors' discretion whether the manuscript should be rejected or withdrawn at this point and re-submitted after substantial revisions, or whether these revisions can be successfully implemented during the limited time of the regular review process.

Critical specific comments

1) Stratification by (small) geographic regions

The authors stratify the data almost exclusively by (small) geographic regions of cutoff genesis. As the only motivation for this approach, the authors claim that these "regions are selected each such that a broad spectrum of regions is covered" (line 310). No qualification of this claim is provided. This choice of stratification has several implications that severely affect the quality of the manuscript. i) Most of the presentation reads like an enumeration of characteristics and pages of discussion are spent on describing, e.g., characteristics of cutoffs over the Sea of Okhotsk as compared to those of cutoffs over the Baltic Sea or the Hudson Bay. This approach makes for a tedious, cumbersome read. More severely, it remains opaque to me what can actually be learned from this kind of presentation. ii) For a manuscript submitted to a journal with "Dynamics" in its name, I would have hoped that this study provides a much more direct assessment of dynamical characteristics and relationships. I would find it much more interesting to learn about how cutoff occurrence, characteristics, and associated transport relates to other features of the large-scale circulation, rather than how they differ in several geographic regions. The authors too rarely attempt to make such direct links to other synoptic-scale features or to characteristics of the large-scale circulation. iii) The authors do attempt to relate cutoffs to other features of the large-scale circulation in an indirect way by providing "composites" of relevant other features. (It remains unclear if the authors re-center the data on the cutoffs or if they merely provide averages of fields whenever a cutoff occurs in the respective region in subsection 4.1. This reviewer suspects the latter.) From this approach we learn that in some regions cutoffs co-occur 50-70% of the time with feature A and that the occurrence frequency of feature B is 5% higher than climatology. Accordingly, rarely any strong conclusion can be drawn and much of the presentation needs to remain suggestive and indicative. Overall, much variability is found and if statistical significance testing were applied to differences between regions, I believe that little of the signal would stand that test. Strictly speaking, the presented observations are not incorrect. To this reviewer, however, this choice of presentation gives away much potential of the feature-tracking approach and provides currently little insight.

On a constructive note, I suggest to more fully exploit the potential of the feature-tracking approach by constructing composites centered on cutoffs (as is done in subsections 4.3 – 4.5) and, importantly, stratify the cutoffs directly by their relation to other dynamical features, e.g., co-occurrence of wave breaking or streamers, their distance to and location relative to the jet, etc. Another advantage of this approach is that lagged composites can be created easily, too, such that this approach would tell us much more directly how cutoff life cycles associated with, e.g., cyclonic wave breaking differ from that associated with anticyclonic wave breaking. Another approach would stratify the cutoffs by cutoff characteristics or by their relative location to, e.g., characteristics of the storm tracks. I believe that such approaches centering on dynamical rather than geographical features will substantially enhance the significance of the results and will make for a much more succinct and concise presentation.

2) Lack of sound evidence: Presentation is suggestive, indicative, speculative

This point partly relates to comment 1). In addition, however, further analysis within the chosen approach could avoid speculations made by the authors. In section 3, e.g., the

authors speculate on cutoff track characteristics based on genesis and lysis location but do not substantiate this speculation with an analysis of the actual tracks (which are available and presented later in the manuscript). Similarly, a claim is made that cutoffs behave differently over land than over the ocean, which could be easily tested with the authors' data. The analysis of cutoff tracks is a further example: Interpreting and drawing conclusions on different track characteristics from visual inspection of Fig. 8 is a stretch, to say the least. I strongly recommend that the authors reduce speculations and hypotheses to a minimum and extend their analysis to substantiate their conclusions whenever possible. Most severely, I find one of the main results, the proposed classification of cutoff life cycles, to be mere speculation. This result appears prominently in the abstract and the conclusions but, strictly speaking, is not a synthesis of the results presented in the main text. Cutoff characteristics in this study are largely characterized by substantial variability. It seems inconsistent that this large variability can be synthesized into the classification of three types of life cycles that then govern other characteristics (stratospheric-tropospheric exchange) also. A sounder analysis would have established statistically significant difference between different aspects of cutoff life cycles before presenting such a strong result. Life cycle type II seems to be particularly dubious. In the results section, one and a half sentences are spent to establish the fact that this life cycle is associated with anticyclonic wave breaking followed by cyclonic wave breaking. I am not aware that this sequence of wave breakings is a well-established concept and I have a hard time to follow the authors description of the respective figure to clearly identify this sequence. The underlying issue is that the classification of life cycles is proposed based on a by-product of mean characteristics in different geographical regions (see comment 1 above). If composites were created, globally not only within the small selected regions, with respect to the wave breaking characteristics, and time-lagged composites were presented that showed the actual mean-evolution, the authors would probably find a much cleaner signal and overall the presented results would be much more convincing. Such an analysis could establish different types of cutoff life cycles based on sound evidence and not on speculation

and would be a great result of this study.

3) Robustness and statistical significance of results

The authors do discuss limitations of their tracking method but further discussion and tests of the robustness of the results are missing. There are five major issues: i) About 70% of potential cutoffs are dismissed based on an arbitrary criterion (section 2.3.2). No doubt there is a need to filter out spurious cutoffs. The authors filter out cutoffs that have small vertical extent based on the flawed argument that these cutoffs are not "dynamically relevant". Theory does not support this argument. According to QG theory, the aspect ratio of a PV anomaly determines the ratio between the associated stability and wind anomalies, but not the extent to which the anomaly is "dynamically relevant". The vertical penetration depth, e.g., depends on the horizontal (not the vertical) scale of a PV anomaly. And the magnitude of a PV anomaly is directly related to the magnitude of the stability and wind anomalies. These two characteristics would thus be more suitable to identify dynamical relevance. In any case, for any free parameter that is introduced to filter out a large amount/ the majority of potential data, the robustness of the results to reasonable variations of this parameter needs to be demonstrated. ii) After showing the geographical locations of genesis, lysis, and tracks (not that I am mistaken: I consider this an important documentation of the occurrence characteristics of cutoffs.) the authors restrict their analysis to small geographic subregions during DJF and JJA. These subregions constitute maxima in genesis frequency, but no information is given what percentage of total cutoffs occur within these regions. Visual inspection of Fig. 4 suggest that this percentage is low. (It's hard to estimate a number but I guess less than 20%.) Cutoffs during MAM and SON are not considered. So overall only about 10% of the data is used to draw conclusions from. In combination with the lack of justification of why these 10% are selected I fail to see how the results can be considered to be robust and how the results would meet the expectation of a global climatology, as suggested by the title. iii) Basing the tracking of cutoffs on trajectory calculation is a major strength of this study. The trajectories are used also

to determine the adiabatic evolution of cutoffs. Nonconservative contributions to the evolution, including stratospheric-tropospheric exchange, are estimated as the difference between the observed and the diagnosed adiabatic evolution, i.e., as a residuum. Estimating nonconservative terms as a residuum requires a very accurate diagnosis of the conservative dynamics and is usually associated with substantial uncertainties in the results. No discussion of this issue is provided in the manuscript. Trajectories are calculated for 6h, which is seemingly short but in fact 6h is the temporal resolution of the underlying data. This means that trajectories, irrespective of the time step used in the calculation, are based on linear interpolation between the start and end times. Linear interpolation is a poor approximation when the displacement of strong gradients needs to be represented, here: the movement of the strong PV gradient and associated winds associated with the jet, wave breaking, and the cutoff itself. In addition, any nonlinearities in the adiabatic 6h evolution will erroneously be diagnosed as nonconservative contributions. Such nonlinearities can be expected during genesis (wave breaking) and reabsorption of cutoffs and I wonder to what extent putative maxima of nonconservative processes at the beginning and the end of cutoff life cycles, e.g. the u-shape signal in Figs. 16 and 17, are related to inaccuracies in the trajectory calculations. The robustness of the results should here be demonstrated preferentially by using higher-resolution ERA5 data for a sufficiently large subset of cases or, as a less accurate but computationally less expensive option, by reducing the temporal resolution of the ERA-interim data to 12h. iv) The authors examine stratospheric-tropospheric transport (STT and TST) based on 6h trajectories. Previous authors considered results based on this relatively short time scale as spurious and advocate using longer time scales. For this issue, the authors do provide a discussion in their final summary and discussion section. To avoid readers' confusion, I would find the discussion more helpful closer to the method or the results section. More severely, I am not sure that I find the comparison of rather general characteristics with previous results to be convincing enough to demonstrate that the transport characteristics diagnosed in the current manuscript for different "types" of cutoffs is robust to the time scale used in the trajec-

tory calculation. A sound test of the robustness would extent the trajectories used in the analysis to longer times, again at least for a sufficiently large subset of cases. v) There is a complete lack of testing the statistical significance of characteristics of cutoffs in geographical regions/ different types of cutoffs. This lack is significant because the descriptions/ results rely on the statistics of the data only and it is often not clearly evident to the trained eye that differences are sufficiently large. The revised version of the manuscript requires such tests.

Major specific comments

1) The conclusion that can be drawn from this study are hard to identify

The final section contains both, an extensive discussion of results in the light of previous studies and the conclusions drawn from the current study. As a consequence, the final section is unusually long and the summary is little helpful for a reader who is interested in the main outcomes of this study. To me, most of the discussion of previous result would have been more useful in the results section, after presenting the relevant result. This re-organization of the text would improve readability of the results section because more context for the results presented therein is provided. More severely, in the current version, it is hard to identify what the main results of the current study are. If the authors chose to keep a long final discussion section, then at least final conclusions should be clearly separated from the comparison with previous studies.

2) Section 3

It is hard for me to see what we learn in section 3. I understand the importance to show and describe the geographical distribution of feature occurrence and characteristics. In its current version, however, the presentation is mostly show and tell and thus rather unsatisfactory. One issue that can be remedied is that the authors partly use very specific geographic references (e.g. New Foundland, Iceland). First, an accuracy of the analysis is suggested that I believe is not robust to reasonable changes of free parameters used for cutoff identification. Second, using more general geographic references

(over land – over ocean, subtropics – polar – midlatitudes) seem more related to potential differences in the dynamics. Most preferably, a description in relation to large-scale features (start – end of storm tracks, at the edge of subtropical anticyclones) would be given.

3) Subsection 4.1: Cutoff genesis

Unfortunately, I have a very hard time to follow the authors' arguments in this subsection based on occurrence frequencies of other features. In addition, I struggle to clearly identify the features described in the text in the figure. In particular, I struggle to identify wave breaking, which plays a crucial role in the authors' argument. Do the authors mean to indicate different types of wave breaking by tropospheric and stratospheric streamers? Inter alia in this particular subsection the authors draw strong results based on rather speculative arguments. To illustrate the temporal evolution leading up to genesis, time-lagged composites (going back in time) could be created, in a feature-centered framework e.g. centered on genesis location. Similar composites are used later to illustrate temporal evolution and are very helpful (Figs. 10 ff.). I recommend substantial revision of the complete subsection.

4) Subsection 4.2: Track analysis

Visual inspection alone is insufficient to draw conclusions about different types of tracks. A cluster analysis of tracks, or at least showing the average tracks of cutoffs in distinct end regions are required to make a point here.

5) Subsection 4.3 – 4.5 are the strongest result subsections but still suffer from the general critical issues noted above.

6) Subsection 4.6: Link to surface cyclones

There have been decades of research based on first principles to understand the interaction between upper-level PV anomalies on surface cyclones. What do we learn here about this interaction by considering mere proximity of the systems? What aspects of

the interaction do you consider? Why is it relevant to consider the age of the cyclone? In what sense does the analysis provide new insight? What is the significance of this result? I see potential in this subsection but there is a severe lack of motivating the important I questions and why proximity of the systems is a useful metric. Potentially the issue here is mostly a matter of wording. The current version, however, leaves the reader guessing why this subsection would provide interesting results.

Minor specific comments:

Fig. 1 is very helpful, but Fig. 2 did not help me to better understand the tracking procedure.

The introduction reads in general very well. Except for the two middle paragraphs on page 3. There it is not clear to me what the open question is and/ or how the respective paragraph motivates the current study. Please clarify.

How do you handle a lost child? Are they simply ignored? If yes, how much of the volume of cutoffs are dismissed due to a lost child?

I fail to understand the first half of subsection 2.3.2 (Construction of tracks)

What motivates your choice of definition of cutoff area? Why not using the vertical average of the area on the individual isentropes? A single isentrope with large area would dominate your metric and I would not consider the area as representative in that case. Or take, e.g., a tilted structure with a similar area at all levels. Your method artificially enlarges these features. Please add some discussion for clarification.

Avoid technical language when describing a physical feature. E.g., the term "cutoff track" is often used to describe what I believe is simply the life time of the cutoff (e.g., line 301 and in subsection 4.3).
* * *

---

## Author Comment (AC1) · 26 Nov 2020

**The three-dimensional life cycle of potential vorticity cutoffs: A global ERA-interim climatology (1979–2017)**

**wcd-2020-30**

**Author comments**

We thank both reviewers for the time they took to read and comment on our manuscript. We carefully considered the many useful remarks from both reviewers in order to revise and improve the paper. Based on the reviewers' comments, we will implement some substantial changes to the manuscript. In addition to improving readability we will provide an improved balance between analyses using the global dataset and the regional subsets. More specifically, we will implement the following main changes in the revised version of our manuscript:

- We slightly change the structure of the paper and first show all analyses that are based on the full (global) set of PV cutoffs, and only then add a specific section about regional characteristics.
- We strongly shorten the part about stratosphere-troposphere exchange in order to keep the overall length of the paper reasonable; instead we will refer to these results as published in the first author's PhD thesis (Portmann 2020).
- We have redone the classification of PV cutoff lifecycles (previously presented late in the manuscript in Section 5.3) with a more objective approach and present these results now earlier as a central part of the paper. This results in a replacement of the regional composites in section 4.1 with cutoff-centered composites based on the full (global) set of PV cutoffs, as suggested by referee #2.

However, we also have been surprised about the, as we felt, harsh and very critical tone of some comments by referee #2. Some of these comments we try to refute (see details below) because we regard parts of the critique mainly as a strong expression of the referee's personal viewpoint of how such a climatological study should have been performed. We already here like to mention the important point that a new climatology of a multi-faceted dynamical flow feature like PV cutoffs naturally offers a broad range of possibilities for specific and detailed further investigations. In this study, we have chosen a few of these possibilities (e.g., to elucidate characteristics of PV cutoffs in different regions), without claiming that these are necessarily the most relevant and/or interesting ones. This PV cutoff dataset will hopefully be used also by other research groups in the future to learn more about the dynamics of these systems.

Below we address the comments from both referees. Their comments are given in black and our replies in blue.

**Anonymous Referee #1**

**General comment**

The authors present a comprehensive global climatology of potential vorticity of cutoffs that considers various crucial aspects of these structures, including their global geographical frequency distribution, where they are generated and dissipate, their life cycles, tracks, decay and absorption processes, vertical structure, the stratospheric tropospheric exchange associated with them and their links to surface processes. The methodology employed is sound and the fact that the limitations of the tracking algorithm have been outlined is also good. This is a critical contribution to the current body of work in cut-off low climatologies because, not only does it broaden the scope of COL climatologies, it has the potential to contribute to regional predictability studies at the medium range forecasting time scale and shorter. I think that the paper is very well written. I therefore recommend publication subject to the following issues having been adequately addressed.

**Specific comments**

The comments in the section of this are considered major in the sense that they might require additional calculations:

1. Line 125: The feature shown in Figure 3 off the coast of Brazil, is in my opinion, a very important phenomenon for that part of the SH but might be underestimated on the 350 K isentropic surface and previous experience suggests that it is most clearly seen on the 360 K isentropic surface. If one plots the climatology 360 K PV = -2PVU contour for DJF for instance, a clear PV overturning region (or at the very least a region of very strong undulations) is seen in that part of the SH, thus showing the dominance of that structure. I therefore suggest that, at least for the SH, the analysis be extended to include the 360 K isentropic surface.

Thank you for this suggestion. We will extend the whole analysis to isentropic levels from 275-360 K (lower levels are motivated by regions where cutoffs occur even below 290 K, e.g. over the western North Atlantic in DJF).

2. Section 3 is good but could benefit from integrating the effect of the zonal flow. So I suggest that the authors consider superimposing the zonal flow isotachs in Figures 3 to 5 and consider seasonal changes in the flow and the impact of these on the seasonal frequencies discussed in this section. I think that we would see dominance of these frequencies in areas of diffluent and confluent flow. It would also provide an opportunity to view the frequencies relative to the jet core. These can be inferred by the experienced reader, but to render this discussion accessible to those readers who are not familiar with RWB dynamics (as noted in the manuscript, the cut-offs are residues of RWB events), this could improve the readability and impact of the study.

Thanks for this valuable suggestion. We will consider it if it does not compromise the readability of the figure.

3. The discussion in Section 4.1 is very interesting and has interesting and important implications for the various regions considered there in. The region off the coast of Brazil might benefit from such considerations (Figure 18 illustrates the importance of

this region and as reviewer I feel it requires as much attention as the other regions that have been considered in the study). Many studies have shown that it is characterised by cyclogenesis. Please consider an additional regional analysis for that region.

Thanks for this comment and for pointing out the importance of this section. We agree that the region you mention would be interesting to study. However, we had to select a limited number of regions for which we can show and discuss the cutoff characteristics in the paper. This selection is to some extent subjective and we do not claim that the regions we show are the most important ones. On the contrary, we encourage the community to use our dataset to study PV cutoffs in other regions. In this paper, we selected 12 different regions in winter and summer and in both hemispheres, and we demonstrated that this dataset can be used to study regional characteristics of cutoffs. Considering additional regions would make the article too lengthy in our opinion. Therefore, we stick to the current selection.

The presentation of the results of the former section 4.1 will however change substantially in the revised version, in response to the critique of the second reviewer to this section. The regional composites will be replaced by cutoff-centered composites based on the full global dataset of cutoffs. Therefore, even if we agree with you that these figures and discussions are interesting and relevant, we decided to remove them in favour of composites based on the global dataset. However, these Figures are still published in the first author's PhD thesis (Portmann 2020).

4. Could the authors consider composite evolution of at least some of the regions? For instance, with no bias from myself as a reviewer but based on some experience of knowledge of the region thus my use of it to illustrate my point. There exists an interesting phenomenon that occurs in the South Atlantic Ocean/South African domain referred to as the ridging of the South Atlantic Ocean Anticyclone (SAOA) that has been shown to be influenced by lower stratospheric processes. It is most clearly seen in the MSLP pressure fields and is defined as an extension of the SAOA eastward and this ridging process has important implications for the region. Also in South America there is a strong presence of Rossby wave trains that propagate into that continent from the west of Drake Passage, aspects of which could be characterised by this analysis (the 250 hPa anomalies indicate this in Figure 6). I am certain that there are other interesting dynamical processes in the other regions that are unique to those regions that could be alluded to in this discussion, if the authors so please. All in all, I think that evolving composite could reveal some of these regional dynamical issues.

Thanks for this comment. We agree that these would be interesting aspects to study in more detail. But going into the specific dynamical processes involved in the PV cutoff evolution in a few particular regions would clearly go beyond the scope of this study. We hope that such interesting questions will be addressed by follow-up studies based on our climatological dataset.

Minor comments
1. In Fig. 6 please use hatched instead of dashed. Also are the composite means statistically significant in that figure?

Thanks for this comment. As mentioned above, this Figure will be replaced by cutoff-centred composites based on the full global dataset.

2. Line 372 please replace South Pacific with Indian Ocean.

Thanks for this comment and apologize for the mistake. We will change the text as suggested.

**Anonymous Referee #2**

**General comments**

The authors present a global climatology of cutoffs for the ERA-interim period. The introduction nicely motivates cutoffs as highly relevant atmospheric features. The study contains several novel and valuable aspects: I commend the authors on taking a potential vorticity and Lagrangian (trajectory) perspective to identify and track cutoffs. This approach enables to authors to provide a global perspective because the different levels on that cutoffs "live" in different latitudes is automatically accounted for. Furthermore, the 3D aspect of the cutoff evolution is of particular interest.

Many thanks for these positive remarks. Indeed, to the best of our knowledge, this is the first global PV cutoff climatology that systematically investigates their vertical structure, employs a Lagrangian tracking, and therefore can provide global maps of PV cutoff genesis, lysis, decay and reabsorption. These results are shown in Figs. 4, 5 and 15. In addition, the paper shows many results about the lifecycle of cutoffs in specific regions, which could only be obtained due to the methodological novelties of our approach. We therefore regard this study as an important step forward in the relatively long history of research on PV cutoffs.

A further potentially intriguing result, the classification of cutoff life cycles, however, remains mere speculation. My assessment of the manuscript in its current version is much more critical than that of the Anonymous Referee #1 (available online at the time of this writing). To me, the authors miss most of the potential that a feature-based climatology offers to link the evolution of cutoffs to other dynamical features of interest.

We fully appreciate the fact that such a climatology also offers to link the PV cutoff evolution to other features, and we apologize that our study did not meet the referee's expectations. As we mentioned at the beginning of this document, we do not at all regard the analysis of this climatological dataset as completed. There are certainly many important further studies to be done, by other research groups or in collaboration. Due to the methodological novelties (consider cutoffs as 3D features; cutoff tracking) we decided to focus in this paper on the characteristics of the identified PV cutoffs (where do they occur, how long do they live, how far do they travel, why do they decay), and from the many options to relate PV cutoffs to other relevant features, we have chosen to consider their link to surface cyclones. This already produces a rather long paper and we therefore hope that the referee agrees that the fact that more – and potentially also more detailed, more demanding, more

relevant – aspects of PV cutoffs shall be analysed, does not per se reduce the quality of this study, without which all these further analyses would not be possible.

My main conceptual concern here is that the authors stratify the data with respect to small geographical regions and then present "composites" centered on these regions, instead of centering composites on the identified feature (here: the cutoffs), which is an often-used and more powerful approach to feature-based analysis. The authors' geographical approach severely affects the quality of this study. Consequence of this approach are that i) only a small percentage of their data contributes to this analysis, casting doubt on the robustness of the results; ii) the relation to other dynamical features is diagnosed in a rather indirect sense and the differences between regions are often not clear enough to provide sound evidence for the authors' results. Too much of the presentation is thus suggestive, indicative, speculative; iii) pages of presentation are spend on discussing differences between, e.g., cutoffs over the Sea of Okhotsk as compared to cutoffs over the Baltic Sea or the Hudson Bay, making for a tedious, cumbersome read.

First, we would like to mention that we investigated several variants of PV cutoff composites, as there are clearly different options of how this can be done. Most options have their advantages and disadvantages. It is true that with our geographical composites, we were focusing on relatively small subsets for each composite (in order to avoid overly strong spatial smoothing of the relevant structures), but this approach provided the (we think) interesting geographical context. The referee criticizes our geographical approach as "less powerful" and "affecting the quality of the study". We think that this is to a certain degree a matter of taste, and there is nothing objectively wrong with the geographical approach. We fully agree that cutoff-centric composites are also interesting, but we would not dare to make a clear judgement about which approach is more meaningful for the scientific community. However, we decided to follow the referee's advice and we will replace the geographical composites for specific regions with cutoff-centered composites for three cutoff categories. This change goes along with revising our approach to categorize the cutoffs.

Nevertheless, we still keep a (shortened) section that addresses regional aspects of cutoff characteristics. Many previous climatological studies about PV cutoffs are regional, i.e., they focused on PV cutoffs in a particular region of interest, and with our regional approach we can link to these studies. We already had very positive responses from the community that they find it particularly interesting to see results about PV cutoffs in their particular area of interest, and that they can compare with cutoffs in other regions. We therefore would like to keep the regional analysis in our study. However, to address the main concern of the referee we will show a classification according to the jet-relative position using the full (global) set of cutoffs and present composites based on this dataset. They will replace the regional composites but several other regional analyses will be kept. Brief responses to the three specific critiques mentioned by the referee are:
- (i) we will better explain that the regional analyses in total contain about XX% of all cutoffs and are therefore not representative for all regions.
- (ii) the relationship to other features is not the main aim of this study (except for the link to cyclones addressed in section 4.6), and we don't understand why our results should be "too much … suggestive, indicative, speculative".

- (iii) we fully agree that the regional approach makes the paper long and that some parts (e.g., about cutoffs in the northern hemisphere) might be less interesting for a reader mainly interested in cutoffs in the southern hemisphere. But to a certain degree, this is the price to pay for a detailed climatological paper. However, in order to reduce the overall length of the paper, we decided to omit the part about STE (see below).

For a manuscript submitted to a journal with "Dynamics" in its name, I would have hoped that this study provides a much more direct assessment of dynamical characteristics and relationships.

Again, we fully agree that more dynamical questions can and should be investigated in the future. However, this is the first study that quantifies globally the percentage of PV cutoffs that disappear via complete diabatic decay vs. those that disappear via reabsorption, and we show that both fates of PV cutoffs are about equally likely. Whereas the first of these processes has been considered as the most important one at least since the study by Hoskins et al. (1985), we show here that reabsorption of PV cutoffs is much more frequent than previously thought. This has important implications for midlatitude dynamics and predictability, because the process of reabsorption – in contrast to diabatic decay – leads to a nonlinear interaction of the PV cutoff with the PV waveguide and can trigger disturbances that propagate downstream. We regard this specific aspect of our study as particularly relevant for atmospheric dynamics.

I have further concerns with respect to a conspicuous lack of testing the sensitivity of results to (inevitable) free parameters and statistical significance. In addition, I question the accuracy of the authors' computation of non-conservative tendencies as a residuum from diagnosed adiabatic evolution. I elaborate on these criticisms below in the section "Critical specific comments", where I also suggest means to alleviate my concerns. Some of this elaboration duplicates comments given above. In addition, I have several "Major specific comments" for the authors' consideration. A few "Minor specific comments" are provided to parts of the manuscript that I believe do not need to undergo major revisions, but I refrain from providing comprehensive minor and technical comments on the manuscript in its current form. Comprehensively addressing my concerns will require substantial further analysis and revisions. I am optimistic, however, that this work will eventually be publishable. It will then make a very valuable contribution to a weather-system perspective on climate dynamics. I leave it to the editor's or the authors' discretion whether the manuscript should be rejected or withdrawn at this point and re-submitted after substantial revisions, or whether these revisions can be successfully implemented during the limited time of the regular review process.

We address these points below. Some of our answers below repeat parts of our replies to the general comments on the previous pages.

**Critical specific comments**

1) Stratification by (small) geographic regions. The authors stratify the data almost exclusively by (small) geographic regions of cutoff genesis. As the only motivation for this approach, the authors claim that these "regions are selected each such that a broad spectrum of regions is covered" (line 310). No qualification of this claim is

provided. This choice of stratification has several implications that severely affect the quality of the manuscript.

i) Most of the presentation reads like an enumeration of characteristics and pages of discussion are spent on describing, e.g., characteristics of cutoffs over the Sea of Okhotsk as compared to those of cutoffs over the Baltic Sea or the Hudson Bay. This approach makes for a tedious, cumber-some read. More severely, it remains opaque to me what can actually be learned from this kind of presentation.

Apologies for the tedious, cumbersome read. We agree that this is often (most likely also in our case) the downside of providing detailed climatological information. With our revisions, we will do our best to shorten the text where possible, however, we would like to keep the regional analysis in our study. Our main reason is that, as revealed by the literature overview in our introductory section, many previous climatological studies about PV cutoffs are regional, i.e., they focused on PV cutoffs in a particular region of interest. With our regional approach we can link to these studies. We already had very positive responses from the community that they find it particularly interesting to see results about PV cutoffs in their particular area of interest, and that they can compare with cutoffs in other regions. Our results show that indeed, there are regional differences in PV cutoff characteristics and some of them are substantial (e.g., in terms of their vertical extent, lifetime, and link to surface cyclones).

ii) For a manuscript submitted to a journal with "Dynamics" in its name, I would have hoped that this study provides a much more direct assessment of dynamical characteristics and relationships. I would find it much more interesting to learn about how cutoff occurrence, characteristics, and associated transport relates to other features of the large-scale circulation, rather than how they differ in several geographic regions. The authors too rarely attempt to make such direct links to other synoptic-scale features or to characteristics of the large-scale circulation.

Thanks for suggesting an alternative, equally valid set of research questions. We truly hope that studies addressing these questions will be made in the future and we would be most happy to offer our climatology as a starting point for such studies. But in this first study based on this particular methodology to identify and track PV cutoffs we decided to focus on cutoff lifecycles and the, we think, important question of diabatic decay vs. reabsorption. About the critique that such an approach does not provide enough "dynamics", we refer to our reply on the previous page about the dynamical significance of PV cutoff reabsorption, which is quantified for the first time in this study.

iii) The authors do attempt to relate cutoffs to other features of the large-scale circulation in an indirect way by providing "composites" of relevant other features. (It remains unclear if the authors re-center the data on the cutoffs or if they merely provide averages of fields whenever a cutoff occurs in the respective region in subsection 4.1. This reviewer suspects the latter.) From this approach we learn that in some regions cutoffs co-occur 50-70% of the time with feature A and that the occurrence frequency of feature B is 5% higher than climatology. Accordingly, rarely any strong conclusion can be drawn and much of the presentation needs to remain suggestive and indicative. Overall, much variability is found and if statistical significance testing were applied to differences between regions, I believe that little of the signal would stand that test. Strictly speaking, the presented observations are not

incorrect. To this reviewer, however, this choice of presentation gives away much potential of the feature-tracking approach and provides currently little insight.

First, yes these are averages of fields whenever a cutoff occurs in the respective regions. Because these regions are relatively small, this is not dramatically different from feature-centred composites. As mentioned above, this concern will be addressed by an additional analysis that categorizes PV cutoffs according to their jet-relative position and by replacing the regional composites by cutoff-centered composites based on the global dataset.

On a constructive note, I suggest to more fully exploit the potential of the feature-tracking approach by constructing composites centered on cutoffs (as is done in sub-sections 4.3 – 4.5) and, importantly, stratify the cutoffs directly by their relation to other dynamical features, e.g., co-occurrence of wave breaking or streamers, their distance to and location relative to the jet, etc. Another advantage of this approach is that lagged composites can be created easily, too, such that this approach would tell us much more directly how cutoff life cycles associated with, e.g., cyclonic wave breaking differ from that associated with anticyclonic wave breaking. Another approach would stratify the cutoffs by cutoff characteristics or by their relative location to, e.g., characteristics of the storm tracks. I believe that such approaches centering on dynamical rather than geographical features will substantially enhance the significance of the results and will make for a much more succinct and concise presentation.

Thank you for these suggestions. We agree that stratifying the cutoffs according to their relation to other dynamical features is a valuable and potentially insightful approach. However, as stated above, we don't think that it is necessarily "better" than the regional approach. It is just asking a different kind of question ("How does the life cycle of cutoffs with different wave breaking scenarios look like?" vs. "How does the life cycle of PV cutoffs in different regions look like?"). For example, if the reader is a meteorologist from South Africa or California, he/she is likely more interested in the results of the regional analysis than of a global analysis that mixes cutoffs over the US with cutoffs over Europe and Australia. But interestingly and as highlighted strongly in the manuscript, we found that PV cutoffs in different regions with similar evolution also seem to have similar jet-relative positions / wave breaking scenarios. Hence, these two questions are in fact strongly linked. This is not at all straightforward and we consider this as an important result of our study.

In conclusion, and as already mentioned above, we decided to do a further analysis that is along the lines of what the referee suggested above and also considers the issue mentioned in the specific comment 1) about how representable the results from the regional analysis are: We separate all PV cutoffs (full global dataset) according to their jet-relative position (resulting in three types, building on the three types discussed in the current manuscript) and discuss the same life cycle characteristics as discussed in the regional analyses for this more representative dataset.

2) Lack of sound evidence: Presentation is suggestive, indicative, speculative. This point partly relates to comment 1). In addition, however, further analysis within the chosen approach could avoid speculations made by the authors.

In section 3, e.g., the authors speculate on cutoff track characteristics based on genesis and lysis location but do not substantiate this speculation with an analysis of the actual tracks (which are available and presented later in the manuscript).

The aim of section 3 is to provide a first overview of the cutoff tracks, genesis and lysis with some explanations about how these patterns can be understood. A detailed analysis of tracks etc. follows later in the manuscript. We will revise the text and make sure to remove speculative statements that are not fully supported by the figures shown.

Similarly, a claim is made that cutoffs behave differently over land than over the ocean, which could be easily tested with the authors' data.

We assume this refers to the statement "Generally, it seems that in summer lysis is enhanced over land (e.g. in JJA over the Iberian Peninsula and the Rocky Mountains/Pacific Coast Ranges in Fig. 5c), whereas in winter lysis maxima often occur over sea surfaces (e.g. in the Mediterranean and in regions of all western boundary currents, especially over the western North Atlantic and east of Australia; see Fig. 5a,c)".
This statement is a description of the general seasonal pattern of the geographical distribution of lysis maxima shown in Fig. 5. It is not claimed that PV cutoffs over the ocean generally behave differently than over land. It just tells that there is a preference for lysis over the ocean in winter and over land in summer. Hence, we don't see a reason here to provide further analysis or remove the statement.

The analysis of cutoff tracks is a further example: Interpreting and drawing conclusions on different track characteristics from visual inspection of Fig. 8 is a stretch, to say the least. I strongly recommend that the authors reduce speculations and hypotheses to a minimum and extend their analysis to substantiate their conclusions whenever possible.

Thanks for pointing out that the interpretations we make about the cutoff tracks are, from your point of view, not straightforward from the figures shown. However, we think that already the distribution of the lysis points gives a fairly good idea of the mobility of the tracks in different regions. For example, it shows that PV cutoffs forming over California can travel far into the North Atlantic. Or that PV cutoffs over the central subtropical North Atlantic can move westward. In our opinion, this Figure well supports the statements in the text. We will make sure to reduce/remove any speculative statements.

Most severely, I find one of the main results, the proposed classification of cutoff life cycles, to be mere speculation. This result appears prominently in the abstract and the conclusions but, strictly speaking, is not a synthesis of the results presented in the main text. Cutoff characteristics in this study are largely characterized by substantial variability. It seems inconsistent that this large variability can be synthesized into the classification of three types of life cycles that then govern other characteristics (stratospheric-tropospheric exchange) also.
A sounder analysis would have established statistically significant difference between different aspects of cutoff life cycles before presenting such a strong result. Life cycle type II seems to be particularly dubious. In the results section, one and a half sentences are spent to establish the fact that this life cycle is associated with

anticyclonic wave breaking followed by cyclonic wave breaking. I am not aware that this sequence of wave breakings is a well-established concept and I have a hard time to follow the authors description of the respective figure to clearly identify this sequence. The underlying issue is that the classification of life cycles is proposed based on a by-product of mean characteristics in different geographical regions (see comment 1 above). If composites were created, globally not only within the small selected regions, with respect to the wave breaking characteristics, and time-lagged composites were presented that showed the actual mean-evolution, the authors would probably find a much cleaner signal and overall, the presented results would be much more convincing. Such an analysis could establish different types of cutoff life cycles based on sound evidence and not on speculation and would be a great result of this study.

We agree that the cutoff characteristics have large variability. However, we clearly identify substantial differences between regions (e.g. lifetime, jet-relative position, vertical evolution, temporal link to surface cyclones). Also, there are characteristics that are common for some regions (e.g. the downward growth for cutoffs in subtropical ocean basins in summer, the cyclogenesis well before PV cutoff genesis for cutoffs in the storm track regions, jet-relative positions, …). We based our classification on these similarities. We agree that this is not a fully objective classification (as one based on a classification algorithm). However, we argue that it is definitely more than "mere speculation". We also agree that the three types are not able to explain the whole variability of cutoff life cycles – on the contrary, the regional analysis shows that there are also characteristics that strongly depend on the particular region (e.g. the frequent diabatic decay of PV cutoffs over the western North Atlantic).

To address these comment(s) and concerns we will (as already stated above) perform an additional analysis where we categorize PV cutoffs objectively into three types, according to their jet-relative position. In combination with the regional analysis, it will be possible to explain much of the variability of PV cutoff characteristics as a combination of differences between the three types and aspects that depend on regional conditions.

3) Robustness and statistical significance of results.

The authors do discuss limitations of their tracking method but further discussion and tests of the robustness of the results are missing. There are five major issues:

i) About 70% of potential cutoffs are dismissed based on an arbitrary criterion (section 2.3.2). No doubt there is a need to filter out spurious cutoffs. The authors filter out cutoffs that have small vertical extent based on the flawed argument that these cutoffs are not "dynamically relevant". Theory does not support this argument. According to QG-theory, the aspect ratio of a PV anomaly determines the ratio between the associated stability and wind anomalies, but not the extent to which the anomaly is "dynamically relevant". The vertical penetration depth, e.g., depends on the horizontal (not the vertical) scale of a PV anomaly. And the magnitude of a PV anomaly is directly related to the magnitude of the stability and wind anomalies. These two characteristics would thus be more suitable to identify dynamical relevance. In any case, for any free parameter that is introduced to filter out a large

amount/ the majority of potential data, the robustness of the results to reasonable variations of this parameter needs to be demonstrated.

Thanks for this comment. We agree that the depth of the cutoff does not per-se determine if a cutoff is "dynamically relevant" (the terminology "dynamically relevant" was a bad use here, apologies). The main reason for using depth as criterion was to select cutoffs that develop into PV anomalies with a well-established vertical structure, similar to what has been found in many previous case studies or how idealized cases have been designed (e.g., Hoskins et al. 1985). Of course, the size and/or the magnitude of the PV anomaly could also be used to filter out spurious cutoffs. We had analysed already the distribution of these three quantities for different times of the life cycles. In general, the cutoffs removed with the "depth" criterion, tend to be also small and have low PV values. For example, if only a criterion was applied that filters out PV cutoffs that do not reach an average PV value of 3 PVU during the life cycle, the number of remaining cutoffs is almost the same as for the depth criterion, and 70% of them are in both samples. This implies that the results are not extremely sensitive to the selected quantity for filtering.

However, we will carefully reconsider the filtering criteria. We will also provide some information on the robustness of the results regarding the selected criterion.

ii) After showing the geographical locations of genesis, lysis, and tracks (not that I am mistaken: I consider this an important documentation of the occurrence characteristics of cutoffs.) the authors restrict their analysis to small geographic subregions during DJF and JJA. These subregions constitute maxima in genesis frequency, but no information is given what percentage of total cutoffs occur within these regions. Visual inspection of Fig. 4 suggest that this percentage is low. (It's hard to estimate a number but I guess less than 20%.) Cutoffs during MAM and SON are not considered. So overall only about 10% of the data is used to draw conclusions from. In combination with the lack of justification of why these 10% are selected I fail to see how the results can be considered to be robust and how the results would meet the expectation of a global climatology, as suggested by the title.

First of all, not all results of the study are based on the regional subset. For example, the frequencies of decay and reabsorption as well as analyses of STE are based on the global dataset. But we understand the reviewers concern and reply as follows.
  (i)    As mentioned above, we will add a new categorization that considers all cutoffs.
  (ii)   We will mention the percentage of the cutoffs used for the regional analysis.

iii) Basing the tracking of cutoffs on trajectory calculation is a major strength of this study. The trajectories are used also to determine the adiabatic evolution of cutoffs. Non-conservative contributions to the evolution, including stratospheric-tropospheric exchange, are estimated as the difference between the observed and the diagnosed adiabatic evolution, i.e., as a residuum. Estimating non-conservative terms as a residuum requires a very accurate diagnosis of the conservative dynamics and is usually associated with substantial uncertainties in the results. No discussion of this issue is provided in the manuscript. Trajectories are calculated for 6h, which is seemingly short but in fact 6h is the temporal resolution of the underlying data. This means that trajectories, irrespective of the time step used in the calculation, are

based on linear interpolation between the start and end times. Linear interpolation is a poor approximation when the displacement of strong gradients needs to be represented, here: the movement of the strong PV gradient and associated winds associated with the jet, wave breaking, and the cutoff itself. In addition, any nonlinearities in the adiabatic 6h evolution will erroneously be diagnosed as non-conservative contributions. Such nonlinearities can be expected during genesis (wave breaking) and reabsorption of cutoffs and I wonder to what extent putative maxima of non-conservative processes at the beginning and the end of cutoff life cycles, e.g. the u-shape signal in Figs. 16 and 17, are related to inaccuracies in the trajectory calculations. The robustness of the results should here be demonstrated preferentially by using higher-resolution ERA5 data for a sufficiently large subset of cases or, as a less accurate but computationally less expensive option, by reducing the temporal resolution of the ERA-interim data to 12h.

iv) The authors examine stratospheric-tropospheric transport (STT and TST) based on 6h trajectories. Previous authors considered results based on this relatively short time scale as spurious and advocate using longer timescales. For this issue, the authors do provide a discussion in their final summary and discussion section. To avoid readers' confusion, I would find the discussion more helpful closer to the method or the results section. More severely, I am not sure that I find the comparison of rather general characteristics with previous results to be convincing enough to demonstrate that the transport characteristics diagnosed in the current manuscript for different "types" of cutoffs is robust to the time scale used in the trajectory calculation. A sound test of the robustness would extent the trajectories used in the analysis to longer times, again at least for a sufficiently large subset of cases.

Reply to comments iii) and iv):
Thank you for pointing out the strength of the study that uses – as the first of its kind - trajectory calculations for the tracking of PV cutoffs. The reviewer criticises the estimation of stratosphere-troposphere exchange (STE) as deviation from the adiabatic evolution as well as he use of 6-hourly trajectories for this purpose. We point out here, that it is not the goal of the study to provide an overall quantification of STE related to cutoffs. STE is used here mainly as estimation for the growth/decay rate of cutoffs and in this sense, the absolute values are less important here. We fully agree that the estimation of STE with this method is subject to errors but we found that it reasonably well describes the growth and decay of cutoffs.

However, in order to reduce the length of the article and enhance its focus, we decided to remove the analyses of STE. As a consequence, in the new version, the trajectories will only be used for the tracking and to identify diabatic decay and reabsorption.

v) There is a complete lack of testing the statistical significance of characteristics of cutoffs in geographical regions/ different types of cutoffs. This lack is significant because the descriptions/ results rely on the statistics of the data only and it is often not clearly evident to the trained eye that differences are sufficiently large. The revised version of the manuscript requires such tests.

Thanks for mentioning this point. However, we are not sure that statistical tests could be easily done and would be meaningful. Some of the regional aspects are different by design (e.g., the tracks shown in Figs. 8 and 9 in the original manuscript); and

other differences are obvious (e.g., the regional differences in the preferred isentropic levels shown in Figs. 10 and 11, or the differences in how often and when cutoffs are related to surface cyclone – see Figs. 19 and 20). In our view, a statistical test would not reveal important information in addition to what is obvious from these panels. And, last but not least, it is not a key objective to document that cutoffs in region A are "different" from cutoffs in region B, we rather would like to portray the spectrum of characteristics of cutoffs and appreciate their fascinating variability. In the revised version, we make sure that our intention to show the regional analyses becomes clearer to the reader. If beneficial, we will also consider statistical tests for the results of the new classification according to the jet-relative position.

**Major specific comments**

1) The conclusion that can be drawn from this study are hard to identify. The final section contains both, an extensive discussion of results in the light of previous studies and the conclusions drawn from the current study. As a consequence, the final section is unusually long and the summary is little helpful for a reader who is interested in the main outcomes of this study. To me, most of the discussion of previous result would have been more useful in the results section, after presenting the relevant result. This re-organization of the text would improve readability of the results section because more context for the results presented therein is provided. More severely, in the current version, it is hard to identify what the main results of the current study are. If the authors chose to keep a long final discussion section, then at least final conclusions should be clearly separated from the comparison with previous studies.

Thanks for pointing out that the conclusions were not clear to you. First, we consistently separated the discussions from the results to achieve a good readability and flow. Apparently, we were not able to convince the reviewer with this structure. We will carefully reconsider the structure of the manuscript and make sure the main conclusions are highlighted in the end. Due to the new categorization, which we will present earlier in the paper, the final section will also be shortened and therefore, hopefully, becomes clearer.

2) Section 3. It is hard for me to see what we learn in section 3. I understand the importance to show and describe the geographical distribution of feature occurrence and characteristics. In its current version, however, the presentation is mostly show and tell and thus rather unsatisfactory. One issue that can be remedied is that the authors partly use very specific geographic references (e.g. New Foundland, Iceland). First, an accuracy of the analysis is suggested that I believe is not robust to reasonable changes of free parameters used for cutoff identification. Second, using more general geographic references (over land – over ocean, subtropics – polar – midlatitudes) seem more related to potential differences in the dynamics. Most preferably, a description in relation to large-scale features (start – end of storm tracks, at the edge of subtropical anticyclones) would be given.

We would like to point out here that the maps shown in section 3 are the very first of its kind. Nobody has ever shown such comprehensive climatological maps of cutoffs (i.e. including the whole globe and all (or most) relevant vertical levels. We even find frequency maxima in regions that previously have not been documented to be

regions of frequent cutoff occurrence. These facts alone justify, in our opinion, to simply describe these geographical patterns in detail.
The reviewers second critique questions the accuracy of the results because of the filtering criterion. As already noted above, we will reconsider the filtering criterion and comment on the sensitivity of the results to this criterion. We already point out here that the frequency maxima are robust to this filtering, i.e. the unfiltered dataset shows frequency maxima at the same locations. Therefore, we consider specific geographical references as useful and appropriate to describe the geographical distribution of cutoffs. Terminologies like "start – end of storm tracks, land/ocean" can be considered useful for generalization of the geographical patterns. Reviewer 1 also suggested to relate the geographical frequencies more to jet positions etc. We also point out that we generalized the geographical distribution of cutoff lysis in different seasons with the terms "over land" and "over the ocean" but this statement was strongly criticized by the reviewer (see specific comment iii).

In conclusion, we will keep this section generally as it is, but likely add some information of the climatological large-scale flow and complement the text with statements that relate the PV cutoff frequencies to the large-scale flow and/or other general patterns.

3) Subsection 4.1: Cutoff genesis. Unfortunately, I have a very hard time to follow the authors' arguments in this sub-section based on occurrence frequencies of other features. In addition, I struggle to clearly identify the features described in the text in the figure. In particular, I struggle to identify wave breaking, which plays a crucial role in the authors' argument. Do the authors mean to indicate different types of wave breaking by tropospheric and stratospheric streamers? Inter alia in this particular subsection the authors draw strong results based on rather speculative arguments. To illustrate the temporal evolution leading up to genesis, time-lagged composites (going back in time) could be created, in a feature-centered framework e.g. centered on genesis location. Similar composites are used later to illustrate temporal evolution and are very helpful (Figs. 10 ff.). I recommend substantial revision of the complete subsection.

Thanks for noting that it was difficult for you to follow the arguments in this section. Apologies if it was not stated clearly that we identify different types of wave-breaking with PV streamer frequencies in combination with upper-level wind speed (to illustrate the barotropic shear of the environment in which the cutoffs form). As already mentioned above, we will revise this section and replace the composites by cutoff-centered composites based on the full dataset.

4) Subsection 4.2: Track analysis. Visual inspection alone is insufficient to draw conclusions about different types of tracks. A cluster analysis of tracks, or at least showing the average tracks of cutoffs in distinct end regions are required to make a point here.

We agree that there are some statements that refer to track characteristics (e.g. the path of cutoffs forming over the Mediterranean) that are difficult to verify from the Figures. However, for other regions (e.g. South Africa, California, ...) such general track directions are well visible in our opinion. Also, many statements in this section refer to the distance of the lysis positions relative to the genesis box, which is an

information that can be inferred straightforwardly from the plots. To make this analysis more robust and clear, we will add average tracks for each region. A cluster analysis is definitely far beyond the scope of this section, which aims to provide a general overview of the paths taken by cutoffs in the different genesis regions.

5) Subsection 4.3 – 4.5 are the strongest result subsections but still suffer from the general critical issues noted above.

Thanks, we will keep these sections generally as they are (except that we remove the section about STE, see comment above). The general critical issues have been addressed already above.

6) Subsection 4.6: Link to surface cyclones
There have been decades of research based on first principles to understand the interaction between upper-level PV anomalies on surface cyclones. What do we learn here about this interaction by considering mere proximity of the systems? What aspects of the interaction do you consider? Why is it relevant to consider the age of the cyclone? In what sense does the analysis provide new insight? What is the significance of this result? I see potential in this subsection but there is a severe lack of motivating the important questions and why proximity of the systems is a useful metric. Potentially, the issue here is mostly a matter of wording. The current version, however, leaves the reader guessing why this subsection would provide interesting results.

Thanks for pointing out that the motivation for this section is not clear to you. We agree that the introductory paragraph in this section is relatively short and does not clearly motivate why we study this temporal link. We will revise this section to make its motivation clearer to the reader. In short, the main motivating aspects are:

• It is well known that upper-level PV anomalies (such as PV cutoffs) can lead to surface cyclogenesis via their dynamical forcing. However, it is not known how frequently this occurs and if regional differences exist.
• Surface cyclones are of major importance for surface weather. Hence, a PV cutoff co-occurring with a surface cyclone is an indication for substantial surface weather impacts related to the cutoff.
• Also, it is known that the formation of PV cutoffs / Rossby wave breaking can go along with surface cyclogenesis. It is also known that different wave breaking types are linked to different surface cyclone evolutions.

By investigating the temporal link of PV cutoffs and surface cyclones we can identify (i) PV cutoff related cyclogenesis, (ii) assess the relevance of PV cutoffs for surface weather, and (iii) gain insight into the temporal evolution of the upper-level and lower-level baroclinic wave signals. For example, the fact that PV cutoffs with genesis in storm track regions, which form from cyclonic wave breaking, are frequently linked to surface cyclones that form a few days prior to PV cutoff genesis provides new insight about the consequences of cyclonic wave breaking.

**Minor specific comments:**

- Fig. 1 is very helpful, but Fig. 2 did not help me to better understand the tracking procedure.

Thanks for this comment. It is difficult to understand from it which aspect of Fig. 2 was not useful/confusing. We will try our best to improve the description of the tracking procedure and associated figures.

- The introduction reads in general very well. Except for the two middle paragraphs on page 3. There it is not clear to me what the open question is and/ or how the respective paragraph motivates the current study. Please clarify.

Thanks for this comment and that you like the introduction in general. The first of the two mentioned paragraphs should state that geographical frequencies are well known from previous climatologies but that they all suffer from the level dependency. This level dependency is resolved in this study.
The second paragraph should state that there are similar systems poleward of the jet stream, which are often not considered as classical cutoffs. This study identifies also these systems and is therefore more comprehensive than many previous studies. We will make sure that the motivation for these paragraphs becomes clearer.

- How do you handle a lost child? Are they simply ignored? If yes, how much of the volume of cutoffs are dismissed due to a lost child?

A lost child is ignored in the sense that the track does not continue with it. However, it can be the start of a new track. The mass is quantified as «splitting» and the corresponding mass fluxes are relatively low (compared to STT and TST mass fluxes).

- I fail to understand the first half of subsection 2.3.2 (Construction of tracks).

Thanks for this comment. We will do our best to make this part as understandable as possible (see comment above).

- What motivates your choice of definition of cutoff area? Why not using the vertical average of the area on the individual isentropes? A single isentrope with large area would dominate your metric and I would not consider the area as representative in that case. Or take, e.g., a tilted structure with a similar area at all levels. Your method artificially enlarges these features. Please add some discussion for clarification.

This is a pragmatic choice and we do not think that there is a right or wrong here as long as we are aware of how the area is defined. In most cases, this area will be the area of the highest isentrope where the PV cutoff is present. Cutoffs that are tilted so strongly that the area is substantially larger than on the highest isentrope, are rare. However, this area definition is only relevant for the computation of STE, which will be removed in the revised version of the manuscript.

- Avoid technical language when describing a physical feature. E.g., the term "cutoff track" is often used to describe what I believe is simply the life time of the cutoff (e.g., line 301 and in subsection 4.3

On line 301 we change «PV cutoff tracks» to «PV cutoffs». However, in subsection 4.3. «PV cutoff track» is the most accurate description in our opinion.

---

## Author Response (AR2)

**The three-dimensional life cycle of potential vorticity cutoffs: A global ERA-interim climatology (1979–2018)**

**wcd-2020-30**

by R. Portmann, M. Sprenger, and H. Wernli

**Author comments**

We thank the three reviewers for their careful evaluation of the revised manuscript. We are pleased to see that this version has been more convincing and easier to read than the initial submission.

Additional minor changes have been made to the manuscript according to the reviewers' comments. We address all comments separately below. The comments by the reviewers are given in black and our replies                                                   in                                                   blue.

**Reviewer 1**: accept as is.

**Reviewer 2:**

Prelude

First of all, it is important for me to state that the tone of my first review was certainly not meant to be harsh. In fact, I took time and efforts and had tried to maintain a constructive note throughout (although, to me, the initial manuscript was a long, tedious read). It appears to me that my suggestions, which meant to be constructive, were misinterpreted as an attempt to impose my "personal viewpoint of how such a climatological study should have been performed". Of course, the authors are free to perform their analyses to their liking and to focus on aspects that they consider to be the most relevant for the current study. My reviewer's obligation, on the other hand, is to check the evidence that is provided by carefully understanding the authors method, description of figures, explanations and conclusions drawn. If I as a reviewer fail to do so too often, I feel obliged to provide (very) critical comments

The tone of my initial review might have been very critical at times because I *was* very critical about the initially submitted version. I am not sure that I consider it helpful, though, to point out a critical tone of a critical review in an authors' response. As a reviewer I can only evaluate the manuscript at hand, i.e., the information, explanations, motivations, interpretations provided in the manuscript. Lack of clarity in writing, omission of information, etc. may add up to the extent that the intentions of a study do no longer become sufficiently clear to a reader. From the authors' response I gather that some of the main intentions of this study did not become clear to me, which has certainly led to much criticism from my side. I understand that the authors may thus have considered my first review to be overly critical. I would hope that the authors understand that my misinterpretations of their initial manuscript were at least partly due to lack of information and clarity and a suboptimal organization of the presentation of the material in the initial submission.
A word on "expectations": I had used this term somewhat carelessly – or at least ambiguously – in my first review. I certainly did not mean to imply that the authors need to meet my expectations as in standards that would need to be met. I meant to use the term to refer to a tacit context that one may

have in mind when reading a manuscript. That tacit context was formed for me by the name of the journal (including the term "dynamics") and the title of the manuscript ("global climatology"). Without grasping the motivation of the authors, I had evidently a very hard time to fit a detailed and mostly descriptive discussion of very regional aspects into that tacit context. I agree that my wording to express this idea in the first review was poor.

We thank the reviewer for his/her remarks. We fully agree with the reviewer's view about the obligation to provide critical comments. In fact, most of the comments helped us a lot in reshaping and improving the paper, in particular its structure and the readability of the results.

Comments on the revised version

The manuscript has improved tremendously in all aspects. This reviewer finds it extremely helpful that there is now a clear distinction between the global aspects of the climatology (as heralded in the title) and regional aspects. In addition, there is now a clear motivation of why these regional aspects are of interest and why detailed descriptions are well justified. Furthermore, I find it extremely helpful that the classification of life-cycle types is now based on the global data and that there is more discussion of relevant previous work that puts the authors contribution into much better context than in the previous version of this manuscript. More, clearer, and better motivated links to other atmospheric phenomena, both in terms of the larger-scale circulation and potential impacts, further strengthen this study. Adjustments to the filtering of the data and some reference to statistical significance round off what is now a very well written quality manuscript.

Basically, the manuscript could be published as is. I have one remaining question, though, and addressing this question may somewhat alter one of the authors' conclusion. I thus recommend accepting this manuscript for publication after minor revision. In addition, I have a few suggestions for the authors' consideration that may help to provide full clarity and further improved readability for future readers.

Thanks for this very positive assessment of our manuscript. We respond to the mentioned aspects below.

Minor comment

Sect. 3.2: The authors compare diabatic decay with reabsorption. I would think that diabatic decay, i.e., diabatic PV erosion, is a gradual and relatively slow process whereas reabsorption is identified at the moment when the 2PVU contour merges and thus I would thus think that this is a very fast process. If my reading is correct, the authors evaluate the two processes instantaneously at the time at which a PV cutoff disappears. Then the relative importance/ frequencies of the two processes are compared. It seems to me, however, that the instantaneous evaluation of the gradual diabatic PV erosion does not do this process full justice, i.e., that the relative importance of processes is biased towards reabsorption. I understand that the 3D cutoff may gradually decay diabatically because the definition here is layer-wise on each isentrope. But does this layer-wise perspective fully take into account the gradual nature of diabatic erosion? My feeling is that a fairer comparison would integrate both processes over the lifetime of the cutoff. I believe that it would be helpful if the authors commented on this issue, in particular because their conclusion based on this result is rather different from synoptic experience (as noted by the authors).

Thank you for mentioning this aspect. First, we would like to note that we do integrate decay and reabsorption along the entire lifetime of the cutoff. Let's look at the following example: A PV cutoff

forms at 310 K and 320 K and diabatic decay occurs after 2 days at 310 K but the cutoff persists at 320 K and is reabsorbed at that level after 3 days. In this case, our approach both considers one event of diabatic decay and one of reabsorption.

However, we agree that, considered on a single isentrope, diabatic decay is a gradual process and identifying decay as disappearance from an isentropic surface does not account for the fact that the actual loss of mass may have occurred at an earlier time. Viewed on a single isentrope, a PV cutoff may start to shrink (i.e., lose mass due to stratosphere-to-troposphere transport) several hours/days before it actually disappears from the isentropic surface. Also, it can occur, for example, that a PV cutoff shrinks for a certain time period but grows again thereafter (due to troposphere-to-stratosphere transport) and is finally reabsorbed. We quantified these potentially complex evolutions of "shrinking" and "growing" with our STT/TST diagnostic, which we presented in the original paper but removed in the current version to enhance the focus of the paper. These results, however, are documented in the openly available PhD Thesis of the first author: https://www.research-collection.ethz.ch/handle/20.500.11850/466735). For the analysis presented in the paper, we argue that growing and shrinking of a PV cutoff is different from the actual lysis scenario (decay and reabsorption). Of course, this is clear for reabsorption, but it may appear a bit artificial for decay (as decay occurs basically at the end of a period of strong shrinking).

We reconsidered this aspect but still, in our opinion, defining decay and reabsorption as two different scenarios of how a PV cutoff can "disappear from an isentropic surface" is the most appropriate and fairest one. This definition does to some extent incorporate the gradual nature of both, decay and reabsorption, in the sense that it identifies these scenarios separately on each isentropic surface. Our approach can capture the evolution of a PV cutoff that decays first on, e.g. 310 K, then 315 K etc., as well as of a cutoff that is reabsorbed first on 330 K, then on 325 K, etc.

Finally, there may be several reasons why the result that reabsorption is as frequent as diabatic decay deviates from synoptic experience:

1) Synopticians may focus on "classical cutoffs" equatorward of the jet stream for which decay is indeed more frequent than reabsorption.
2) They may focus on lower isentropic levels, on which the cutoff is typically well separated from the main stratospheric reservoir. At higher levels, where the cutoff is closer to the reservoir (because there the reservoir larger), reabsorption is more frequent and can also occur transiently (i.e. attachment to the reservoir is followed by detachment). The PV evolution on these levels often looks "messy" and complicated and most PV cutoff studies therefore focus on the lower levels where diabatic decay dominates.

Comments for further consideration

Title: The authors may want to consider extending the title to point potential future readers also to the regional aspects, which form a substantive part of their study.

Thanks for this suggestion. We adopted the title to: The three-dimensional life cycles of potential vorticity cutoffs: A global and selected regional climatologies in ERA-Interim (1979-2018)

l24: suggest adding „as defined above" after "types" for full clarity.

Suggestion has been adopted.

top of page 7: For full clarity: It would be helpful to explicitly define STT/TST as the PV change of an air parcel that crosses the threshold of 2 PVU because not every reader may be familiar with this. (I might have missed such a definition above in the manuscript.)

Thanks for pointing this out. We added the following sentence: "Note that here, STT is defined as the Lagrangian PV change of an air parcel from more to less than 2 PVU, and vice versa for TST."

Sect. 3.1: I fully agree with the authors that the in-depth discussion in this subsection is justified. For the reader's convenience, the authors may want to consider introducing subsubsection for, e.g., frequencies, genesis/ lysis, and comparison with previous studies. The discussion at the end of this subsection now motivates very well the classification of life-cycle types presented below.

Thanks for this very useful suggestion. We introduced subsubsections to better structure this rather long subsection.

l407: "During lysis of 3D PV cutoffs": It would be helpful to clarify: Is this the instance at which the 3D cutoff completely disappears, e.g., the moment at which the cutoff disappears from the last remaining isentrope?

Thanks for suggesting clarification. We added the following statement: "During lysis of 3D PV cutoffs, i.e., if only the last time step of each PV cutoff life cycle is considered, …"

l409ff: reabsorption occurring predominantly at higher levels: Just a thought for the authors' consideration: What you describe here, is that consistent with the partial erosion of PV at the lower levels of the 3D cutoff and the final decay by reabsorption then being associated with the remaining parts of the cutoff at higher levels?

Thanks for this way of rephrasing. Yes, this is a possible evolution. First diabatic decay at lower levels and then reabsorption at higher levels. However, it could also be the other way around: reabsorption occurs first at higher levels and then decay at lower levels. It would be an interesting further study to look at the temporal sequence of decay and reabsorption during the life cycle of a 3D PV cutoff in detail.

l420: for full clarity: I suggest adding „of the 3D cutoff" after "life cycle".

Thanks for this suggestion, we adopted it.

Sect. 3.3: denomination of life-cycle types: The authors may want to reconsider their life cycle names and consider introducing more telling acronyms, e.g., just for illustration, types POL, 2J, EQ. Not being familiar with the authors' life-cycle terminology, I found myself switching back and forth while reading to remind myself how types I, II, III are defined.

We agree, roman numbers are not intuitive. However, we also failed in finding really good acronyms and we therefore decided to remind the reader once per section about the meaning of types I, II and III by writing "Typ I (equatorward of jet), Typ II (between jets), Typ III (poleward of jet)".

l581: "active cutoff tracks". I am not sure I fully understand the meaning here. Consider revising.

Thanks, we clarified with "(…) number of cutoff tracks that still exist at a certain time after genesis (hereafter: active tracks)".

l736ff: The discussion here reminds me on a third type of baroclinic life cycle that was noted in Shapiro et al.'s chapter in the book The Life Cycle of Extratropical Cyclones. The third author of this study was a co-author of that study so he can best evaluate if reference to that work is relevant here.

In retrospect, introducing LC3 in this book chapter was maybe misleading, and the suggested 3-type terminology has not been picked up by the community. We think it is safer to not refer to LC3 in our paper.

**Reviewer 3:**

This is my first review of this manuscript, although it has already been through one round of reviews.

This is a thorough and comprehensive analysis of PV cutoffs that includes consideration of their 3-d structure, which has not been attempted before. The method used to track the features (trajectory analysis) also enables the authors to determine in what way the PV cutoffs are disappearing at the end of their lives – do they decay or are they reabsorbed. The method is complex but is very clearly described, and I appreciated the schematic diagrams to help with this. A global climatology is presented, which shows the locations of cutoffs and their general characteristics. The features are then grouped according to their position relative to the jet, which yields three quite distinct classes with different characteristics and surface impacts. I particularly like that this way of categorizing the PV cutoffs allows the authors to demonstrate the importance of the two baroclinic life cycles, LC1 and LC2. Finally the authors show variations in the PV cutoffs in different geographical regions. This is complementary to the other type of classification used.

I know the manuscript has undergone significant reworking after the first reviews, and I find it to have a very clear structure. The conclusions are nicely presented to demonstrate the key takeaway messages of the paper. Overall I very much enjoyed reading it and think the dataset will be very useful to the community.

I only have a few very minor suggested edits.

Thank you very much for this feedback on our work and that you enjoyed the reading. And many thanks for accepting to carefully read the paper in the second round. We address your minor points below.

1. Line 57: Meditarranean > Mediterranean.

2. Line 163: do > to?

3. Line 173: foreward > forward.

Thanks for the above corrections, we adopted them.

4. Lines 203-205: I find this sentence confusing. Please can you reword or clarify?

We are not 100% sure which part is confusing. We tried to rephrase the sentence and hope it is clearer now.

5. Lines 407-408: I think this sentence needs some more commas to be clearer.

We added two more commas: "During lysis of 3D PV cutoffs, reabsorption is, with a share of 54 %, even a little more frequent than diabatic decay"

6. Figure S1 – I think this would be useful to have in the main paper rather than in the supplement.

We agree that Figure S1 is important and could also be placed in the main manuscript. However, the manuscript has already 17 Figures, which is quite a lot. In addition, there is some redundancy in the information in this Figure and Figure 3 (both contain climatological frequencies of cutoff occurrence). Therefore, we decided to reduce the information on the frequencies of the three types to descriptions in the text with reference to the supplement.

7. Line 669: Is the 10% value given found by adding the values in the two boxes (0.5-1 and 1-1.5)? I was a bit confused initially – could you clarify this?

Thanks for pointing towards this slight confusion. We clarified it.

8. Line 674: Sentence starting "A similar scenario…" is unclear and I think needs rewording.

Thanks, we rephrased this sentence.